# Mesoscale contribution to the long-range offshore transport of organic carbon from the Canary Upwelling System to the open North Atlantic

Elisa Lovecchio[1], Nicolas Gruber[1], and Matthias Münnich[1]

[1]ETH-Zürich, Universitätstrasse 16, 8092 Zürich, Switzerland

*Correspondence to:* Elisa Lovecchio (elisa.lovecchio@usys.ethz.ch)

**Abstract.** Several studies in upwelling regions have suggested that mesoscale structures, such as eddies and filaments, contribute substantially to the long-range transport of the organic carbon from the nearshore region of production to the offshore region of remineralization. Yet a comprehensive analysis of this mesoscale flux and of its impact across the Canary Upwelling System (CanUS) has not been provided. Here, we fill this gap using simulations with the Regional Oceanic Modeling System (ROMS) coupled to a Nutrient, Phytoplankton, Zooplankton, and Detritus (NPZD) ecosystem model. We run climatological simulations on an Atlantic telescopic grid with an eddy-resolving resolution in the CanUS. Using both a Reynolds flux decomposition and structure-identification algorithms, we quantify and characterize the organic carbon fluxes driven by filaments and eddies within the upper 100 m and put them in relationship to the total offshore transport. Our analysis reveals that both coastal filaments and eddies enhance the offshore flux of organic carbon, but that their contribution is very different. Upwelling filaments, with their high speeds and high concentrations, transport the organic carbon offshore in a very intense, but coastally-confined manner, contributing nearly 80% to the total flux of organic carbon at 100 km offshore. The filament contribution tapers off quickly to near zero values at 1000 km off the coast, leading to a strong offshore flux divergence that is the main lateral source of organic carbon in the coastal waters up to 1000 km offshore. Some of this divergence is also due to the filaments inducing a substantial vertical subduction of the organic carbon below 100 m. Owing to the temporal persistence and spatial recurrence of filaments, the filament transport largely constitutes a time-mean flux, while the time-varying component, i.e., the turbulent flux, is comparatively small. At distances beyond 500 km from the coast, eddies dominate the mesoscale offshore transport. Although their contribution represents only 20 % of the total offshore flux and its divergence, eddies, especially cyclones, transport organic carbon offshore to distances as great as 2000 km from the coast. The eddy transport largely represents a turbulent flux, but striations in this transport highlight the existence of typical formation spots and recurrent offshore propagation pathways. While they propagate slowly, eddies are an important organic carbon reservoir for the open waters, since they contain, on average, a third of the organic carbon in this region, two third of which is found in cyclones. Our analysis confirms the importance of mesoscale processes for the offshore organic carbon transport and the fueling of the heterotrophic activity in the eastern subtropical North Atlantic, and highlights the need to consider the mesoscale flux in order to fully resolve the three-dimensionality of the marine organic carbon cycle.

# 1 Introduction

The Canary Upwelling System (CanUS) is one of the four major Eastern Boundary Upwelling Systems (EBUS), i.e., coastal regions along the western boundaries of the continents characterized by equatorward winds inducing an offshore Ekman transport. This causes the upwelling of cold, nutrient-rich water in the nearshore region, fueling intense biological activity near the coast (Chavez and Messié, 2009; Carr, 2002). This coastal upwelling is embedded within the equatorward flowing branch on the eastern side of the subtropical gyres, i.e., the relatively sluggish eastern boundary currents. Coastal shear and irregular topography, obstacles such as islands, and the density gradient generated by the upwelling of cold waters at the coast produce substantial variability in the flow of the coastal currents, generating instabilities that give rise to mesoscale fronts, filaments, eddies and other forms of turbulent structures (Barton et al., 1998; Capet et al., 2013). This variability, especially that on the oceanic mesoscale (scales of $\sim$20 km to $\sim$200 km), modulates the spatial distribution of tracers with an important impact on the biological activity (McGillicuddy, 2016; Mahadevan, 2014; Rossi et al., 2008; Gruber et al., 2011). Further, it is expected to have an important role in the offshore transport of the coastally-produced organic matter, fueling the biological activity of the oligotrophic open waters (Álvarez-Salgado, 2007; Sangrà, 2015; Pelegrí et al., 2005).

Coastal filaments are narrow ($<$50 km wide) structures that extend from the coast to several hundred kilometers into the open sea and are characterized, in their interior, by rather large flow velocities (between 0.1 and 0.5 m s$^{-1}$), often recirculating the water in vortices at their extremity (Strub et al., 1991). Transport by filaments is typically accompanied by a deepening of the tracer fluxes in the offshore direction, due to the high density of the upwelled water (Cravo et al., 2010). This is also associated with intense downwelling induced by the generation of ageostrophic cross-frontal circulation at the edges of the filament (McWilliams et al., 2009; Nagai et al., 2015). Filaments, with their high concentration of organic carbon (C$_{org}$), have been shown to export laterally about half of the coastal production from their region of origin (Gabric et al., 1993; Pelegrí et al., 2005; Arístegui et al., 2003). Locally, this transport can exceed the mean Ekman transport in the nearshore several times (Rossi et al., 2013; Álvarez-Salgado, 2007). The laterally exported C$_{org}$, part of it in the form of particulate organic carbon (POC) and part of it in the form of dissolved organic carbon (DOC) (García-Muñoz et al., 2005), may accumulate then in the oligotrophic open ocean regions, eventually fueling heterotrophic activity there (Álvarez-Salgado, 2007). In addition to C$_{org}$, filaments are also responsible for the lateral export of chlorophyll, nutrients (Cravo et al., 2010) and living organisms (Brochier et al., 2014) to the open sea.

Long living mesoscale eddies generated by coastal instabilities, irregular topography and obstacles such as islands are responsible for the long-range transport of physical and biogeochemical properties (Stammer, 1998; Zhang et al., 2014; Amores et al., 2017). These non-linear structures propagate with velocities of a few centimeters per hour, about one order of magnitude slower than the filaments, while rotating much faster around their center (Chelton et al., 2007; Schütte et al., 2016a; Klocker and Abernathey, 2014). Stable eddies can live for several months up to years, and therefore propagate for hundreds or even thousands of km from their region of origin (Chelton et al., 2011), despite their low translational speed. They thus reach substantially farther offshore than the filaments (Sangrà et al., 2009; Combes et al., 2013). Eddies trap water and tracers in their core during their formation. In stronger eddies, the degree of lateral isolation of the eddy core from the surrounding environment

can be quite high, possibly resulting in the entrainment and long-range transport of trapped tracers at formation (Chelton et al., 2011; Karstensen et al., 2015). Through the initial trapping of tracers, and the subsequent upwelling/downwelling, mixing, and the interaction with external forcings such as wind (McGillicuddy, 2016; Gaube et al., 2015), mesoscale eddies produce important perturbations of chlorophyll, phytoplankton biomass and all the other biogeochemical properties in the euphotic

layer (Pelegrí et al., 2005; Gaube et al., 2014). This has strong impacts on the local biogeochemical fluxes and ecosystem composition (Baltar et al., 2009; Doblin et al., 2016; Rossi et al., 2008, 2009). Due to the long life span of the eddies, the isolation of their cores, and the substantial biogeochemical transformations, the tracer composition of the eddy center as well as the eddy community structure on many trophic levels may be substantially different from that of the surrounding waters (Löscher et al., 2015; Karstensen et al., 2015). As a result, the eddy can considerably modify the properties of the surrounding

waters when it releases its content upon its death (Mahadevan, 2014; Stramma et al., 2013).

Relative to the other EBUS, the CanUS has a moderate level of eddy activity (Lachkar and Gruber, 2011), but hosts some of the largest filaments (Ohde et al., 2015). Mesoscale activity within the CanUS differs substantially between the different subregions. This is a consequence of the complex circulation pattern that characterizes the region (Mackas et al., 2006; Sangrà, 2015). The Cape Verde frontal zone, along which the coastal Canary and Mauritanian currents are deflected offshore, defines

a natural boundary for the flow of water masses and tracers in the region (Pelegrí and Peña-Izquierdo, 2015). This front separates the CanUS into a northern and a southern sector that differ in both biological activity, seasonality of the upwelling and circulation (Arístegui et al., 2009; Pelegrí and Benazzouz, 2015).

Most of the coastal filaments in the CanUS are observed north of the Cape Verde front, generally associated with the numerous capes that characterize this part of the CanUS (Arístegui et al., 2009; Pelegrí et al., 2005). The most prominent

filament in the northern subregion is that associated with Cape Ghir. This quasi-permanent filament was estimated to export between 30 % and 60 % of the average annual primary production in its region of formation stretching offshore for at least 200 km (Santana-Falcón et al., 2016; García-Muñoz et al., 2005; Sangrà, 2015). South of Cape Ghir, numerous minor filaments with more variable origin are often found, among which the Cape Juby and Cape Bojador filaments stand out. These filaments interact, feed into and wrap around numerous coastally generated eddies that often reach the Canary Archipelago, forming

a filament-eddy coupled system (Barton et al., 2004; Garcìa-Muñoz et al., 2004; Rodríguez et al., 2004). These nearshore-generated eddies, together with the eddies shed by the Canary Archipelago through the destabilization of the flow of the Canary Current, form the so-called Canary Eddy Corridor, which has been demonstrated to strongly enhance primary production in the region due to the intense biological activity in the eddy cores (Barton et al., 1998; Arístegui et al., 1997). Eddies in the Canary Eddy Corridor may live for more than a year, propagate far westward and thus drastically enhance the offshore reach

of the coastally produced matter, with an estimated annual mean integrated transport of 1.3 Sv (Sangrà et al., 2009, 2007).

Bounding the northern and the southern CanUS subregions and originating in the region of formation of the Cape Verde front (21 °N), the giant Cape Blanc filament is the most intense upwelling filament of the system. In fact, it is one of the largest filaments among all EBUS, extending more than 700 km in the open waters in the winter season (Ohde et al., 2015). This structure has been reported to export chlorophyll for about 400 km from the coast, and sinking POC up to 600 km

offshore at intermediate depths of 400 m to 800 m (Fischer et al., 2009), accounting for a total lateral export of about 50 %

of the newly produced particulate matter on the shelf (Gabric et al., 1993). The whole system of filaments that form along the northern CanUS sector from Cape Blanc (21 °N) and Cape Beddouza (33 °N) has been estimated to account for a total offshore transport of about 6 to 9 Sv (Barton et al., 2004), responsible for between 2.5 and 4.5 times the offshore $C_{org}$ export carried by the Ekman transport (Álvarez-Salgado, 2007).

Fewer studies focused on the upwelling and mesoscale dynamics of the region south of the Cape Verde frontal zone. Here, filaments have a more transient nature compared to those generated in the northern CanUS sector (Arístegui et al., 2009). Most of the filaments in the southern sector are found between Cape Verde and Cape Blanc, they extend between 100 km to 200 km offshore and have a typical lifetime of a few weeks (Menna et al., 2016). Eddies in the southern CanUS tend to be generated mainly near the coast near some topographic hotspots, and to move along distinct eddy corridors, having a major role in the

westward transport of physical properties, with significant differences between cyclonic and anticyclonic eddies (Schütte et al., 2016a).

     Given the complexity and the intermittency of the mesoscale dynamics, a quantification of the integrated mesoscale transport and of the relative contribution of eddies and filaments in upwelling regions is extremely challenging to achieve with observations alone. In response, most studies have employed results from model simulations. A comparative study of the CanUS and

15 California Upwelling System (CalUS) addressed the role of the cross-shore eddy diffusivity in the redistribution of physical and biogeochemical properties, finding that systems with high eddy activity, such as the CalUS, are characterized by a much more dispersive environment and are therefore more likely to see also their coastal tracers concentrations eroded by the lateral eddy mixing (Marchesiello and Estrade, 2006). Strengthening this claim, Nagai et al. (2015) demonstrated that mesoscale structures in the CalUS were largely driving the offshore transport of $C_{org}$ in this system. Furthermore, model simulations for

both the CalUS and CanUS also showed that eddies tend to reduce coastal production through the lateral export and subduction of the upwelled nutrients, leading to a lower nearshore nutrient inventory (Gruber et al., 2011; Lachkar and Gruber, 2011). Focusing on the CalUS and with the use of a passive tracer, Combes et al. (2013) have demonstrated that mesoscale eddies, and in particular cyclonic ones, exert a strong control on the horizontal offshore transport. Combining a simple ecosystem model with modeled and observed velocity fields, the mesoscale transport in the Benguela Upwelling System was estimated to

account for 30 %- 50 % of the total offshore flux of biogeochemical tracers (Hernández-Carrasco et al., 2014).

     Despite these previous efforts, the long-range integrated mesoscale transport in the CanUS and its contribution to the total transport of $C_{org}$ (Lovecchio et al., 2017) remains ill quantified. Here we aim to fill this gap using a fully coupled physical and biogeochemical model that we employed earlier to study the total offshore flux in this system. The model is configured on a full Atlantic basin grid with an eddy resolving resolution in the region of study that allows us to study the fluxes up to 2000

30 km offshore. Performing a Reynolds decomposition we present a quantification of the turbulent lateral and vertical transport of organic carbon as a whole; then, using a filament and eddy identification algorithms, we study the specific impact of filaments, cyclonic and anticyclonic eddies on the $C_{org}$ budget and transport.

## 2 Methods

We employ the same simulation results as used by Lovecchio et al. (2017) to study the time mean total offshore transport of $C_{org}$ in the CanUS. The model consists of the UCLA-ETH version of the Regional Ocean Modeling System (ROMS) (Shchepetkin and McWilliams, 2005) coupled to a Nutrient, Phytoplankton, Zooplankton, and Detritus (NPZD) biogeochemical ecosystem model (Gruber et al., 2006). The employed Atlantic telescopic grid is curvilinear, covers the entire Atlantic, and has a strong grid refinement towards the north-western African coast (Figure 1). This setup allowed us to model the large-scale flow in the whole Atlantic basin, while maintaining an eddy resolving resolution of between 4.9 km and 20 km in the region of interest. The model was run for 53 years, of which the first 29 years are considered spinup and the last 24 years are used for the analyses. The output was saved in the form of bi-daily mean fields, a time resolution that allows us to identify and track the rapidly evolving mesoscale structures.

For our analyses, the $C_{org}$ pool is inferred from the explicitly modelled organic nitrogen pools through a constant stoichiometric ratio of C:N = 116:16. The following pools make up the total organic matter: a non-sinking zooplankton pool, a sinking phytoplankton pool, and two detritus pools, of which one is sinking fast, and one is sinking slowly. The biogeochemical model includes an explicit sediment layer at the bottom of each water column grid point. There, all deposited organic matter is remineralized slowly back to its inorganic constituents, which are then released back to overlying waters. Thus, the sediments act as a temporal buffer, but not as a net sink of any biogeochemical element. An important potential shortcoming is the lack of consideration of an explicit DOC pool. However, Lovecchio et al. (2017) demonstrated already that our slowly-sinking detritus pool acts akin to semi-labile DOC, so that our model captures important elements of this pool as well. We refer to Lovecchio et al. (2017) for further discussions of the strengths and limitations of our modelling approach.

Using a Reynolds decomposition approach ("turbulence-based" approach), we separate the advective organic carbon flux, $\boldsymbol{F} = \boldsymbol{u}\,C_{org}$, with $\boldsymbol{u}$ being the velocity vector, into a mean component ($\overline{\boldsymbol{F}}$) and into a time-varying ($\boldsymbol{F}'$), i.e., turbulent component. The overbar denotes a time averaging operator, here chosen such that the reference mean of each field varies smoothly following the seasonal climatology. The prime indicates the temporal deviations from this time-varying mean. The two components are computed by first calculating $\overline{\boldsymbol{u}}$ and $\overline{C}_{org}$. We then determine the temporal deviations by subtraction, i.e., $\boldsymbol{u}' = \overline{\boldsymbol{u}} - \boldsymbol{u}$ and $C'_{org} = \overline{C}_{org} - C_{org}$. This permits us to decompose the long-term mean advective flux of $C_{org}$ into three components, i.e.,

$$\langle \boldsymbol{F} \rangle \quad = \left\langle \overline{\boldsymbol{u}}\,\overline{C}_{org} \right\rangle + \left\langle \boldsymbol{u}'C'_{org} \right\rangle + r, \tag{1}$$

where the angled brackets indicate the average over our analysis period. The first term on the right hand side represents the long-term average of the (time-varying) mean flux, i.e., $\left\langle \overline{\boldsymbol{F}} \right\rangle$. The second term represents the long-term average turbulent contribution $\langle \boldsymbol{F}' \rangle$ to the overall flux. The last term, $r = \left\langle \overline{\boldsymbol{u}}C'_{org} \right\rangle + \left\langle \boldsymbol{u}'\overline{C}_{org} \right\rangle$, represents the sum of the residuals, which arises from our use of a time-varying reference mean. We neglect this residual as it is at least one order of magnitude or more smaller than the other terms. The time-varying reference means for the decomposition are computed from the climatological monthly means of velocities and concentrations calculated from the 24 years of the run used for the analysis, and then interpolated linearly to bi-daily fields. This choice allows us to avoid the inclusion of both the dominant seasonal variability (Chavez and Messié, 2009) and the recurrent monthly oscillations of the fields into the turbulent components. As a consequence, our

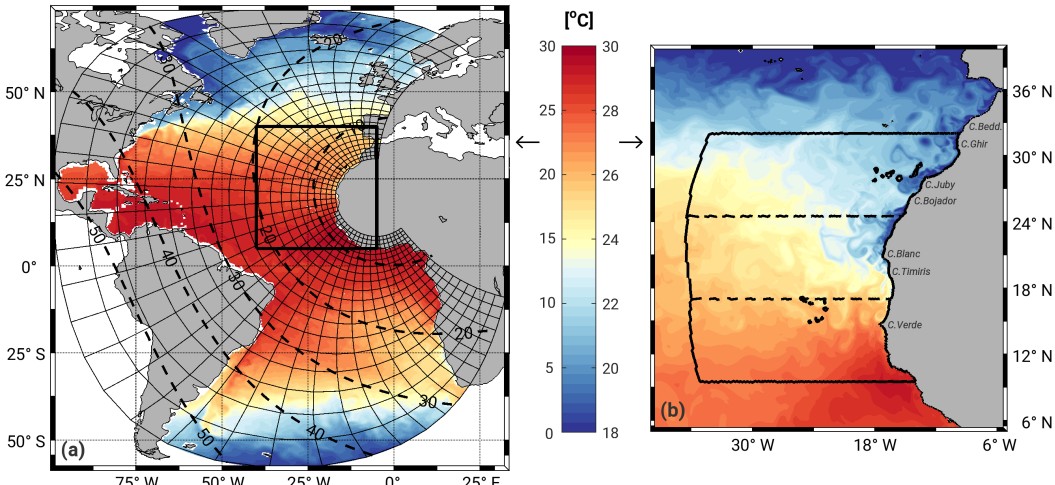

**Figure 1.** Model grid and CanUS domain superimposed on an instantaneous snapshot of Sea Surface Temperature [$^o$C] from Dec. 1 of year 30 of the run (01/12/0030). (a) Atlantic telescopic grid showing every 20th grid line (solid lines) with isolines of grid resolution in km (dashed lines); the black square highlights the CanUS region used for the zoomed-in plot. (b) Zoom on the CanUS region including the names of the major capes and the boundaries of the regional and subregional domains used for the analyses. The western boundary (solid line) is located at 2000 km offshore distance, the northernmost and southernmost regional boundaries (solid lines) are set at 9.5°N and 32°N, respectively. The subregional boundaries (dashed lines) are located at 17°N and 24.5°N and divide the CanUS region into 3 subregions: southern, central and northern CanUS.

turbulent fields represent only those signals that vary on timescales shorter than approximately one month. The vertical flux components were calculated using the purely vertical velocities from the model output.

To quantify the contribution of mesoscale eddies to the $C_{org}$ budget and transport, we use a sea surface height (SSH) based eddy identification algorithm (Faghmous et al., 2015). According to this method, eddies are defined as the outermost closed SSH contour containing a single extreme (either maximum or minimum). The algorithm was shown by Faghmous et al. (2015) to be able to retrieve about 96 % of the eddies identified by domain experts. Our visual analysis of animations of the eddy contours and centers on the modeled SSH and horizontal velocity fields confirmed the good performance of the method. Using this eddy identification method, we save at each time step the position of all the eddy centers, the grid points covered by their areas and their signatures (cyclonic or anticyclonic). Eddies with a radius smaller than 15 km are filtered out to avoid noise in the nearshore high-resolution region.

To estimate the contribution of filaments to the mesoscale transport, we developed a new SST-based upwelling filament-identification algorithm. This algorithm builds on several key characteristics of filaments, i.e., that they are generated near the coast, that they extend in a continuous manner offshore, and that they are distinctly colder than their environment, owing to them carrying recently upwelled water within them. A detailed description of the filament identification algorithm, including

a discussion of its sensitivity to the parameters and an analysis of its performance in the system of interest is provided in the supplementary material. An example of one of the filament masks is given in the Appendix in Figure B1.

We allocate all areas not covered by filaments and eddies to the non-filament/non-eddy (NF-NE) mask. As a consequence, this mask inherits the uncertainties and biases of the filament and eddy detections. For example, the filament detection method likely attributes a too large fraction of the nearshore 50 km to filaments, thus underestimating the contribution of the NF-NE transport in this very nearshore band. Unfortunately, we cannot easily correct for this likely bias, given the lack of a clear definition of the boundaries of a filament on the shelf. As a consequence, we disregard the results from the structure identification algorithms in the first 50 km.

In the vertical direction, we assume both eddies and filaments to have a prismatic volume, i.e., at each depth, $k$, they occupy the same horizontal $i, j$ positions as those identified at the surface. We use these 3D masks to decompose the $C_{org}$ concentration and the $C_{org}$ fluxes into their filament, cyclonic, anticyclonic components ("structure-based" approach). To calculate the $C_{org}$ concentration or concentration anomalies in the mesoscale structures, we multiplied these variables by the mask of interest at each time step and averaged the whole time series. The same procedure was followed for the computation of the fluxes. To calculate the mesoscale and NF-NE components of the $C_{org}$ fluxes, we multiply $C_{org}$ concentration and velocity fields by the mask of interest at each time step, multiply the obtained fields by each other and finally average in time the obtained fluxes.

We focus our analysis on the top 100 m, corresponding roughly to the average depth of the euphotic layer in the CanUS domain. This is largely motivated by our interest in the impact of the lateral redistribution of $C_{org}$ in the biologically most active region of the ocean. Furthermore, this depth layer is responsible for the majority of the offshore transport, with a share of nearly 100 % of the total flux at 100 km offshore and ∼80 % at 500 km offshore, where offshore transport is the most intense (Lovecchio et al., 2017).

## 3  Evaluation

The extensive evaluations performed already by Lovecchio et al. (2017) revealed that the physical variables such as sea surface height (SSH), sea surface temperature (SST), sea surface salinity (SSS) and mixed layer depth (MLD) are well represented by our model in the region of study. Relevant long-term mean biases in SST (∼ +0.75 °C) and SSS (∼ +0.4) are found only in the region south of Cape Blanc (21°N), where the modeled circulation is too sluggish compared to observations. Biological variables such as net primary production (NPP), chlorophyll (CHL) and POC are also well represented north of Cape Blanc and in its proximity, while south of this Cape they show too deep maxima. Correlations with the observed fields are always higher than 0.85 in the annual mean for physical variables and lay in the range of 0.6 to 0.88 for the relevant biological variables (see Appendix: Figure B2)

Extending these evaluations to include aspects that are particular to this study, namely the model's representation of turbulence and mesoscale processes, reveals also a good performance and a few weaknesses. The magnitudes of the Turbulent Kinetic Energy (TKE) and of the standard deviation of the sea-surface height (STD(SSH)) as observed by AVISO (see Appendix, Table A1) are overall well captured by the model (Figure 2). This is especially the case in the central sector of the

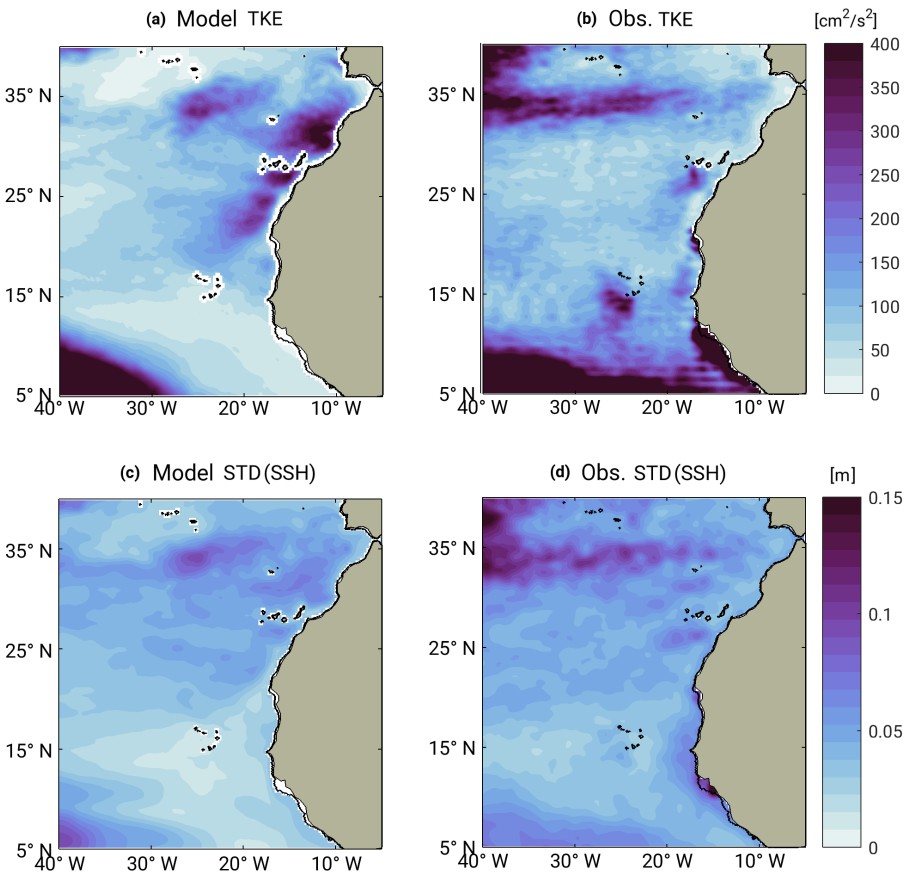

**Figure 2.** Comparison of (a) modeled ROMS and (b) observed AVISO Turbulent Kinetic Energy, TKE [cm$^2$/s$^2$]; Comparison of (c) modeled ROMS and (d) observed AVISO standard deviation of SSH, STD(SSH) [m]. Modeled velocities were calculated offline using geostrophy from modeled SSH and regridded on the AVISO DUACS 2014 product grid, 1/4 °resolution. TKE was calculated from bi-daily (ROMS) and daily (AVISO) turbulent velocities, defined as differences of the total fields from the monthly climatology interpolated to daily (AVISO) or bi-daily (ROMS) time resolution. A detailed description of the data used is provided in Appendix A: Datasets, Table A1.

CanUS, corresponding to the region between the Canary and the Cape Verde archipelago. However, as expected from our assessment of the mean circulation, the two quantities are underestimated south of Cape Blanc, especially in the proximity of the North Equatorial Counter Current and south of the Cape Verde archipelago. Similarly, at the north-western boundary of the CanUS, the TKE associated with the incoming Azores Current is smaller than observed. Both the Azores Current and the Northern Equatorial Counter Current are, in fact, generated away from the north-western African coast, i.e., in regions where our grid has a low resolution, hindering the formation of mesoscale variability. Along the coast of the northern CanUS, and especially north of the Canary Archipelago, the modeled TKE and STD(SSH) are actually higher than observed. This region is highly populated by coastal upwelling filaments, which may be only partially resolved by the employed satellite product. Thus, it is conceivable that the TKE and STD(SSH) products are biased low in this region.

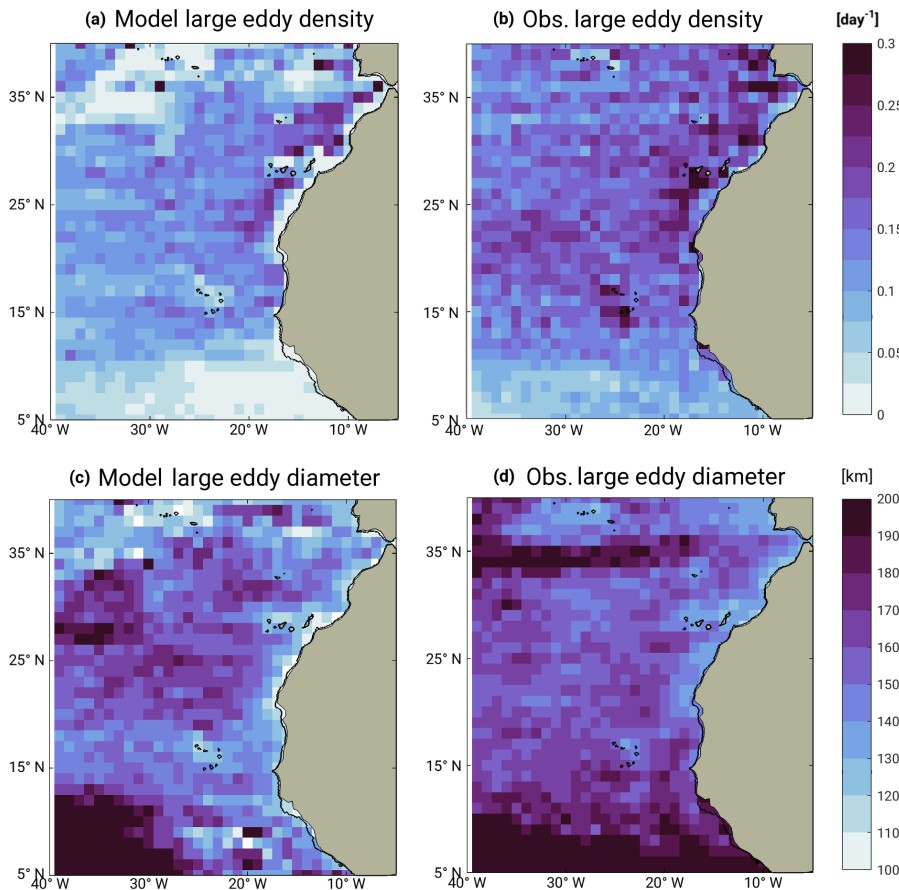

**Figure 3.** Comparison of the mean number of large eddy (R≥50 km) centroids from (a) modeled ROMS and (b) observed AVISO, binned to 1°x 1°bins. Comparison of the mean diameter [km] of large eddies (R≥50km) from (c) modeled ROMS and (d) observed AVISO, binned to 1°x 1°bins.

To evaluate the eddy field simulated by our model, we ran the eddy-finding algorithm on both modeled and AVISO SSH on their respective native grids. Given the AVISO grid resolution and the minimum threshold of 9 grid points to identify an eddy, the distribution of eddy radii (R) from AVISO drops for small R values. We therefore limit the comparison to eddies with a radius R of more than 50 km. The density of these "big" eddy centroids (number of eddies per 1 degree bin) from model and observations show a similar spatial pattern, even though the modeled density of big eddies is slightly biased low (Figure 3a,b). In line with our analyses of TKE and STD(SSH), the modeled density of big eddy centroids shows the closest match to the observations in the central region of the CanUS, while the main differences are found at the northern and southern edges of the CanUS domain (in correspondence of the Azores Current and North Equatorial Counter Current), south-west of the Cape Verde archipelago, and around the Canary Islands. This goes along with our model struggling in these regions to reproduce the very big eddies with ⟨D⟩ > 300km (Figure 3c,d). At the northern and southern boundaries, this deficiency could again be a

consequence of the eddies originating from outside our main region of interest, i.e., from regions of low grid resolution. The lack of eddies southwest of Cape Verde archipelago may be due to a northern shift of the modeled flow of the Cape Verde front, which hits the islands with a lower intensity, thereby generating fewer instabilities, and hence eddies, than observed. Also, in the proximity of both the Cape Verde archipelago and of the Canary Islands, the presence of land points (absent in the AVISO product) may hinder the identification of large eddies. In the Canary Islands region, in fact, the high TKE in the absence of large eddies suggests a high abundance of detected small eddies, as confirmed by our plot of the density of the entire eddy population (R > 15 km) used for the analysis (see Appendix: Figure B4).

Anticyclonic eddies (ACE) and cyclonic eddies (CE) in the CanUS differ marginally in terms of their statistical properties. ACE are slightly less abundant than CE throughout the CanUS, representing about 49% of the total eddy population, in agreement with previous analyses (Schütte et al., 2016a). ACE and CE have a mean diameter of respectively $100 \pm 59$ km and $90 \pm 51$ km with the distributions of the diameters peaking at around 45 km in both cases (see Appendix: Figure B3). The two types of eddies occupy on average, respectively, 15% and 10% of the entire surface of the CanUS, summing to $\sim 25$ % of the total area, in line with results of previous studies (Chaigneau et al., 2009). Both kinds of eddies reach a maximum area occupation at around 250 km offshore, i.e., in the range where they are likely generated (see Appendix: Figure B5). With increasing distance, the share of area occupied by CE remains pretty stable, while the share occupied by ACE shows a slow but persistent decline. The latter may be an indication of the reduced stability and shorter life span of ACE, in line with the slight prevalence of CE over ACE detected in satellite-based studies for propagation distances of 2500 km (Chelton et al., 2011), with important consequences on their role for offshore transport. A comparison of these trends with those derived from the AVISO daily eddy field shows overall an excellent level of agreement. Our model slightly underestimates the total surface occupied by CE (by 2.5 %) and by ACE (by 5 %), which results in a slight bias in favor of CE. However, the offshore evolution of the surface occupation of the two types of eddies compares very favorably with the observed trends, giving us a substantial amount of confidence in terms of our analyses of eddy evolution.

## 4 Results

### 4.1 Turbulence in the organic carbon field

At any moment, the $C_{org}$ pattern is shaped by the interactions between the biological and physical processes that add, remove, and redistribute the organic matter in the ocean. Due to the interplay of these processes, the concentration of $C_{org}$ along the north-western African coast exhibits a complex pattern that combines a large-scale offshore gradient with smaller-scale anomalies visible as swirls, squirts and fronts that correspond closely with the pattern of SSH and currents (Figure 4a,b). This pattern can be conceived as the superposition of a mean $C_{org}$ field and of a turbulent deviation around it (Figure 4c,d). The turbulent component of this pattern, characterized by strong positive and negative anomalies, clearly evidences the important role of mesoscale eddies and filaments. Thin and short-lived filaments channel the carbon away from the coast, while slower, but persistent eddies create islands of enhanced or reduced $C_{org}$ concentration that propagate toward the open ocean.

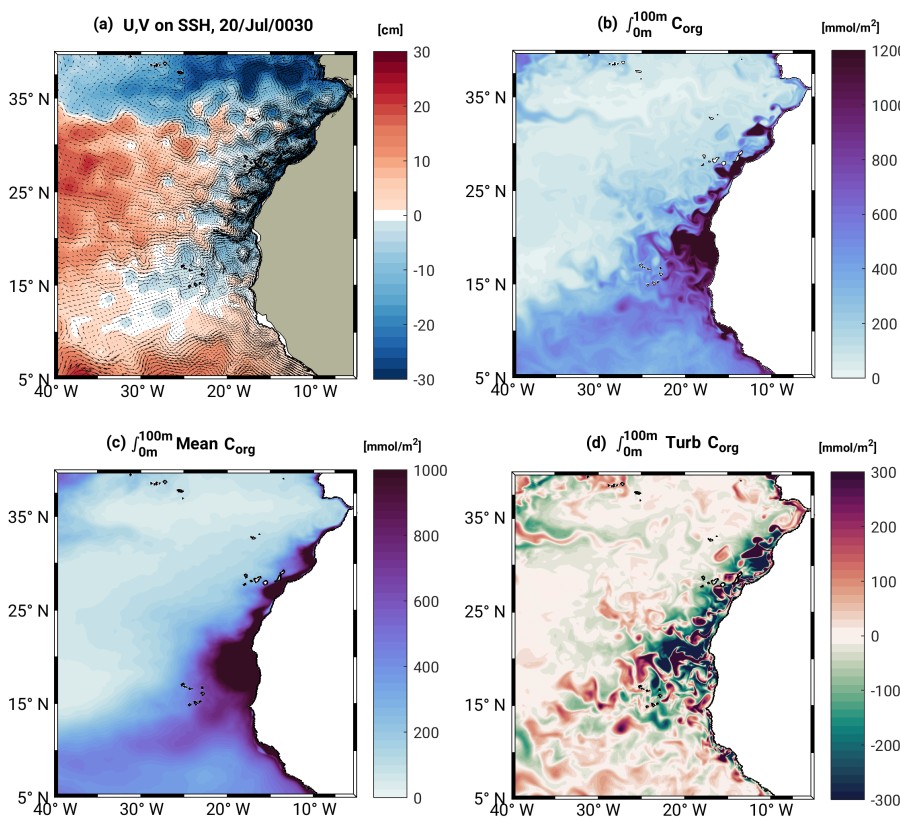

**Figure 4.** Modeled 2-day mean variables for July 19-20 of year 33 of the simulation. (a) SSH [cm] (color shading) and currents $\boldsymbol{u}$ (vectors); (b) Total organic carbon stock ($\int_0^{100m} C_{org}$) as well as its (c) mean ($\int_0^{100m} \overline{C}_{org}$) and turbulent part ($\int_0^{100m} C'_{org}$), respectively. The $C_{org}$ is integrated throughout the first 100 m of depth.

In the following, we employ two complementary approaches to quantify the relative role of these mesoscale variations to the total offshore transport of $C_{org}$. In the first "turbulence-based" approach, we use a Reynolds decomposition to separate these two components, while in the second "structure-based" approach, we use filament and eddy masks to quantify the specific contribution of these two kinds of mesoscale structures to the $C_{org}$ transport.

## 4.2 Mean and turbulent transport

The time-mean fluxes from the Reynolds decomposition (Figure 5a-c) clearly reflect the regional circulation and the wind stress curl pattern that characterizes the CanUS (Arístegui et al., 2009; Pelegrí and Peña-Izquierdo, 2015; Lovecchio et al., 2017). While these mean fluxes dominate the total fluxes (see also: Lovecchio et al. (2017), their Figure 11), the turbulent fluxes contribute substantially to the total fluxes and to their divergence (Figure 5d-f). In the zonal direction, the turbulent transport strengthens the offshore transport of $C_{org}$ at every latitude, with the negative signature extending far into the open

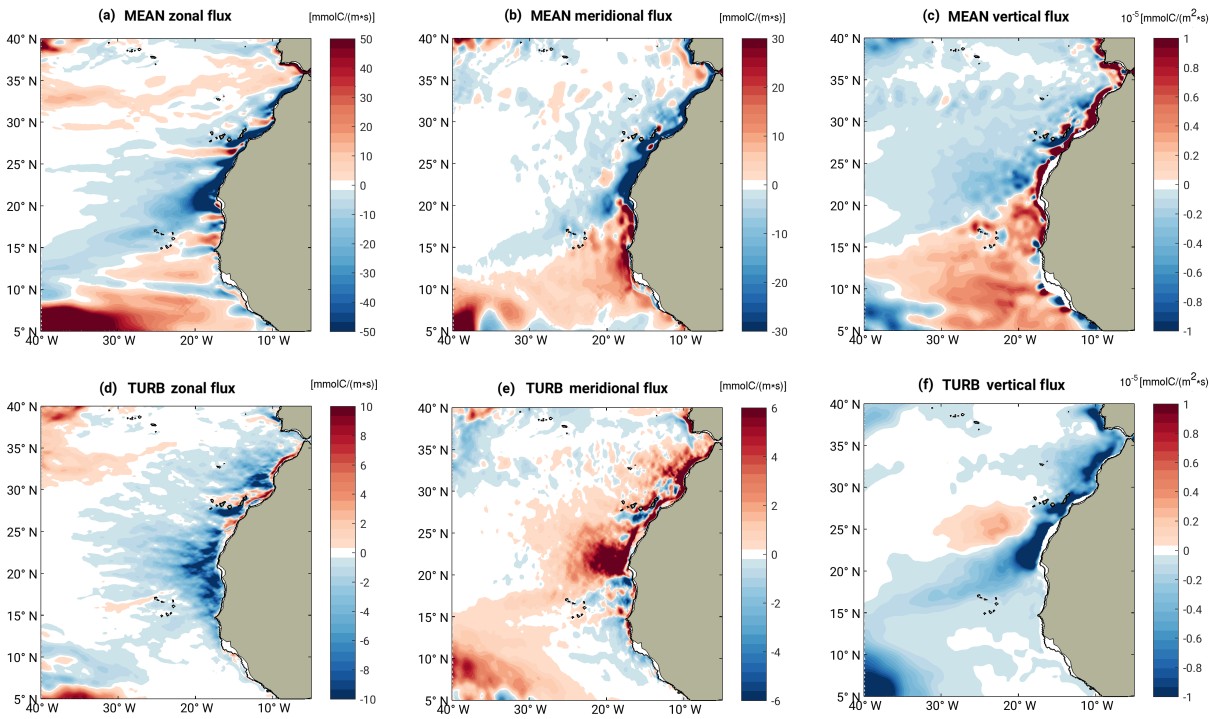

**Figure 5.** Reynolds decomposition of the lateral and vertical advective fluxes of $C_{org}$ into their average mean $\langle \overline{\boldsymbol{u}}\,\overline{C}_{org} \rangle$ and average turbulent $\langle \boldsymbol{u}'C'_{org} \rangle$ components, as defined in the methods section. Lateral fluxes are integrated in the first 100m depth, the vertical flux is sliced at 100m depth.

waters (Figure 5d). The magnitude of this turbulent transport varies within the CanUS from a minimum of 5 % to up to above 30 % of the mean lateral transport (Figure 6a,b). The maximum relative contribution is reached at about 200 km from the coast and slowly declines further offshore. The turbulent component contributes also about a third to the total zonal divergence (Figure 6a, insert), i.e., the amount of $C_{org}$ released by the zonal flux on the way to the open ocean. This divergence significantly enhances the $C_{org}$ stock in the offshore waters.

The contribution of the turbulent offshore flux is particularly important in the northern and southern CanUS. In the northern subregion (Figure 6c), the mean flux declines much faster in offshore direction than the turbulent flux, such that the latter is responsible for more than half of the total transport in the region beyond 200 km. However, in terms of divergence, the turbulent contribution becomes important only beyond 500 km and then really dominant beyond 1500 km. In the southern subregion, the mean flux is, on average, directed onshore in the first 1000 km, while the turbulent flux opposes it, redirecting part of the $C_{org}$ towards the open waters (Figure 6e). In terms of divergence, the turbulent flux opposes the mean fluxes, adding $C_{org}$ with maximum rates of about 1/3 of the absolute value of the mean divergence between 100 km and 500 km offshore. In contrast, the central CanUS (Figure 6d) is characterized by a very intense mean offshore flux, connected to the currents that create the

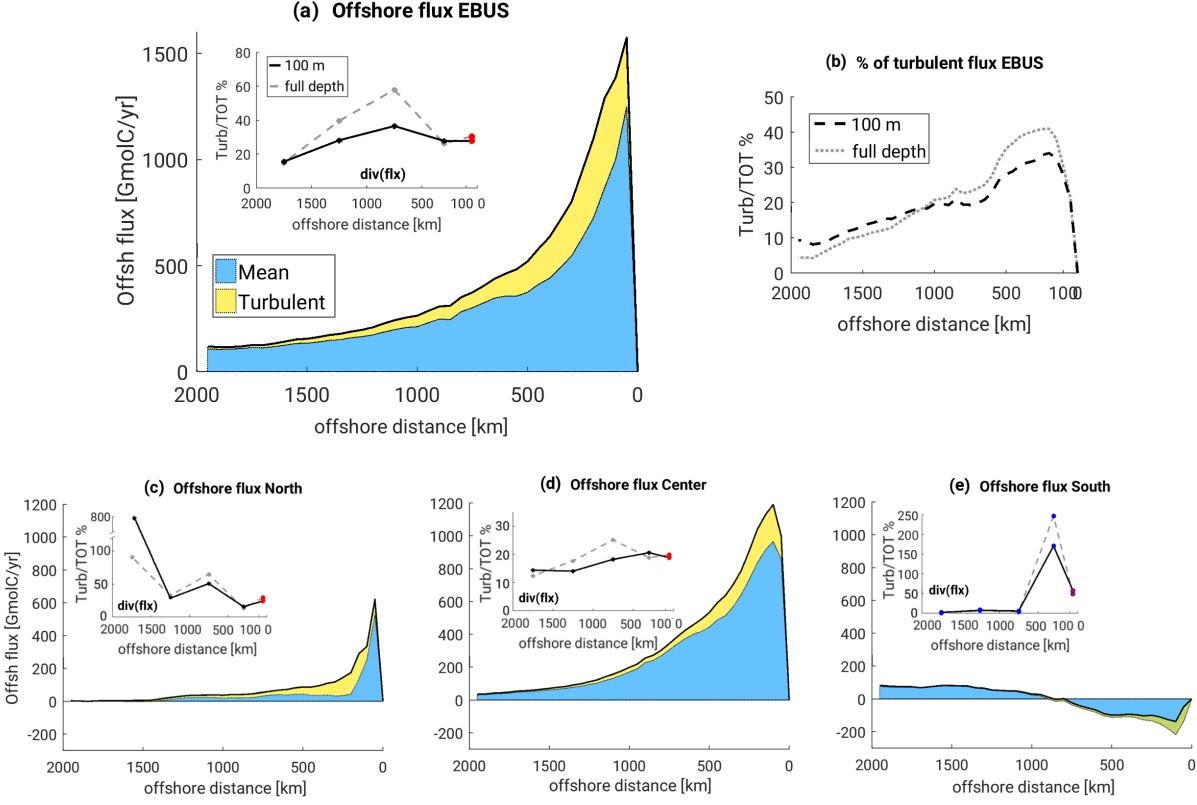

**Figure 6.** Magnitude of the mean and turbulent components of the offshore flux [GmolC yr$^{-1}$]: (a) Canary EBUS as a whole; (c) northern CanUS; (d) central CanUS; (e) southern CanUS. In all area plots: the black thick line is the total offshore flux, sum of mean and turbulent fluxes; fluxes are integrated in the first 100 m depth. Plot (b): only for the full EBUS we show the ratio of the magnitude of the turbulent/total offshore flux, integrated in the first 100 m and throughout the whole watercolumn. Inserts: ratio between the absolute value of the divergence of the turbulent component and that of the total flux, integrated in the first 100 m depth (black line) and over the full water-column (gray line). Per each offshore range, divergences are calculated as finite differences at the boundaries shown on the x-axis. Dots are red when both total and turbulent divergences are negative (fluxes remove C$_{org}$); dots have the same color of the line when both divergences are positive (fluxes add C$_{org}$). In the Southern subregion only: blue dots indicate that the divergence of the turbulent flux (positive) opposes the divergence of the mean flux (negative); the opposite is true for red dots with a blue contour. The green area shading results from the overlapping of opposing mean and turbulent flux contributions.

Cape Verde front. However, even in this region, the turbulent flux contributes up to 25% and always more than 5 % to the total offshore flux of C$_{org}$. The contribution of the turbulent offshore flux to the divergence is of similar magnitude.

The turbulent transport has an important role also in the meridional direction, as it opposes the mean flow and recirculates C$_{org}$ against the direction of the mean currents (Figure 5b,e). This is especially the case along the coast, where the turbulent

component reaches a magnitude of about 20% of the mean flux. The strongest turbulent meridional transports are associated with the Canary and Mauritanian currents.

The most important contribution of the turbulent transport occurs in the vertical with the vertical component of the turbulent flux exceeding the magnitude of the mean fluxes (Figure 5c,f). This occurs especially in the nearshore northern and central CanUS, where the turbulent vertical component at a depth of 100 m is strongly downwelling, opposing the mean upwelling at the coast. As a result, the coastally-produced $C_{org}$ gets subducted below the euphotic layer and potentially exported further offshore towards the center of the North Atlantic gyre.

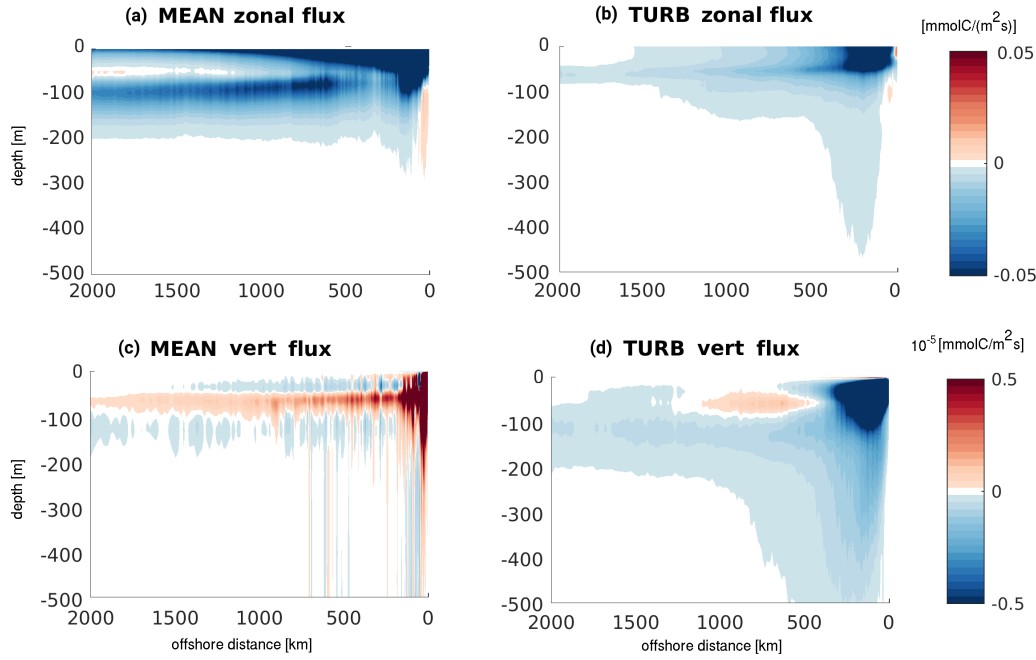

**Figure 7.** Reynolds mean and turbulent components of the advective fluxes of $C_{org}$ averaged along lines of equal distance from the coast in the whole CanUS. (a) Mean zonal advective flux; (b) Turbulent zonal advective flux; (c) Mean vertical advective flux; (d) Turbulent vertical advective flux.

Analyzing the offshore transport as a function of depth reveals that once averaged along the coast, the $C_{org}$ is transported offshore at every depth, with the majority of this transport occurring in the top 100 m and almost none below 200 m (Figure 7a), . The turbulent component of this zonal offshore transport is characterized by a thin and shallow maximum confined to the top 100 m, with only little transport occurring below (Figure 7a). In contrast, the mean zonal flux tends to extend deeper, especially further offshore. Both flux components show an offshore deepening of the transport, likely as a consequence of the aforementioned vertical subduction, but also because of the deepening of production (c.f., Lovecchio et al. (2017), Figure B6). The strong near surface confinement of the offshore transport leads to a very low sensitivity of the results to the choice of integration depth, i.e., the regional pattern does not change substantially when the fluxes are integrated over the first 100 m

depth (Figure 5a,d) or from 100 m to the bottom of the water column (see Appendix: Figure B6). An exception is the nearshore, where the mean transport is directed onshore between about 100 and 200 m, reflecting the presence of the upwelling cell.

Compared to the vertical distribution of the zonal fluxes, the vertical fluxes show more pronounced differences between the mean and turbulent components (Figure 7c,d). Near the coast, the mean vertical transport is dominated, as expected, by the signature of the coastal upwelling. Offshore, the different sign of the wind stress curl (negative and positive, respectively) characterizing the latitudes located north and south of the Cape Verde front, results in an opposite sign of the time-mean vertical flux of organic carbon in the two zonal bands (see Figure 5c). These fluxes, averaged along the entire CanUS, give rise to an alternate pattern of upward and downward mean vertical transport. On the contrary, the turbulent component of the vertical flux of $C_{org}$ is directed downward in all the subregions, and particularly intense in the nearshore. However, in opposition to the zonal flux, the turbulent vertical transport extends much deeper than the mean transport, reaching 200 m depth everywhere and more than 500 m within the first 500 km from the coast.

### 4.3 From turbulent anomalies to the mesoscale contribution to the organic carbon stock

The important contribution of the offshore transport by turbulent anomalies can be visualized through the use of Hovmoeller diagrams, that show how positive and negative anomalies of SSH, i.e. SSH$'$, and $C_{org}'$, i.e. $C_{org}'$, are moving offshore (See Appendix, figure B7).These signals propagate coherently from the coastline through the whole 2000 km offshore range in about 1.5 years. Their mean propagation speed of a few cm s$^{-1}$ corresponds to that of first baroclinic mode Rossby waves at these latitudes (Klocker and Abernathey, 2014), suggesting an important role of coherent mesoscale eddies in the turbulent offshore transport. Analogous Hovmoeller diagrams can be plotted using the eddy and filament associated $C_{org}'$, isolated with

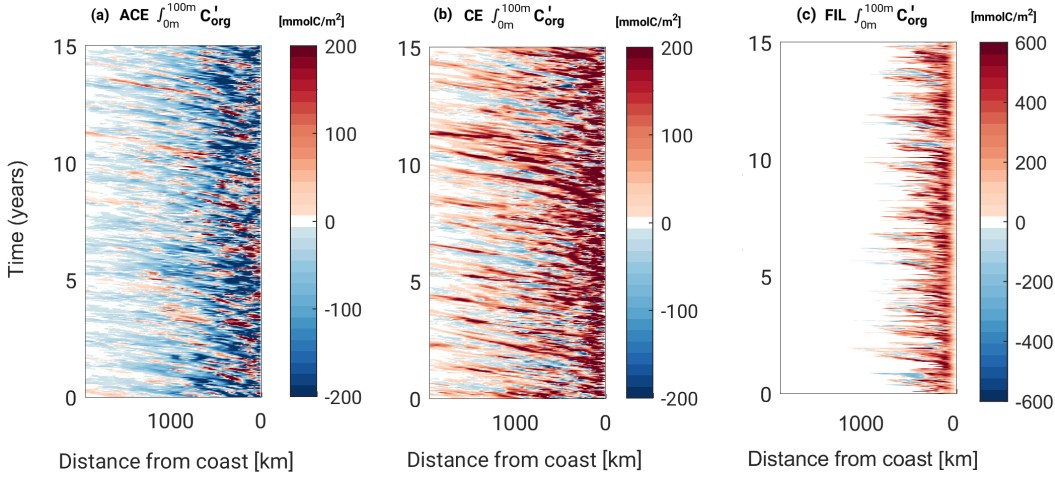

**Figure 8.** Hovmoeller diagrams of the top 100 m stock anomalies of $C_{org}$ in the central CanUS subregion for (a) anticyclones (ACE), (b) cyclones (CE), (c) filaments (FIL). First 15 years of analysis data.

the use of the structure-identification masks (Figure 8). These diagrams show that the propagation speed of the eddy anomalies

is close to that of the above-mentioned turbulent anomalies, and highlight the opposing signature in the $C'_{org}$ associated with ACE and CE. Filament anomalies, instead, show a different character: Both their magnitude and their advection speed generally exceeds by several times that of the eddy anomalies. Typical zonal velocities in the top 100 m of a filament are about 0.15 m s$^{-1}$, while the corresponding velocities for eddies are generally less than 0.05 m s$^{-1}$, with no significant difference between ACE and CE. But even though the filament transport is fast, it stays largely confined to the nearshore 500 km.

The mean correlation and cross-products between SSH′ and $C'_{org}$ provide an additional link between the propagation of the turbulent anomalies and the mesoscale contribution to the long-range transport. These anomalies (Figure 9 a,b) are anti-correlated in most of our analysis domain (Figure 9 a,b), i.e., on average, $C_{org}$ is enhanced when SSH′<0 (typically associated to intense CE and filaments) and dampened when SSH′>0 (associated to ACE). Exceptions are found in the surroundings of the incoming Azores and North Equatorial Counter Currents. The cross-product of SSH′ and $C'_{org}$ is particularly negative in

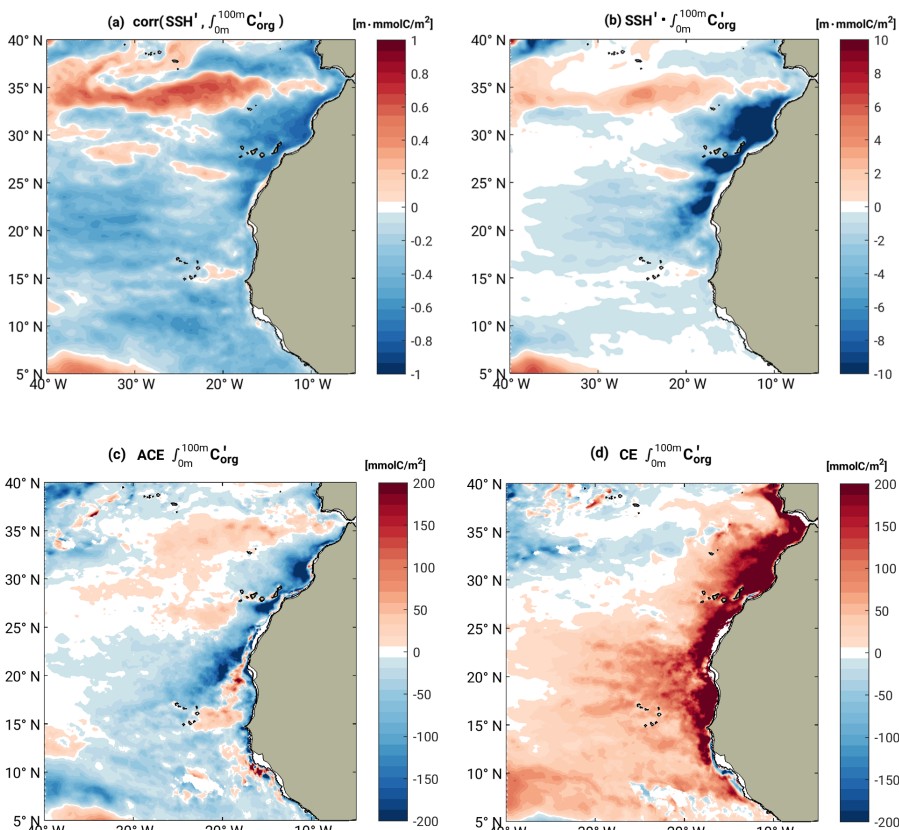

**Figure 9.** (a) Correlation and (b) covariance of the turbulent SSH anomalies (SSH′) with the top 100 m organic carbon stock anomalies, i.e. $\int_0^{100m} C'_{org}$. Mean $\int_0^{100m} C'_{org}$ stock contained in (c) ACE and (b) CE, respectively.

the nearshore northern and central CanUS subregions (Figure 9b), where we find the most intense and recurrent upwelling filaments, by definition characterized by strongly negative SSH′ (Cravo et al., 2010).

The overall negative correlation between SSH′ and $C'_{org}$ stems indeed from ACE having, on average, a lower $C_{org}$ content, and CE a higher one (Figure 9 c,d). Many potential mechanisms can contribute to this commonly made observation. These include a local vertical displacement of the nutricline at the early stages of eddy intensification (McGillicuddy, 2016) and the trapping of the tracer anomalies at formation in the proximity of the southward flowing Canary Current, which favors the

5  formation of high $C_{org}$ CE and low $C_{org}$ ACE (Gaube et al., 2014).

Deviations from the negative correlation of SSH′ and $C'_{org}$ can also be explained in terms of eddy anomalies. In the nearshore southern CanUS, the positive $C'_{org}$ in ACE might stem by the shedding of eddies from the northward flowing Mauritanian Current, which favors the trapping of $C_{org}$-rich coastal waters in ACE at the time of their formation. At the northern CanUS boundary, the signature of the eddy $C'_{org}$ is connected to the presence of incoming large eddies formed in the Azores current

10  (Gaube et al., 2014, Figure 1), leading to ACE with positive $C'_{org}$ and CE with negative $C'_{org}$. Also in the offshore waters of the northern CanUS, ACE have a positive $C'_{org}$. This is possibly due to the sharp offshore gradient in $C_{org}$ at these latitudes, where coastally generated ACE can carry the elevated $C_{org}$ to the offshore waters even though they are generally not very efficient in doing so.

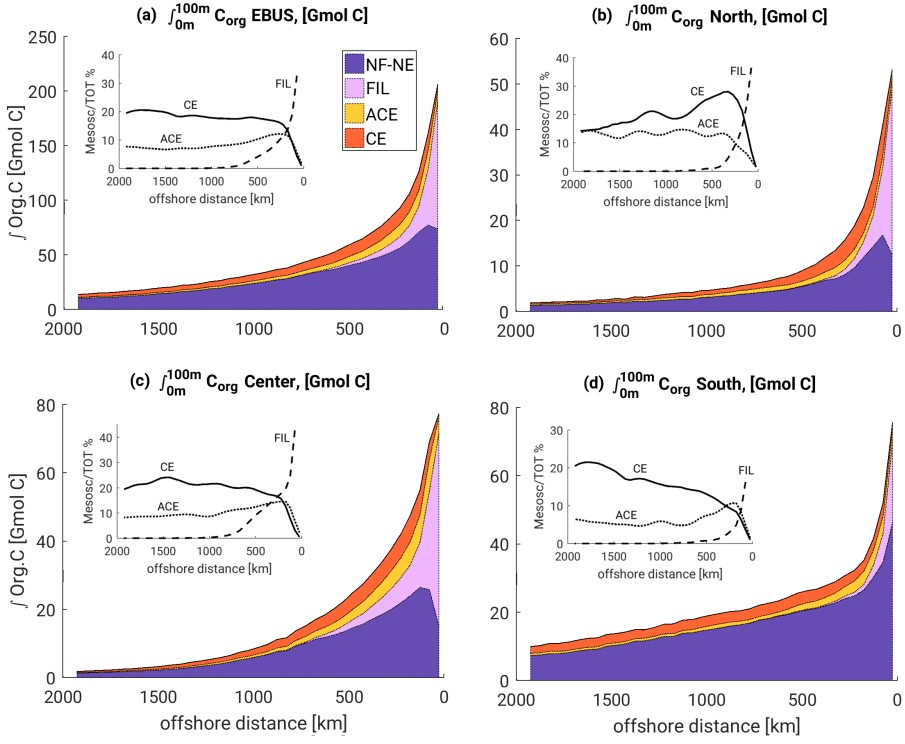

**Figure 10.** Top 100 m organic carbon stock ($\int_0^{100m} C_{org}$) contained in filaments (FIL), cyclonic eddies (CE), anticyclonic eddies (ACE) and outside of the detected mesoscale structures (NF-NE) by offshore distance. The $\int_0^{100m} C_{org}$ is integrated across the horizontal extension of: (a) the entire Canary EBUS; (b) the northern CanUS; (c) the central CanUS; (d) the southern CanUS. Inserts: Percentage of the total organic carbon stock found within ACE, CE and FIL, by offshore distance; the FIL share of $C_{org}$ in the first 50 km offshore is not shown.

Integrated across the top 100 m, the fraction of the $C_{org}$ stock that is contained inside mesoscale structures differs drastically between filaments and eddies (Figure 10). On average, filaments are the mesoscale structure that contains most of the $C_{org}$ in the first 200 km offshore, representing between 20 % and 40 % of the total $C_{org}$ share. Their contribution declines nearly exponentially with distance, reaching nearly zero at about 700 km offshore. This offshore decrease is closely connected to the

5 large aspect ratio of filaments that extend offshore with a particularly narrow stream of typical widths of a few tens of km, and occupy therefore a small portion of the CanUS area. On the contrary, ACE and CE see a sharp increase of their $C_{org}$ content in the first few hundreds km from the coast, where most of them are generated (see Appendix: Figure B5). At offshore distances larger than 200 km offshore, ACE and CE together contain, on average, about 30% of the total $C_{org}$ in the top 100 m, most of which is found in CE. The relative share of CE is around 20 % at every distance beyond 200 km from the coast, with a weakly

increasing share with increasing offshore distance. This constitutes more than the portion of area occupied on average by CE (see Appendix: Figure B5), therefore indicating that CE constitute an important carbon reservoir for the open waters. The opposite trend is observed in ACE that have the highest $C_{org}$ share at about 200 km offshore, while at 1500 km distance, they represent only around 10% of the total. This is due both to the faster decrease in the number of ACE offshore and to a decline in their $C_{org}$ content. Moreover, the two types of eddies are characterized by strong differences in their nutrient concentrations

and in their production rates (see also Appendix: and Figure B8). This likely exacerbating the differences in $C_{org}$ between CE and ACE, and especially for the long living ones.

### 4.4 Filament and eddy transport of organic carbon

The magnitude and pattern of the "structure-based" estimation of the different contributions (eddy, filament, and NF-NE) to the total offshore transport of $C_{org}$ shows similarities, but also important differences to the results from the Reynolds decomposi-

20 tion. The NF-NE contribution to the zonal transport (Figure 11a), is overall very similar to the pattern of the mean fluxes from the Reynolds decomposition. As was the case for the latter, the pattern of NF-NE component is primarily shaped by the mean surface currents. An exception is the intensification of the NF-NE flux in the range between 100 km and 500 km, resulting in a negative offshore flux divergence (Figure 13). This is primarily the result of the transfer of $C_{org}$ from the filament transport to the transport driven by NF-NE. Only in the nearshore upwelling band ($\sim$ 50 km offshore) this negative divergence could also

be a consequence of our likely underestimation of the NF-NE contribution (see methods).

The filament flux dominates the offshore export in the first few hundred kilometers, both in terms of absolute magnitude and its divergence (Figure 12 and Figure 13b,d). In particular, at an offshore distance of 100 km, where the $C_{org}$ offshore flux reaches its maximum magnitude, the filament flux accounts for nearly 80% of the total offshore transport. Beyond 500 km, its contribution declines rapidly and reaches zero at about 1000 km offshore (Figure 12a,b). Between 100 km and 500 km

from the shore, the horizontal (negative) divergence of the filament transport exceeds the total (negative) divergence of the offshore $C_{org}$ flux, representing more than 120 % of its value. The difference is caused by the positive divergence of the NF-NE transport. Even though the filament flux decreases rapidly with offshore distance, its divergence still account for $\sim$50% of the total divergence between 500 km and 1000 km offshore. Therefore, the filament transport constitutes overall the most important source of coastally-produced $C_{org}$ in the first 1000 km, a consequence of their high $C_{org}$ concentrations as well as of the high

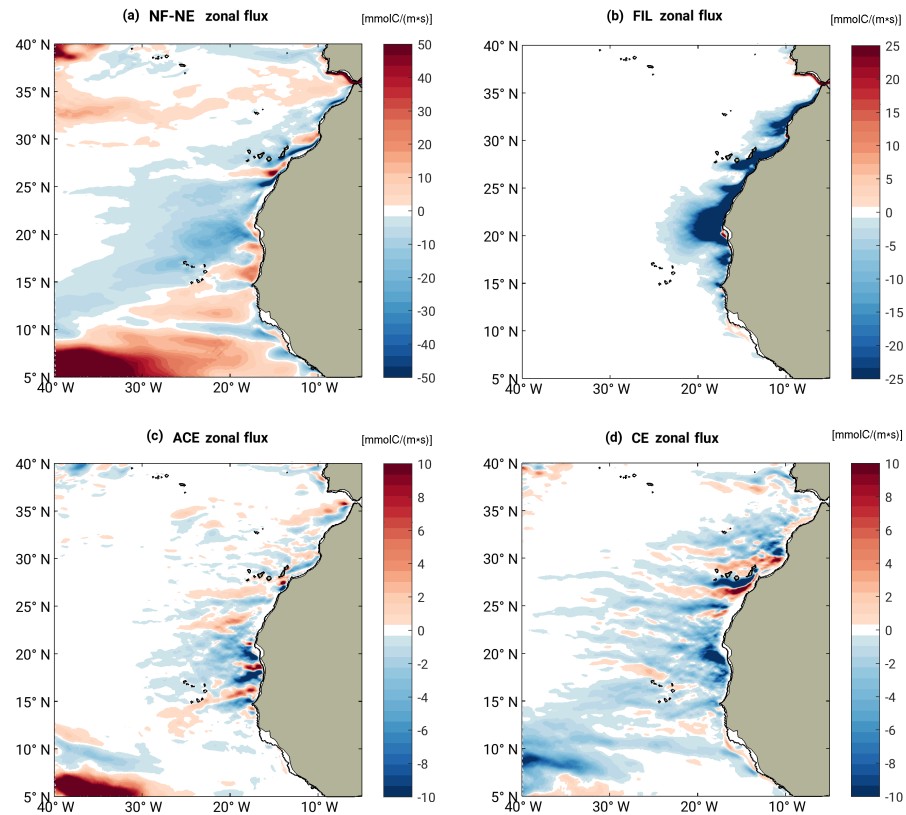

**Figure 11.** Decomposition of the zonal advective transport of $C_{org}$ into: (a) non-filament-non-eddy NF-NE , (b) filament FIL , (c) anticyclonic ACE and (d) cyclonic CE flux. Fluxes are integrated within the top 100m.

zonal advective speeds. Moreover, as filaments often shed long-living eddies that travel eastward with large amounts of $C_{org}$ and nutrients embodied from the filaments, they extend their influence deep into the open ocean.

    Across the whole domain, the eddy-associated fluxes (Figure 11c,d) are about five times smaller than the NF-NE fluxes. This is similar to the ratio found for the Reynolds decomposition based separation between the turbulent and mean transports (see also Figure 13). This is largely a consequence of the limited lateral drift speeds of eddies. But still, eddies have a large offshore reach due to the their long-range propagation. The eddy-associated offshore transport is primarily carried by CE, with ACE playing a minor role (Figure 12b). This is expected from ACEs having a lower $C_{org}$ content and a shorter life span, resulting also in an offshore decline of their surface area (see Figure 10 and Figure B5).

    The eddy offshore transport reveals also distinct striations, e.g., near zonal bands of offshore transport next to bands of onshore transport. These eddy striations are slightly deflected northward in the case of CE and southward in the case of ACE, consistent with the observed mean deflection of their trajectories (Chelton et al., 2011; McWilliams and Flierl, 1979).

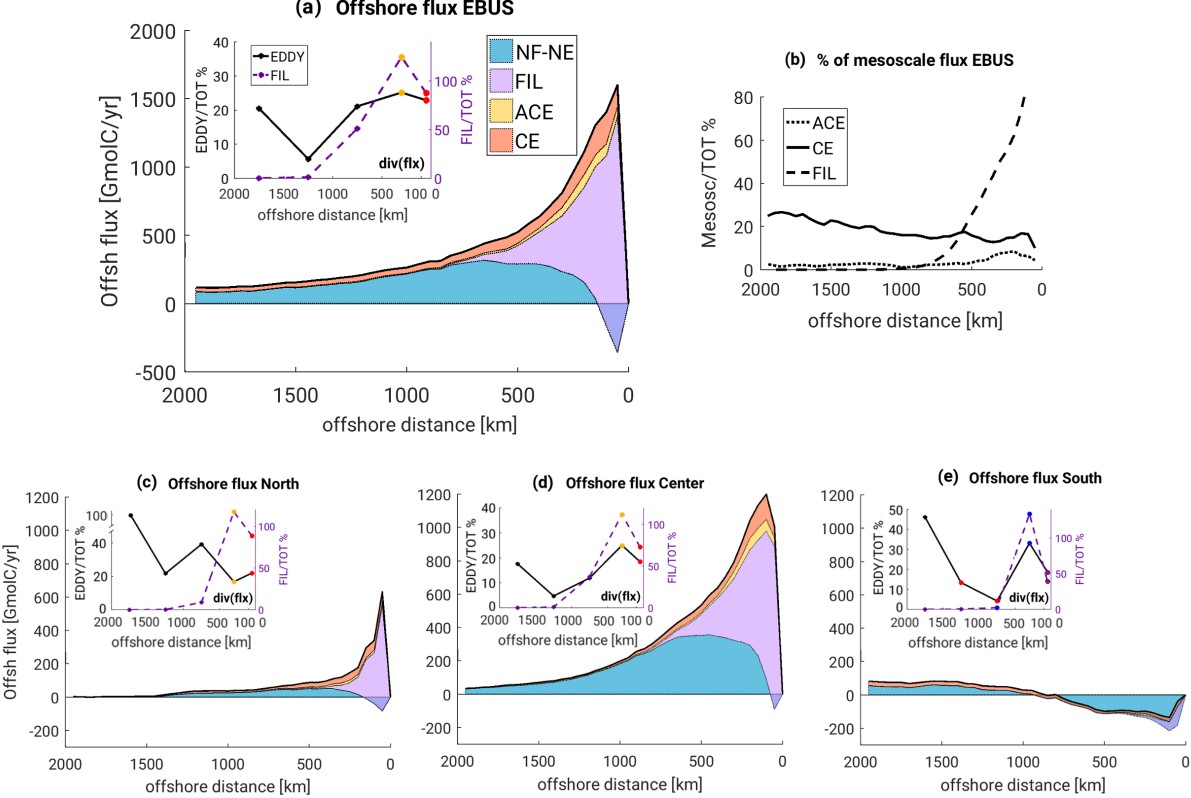

**Figure 12.** Magnitude of the non-filament-non-eddy (NF-NE), filament (FIL), anticyclonic (ACE), cyclonic (CE) components of the offshore flux [GmolC yr$^{-1}$]: (a) Canary EBUS as a whole; (c) northern CanUS; (d) central CanUS; (e) southern CanUS. In all area plots: the black thick line is the total offshore flux, sum of the 4 components; fluxes are integrated in the first 100 m depth. The purple area shading results from the overlapping of opposing NF-NE and filament flux contributions. Plot (b): only for the full EBUS we show the ratio of the magnitude of the mesoscale/total offshore flux for FIL, ACE and CE. Inserts: ratio between the absolute value of the divergence of the filament or eddy (ACE+CE) component and that of the total flux within the 100 m. Per each offshore range, divergences are calculated as finite differences at the boundaries shown on the x-axis. Dots are red when total and mesoscale divergences are negative (fluxes remove $C_{org}$); dots are yellow if only the NF-NE divergence is negative; dots have the same color of the line when all divergences are positive (fluxes add $C_{org}$). For the southern subregion only: blue dots indicate that the total flux divergence is negative and opposes the mesoscale flux divergence; dots are red with a blue edge when the opposite is true.

Integrated across the entire CanUS, ACE and CE together are responsible for about 20 % of the total offshore transport at every distance beyond the first 200 km from the coast, with the CE transport accounting for several times as much as the ACE transport (Figure 12b). As is the case for the filaments, the divergence of the eddy transport enhances the $C_{org}$ availability at every distance from the coast beyond the first 100 km, contributing between 10 % and 20 % to the total divergence. A larger

5  impact of the eddy flux divergence on the local $C_{org}$ stock is found in the northern subregion, where the NE-NF flux declines

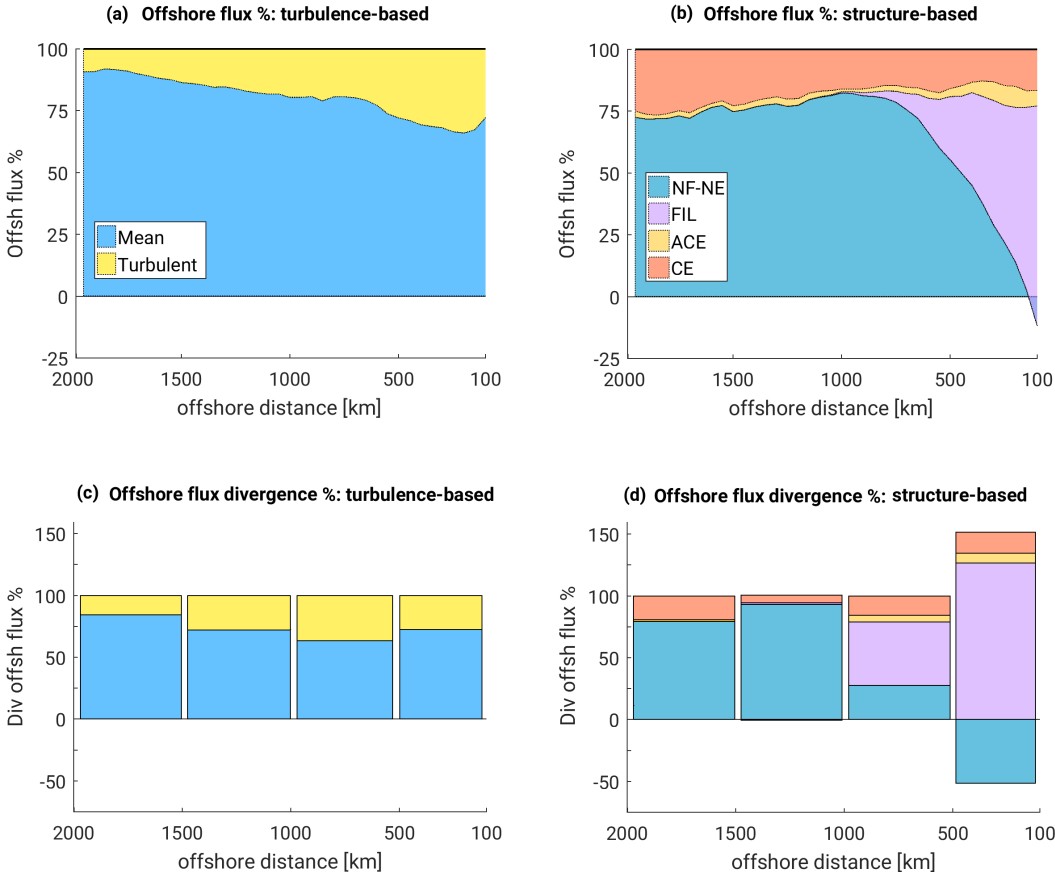

**Figure 13.** Comparison between the results of the turbulence-based and structure-based methods for the entire EBUS in the range of 100 km to 2000 km from the coast, where the divergence of the total offshore flux is positive, therefore resulting in an increase of the organic carbon availability (see also Lovecchio et al. (2017), Figure 10 for a quantification of the divergence of the total offshore flux). Percent of the total offshore flux carried on by each flux component: (a) the turbulence-based method; (b) structure-based method. Percent of the total offshore flux divergence by each flux component: (c) the turbulence-based method; (d) structure-based method.

quickly offshore, similar to what is seen in the mean Reynolds flux. Only in the southern CanUS eddies contribute with a positive flux divergence to the removal of $C_{org}$ between 500 km and 1000 km offshore.

Eddies and filaments have an important role also in the meridional and vertical redistribution of $C_{org}$ (see Appendix: Figure B10 and Figure B11). In particular, ACE are responsible for the northward recirculation of the $C_{org}$ through the asymmetric
5   stirring of the background regional gradient. This is a consequence of the unique combination of an offshore negative gradient (strongest in the nearshore) and a southward positive gradient in $C_{org}$ (Arístegui et al., 2009). Due to their relatively fast decay and the slowing down of the clockwise rotation while they move offshore, the ACE induce a net northward transport of $C_{org}$. This is because a younger and more energetic ACE stirs the $C_{org}$ northward in the nearshore areas, while an older and weaker

ACE stirs it southward in the offshore areas. CE, being more stable along their tracks, have a weaker effect on the net meridional transport.

In the vertical, eddies have a minor role in the transport of $C_{org}$ compared to the filaments, and their signature cannot be clearly distinguished from that of the NE-NF transport. Inspections of the model output reveal (not shown) that mesoscale vertical velocities in well-developed eddies present a dipolar or quadrupolar pattern of upwelling and downwelling similar to the one observed by in-situ measurements (Barceló-Llull et al., 2017). This compensating nature of the vertical velocities results in a minor net contribution of these mesoscale structures to the vertical flux. Filaments, on the contrary, are responsible for a strong nearshore downwelling, which is well captured by the turbulent component of the Reynolds fluxes (see Figure 5). The filaments advect below the euphotic layer a significant amount of $C_{org}$ in the range between 200 km to 500 km from the coast, with the maximum distance reached in the region of the Cape Blanc filament.

## 5    Discussion

### 5.1    Turbulence and the mesoscale: insights from two complementary perspectives

Our analysis of the mesoscale transport in the CanUS is based on two different, but complementary perspectives: A Reynolds decomposition of the fluxes into a mean and a turbulent component, and an independent decomposition of the total fluxes into an eddy, a filament and a non-filament-non-eddy component, derived from the use of filament and eddy masks.

In the nearshore areas, the filament zonal transport is very similar to the mean zonal flux from the Reynolds decomposition, both with respect to sign and its spatial structure. This implies that much of the zonal filament transport has a time-mean character. This is not surprising, given that filaments tend to be recurrent and localized features of the CanUS and are generally characterized by smooth and persistent flows during their lifetime (Navarro-Pérez and Barton, 1998; Arístegui et al., 2009). In a detailed study of the properties of a filament, Bettencourt et al. (2017) described the structure as a narrow corridor, in which the fluid progresses towards the offshore without deformation, i.e., with very little divergence of the trajectories along the path, confirming the non-turbulent character of the inner filament flow. Moreover, according to our filament detection algorithm, filaments are found in association with the major CanUS capes for more than 25% of the time, confirming their semi-permanent character (see Appendix: Figure B12).

Filaments dominate the mesoscale fluxes in the vertical direction, with an intense downwelling of organic matter to depths below 100 m occurring in the first 300 km away from the coast. Even though these structures have a typical thickness of about 100 meters or less (Cravo et al., 2010; Rossi et al., 2013), filament subduction well below their depths has been documented in observations (Kadko et al., 1991; Washburn et al., 1991; Brink, 1992; Barth et al., 2002; Peliz et al., 2004) and numerical studies (Nagai et al., 2015; Bettencourt et al., 2017; McWilliams et al., 2009). In contrast to the zonal flux, the pattern of the vertical downwelling associated with filaments clearly resembles the turbulent component of the vertical Reynolds fluxes. Animations of the vertical velocities at 100 m depth in the filaments show transitory and irregular hotspots of strong downwelling, which evolve quickly and often drift laterally following the filament flow. Therefore, the vertical transport inside filaments is highly variable both due to horizontal movements of the downwelling cells and due to the intermittency of the process. With a

numerical study, Nagai et al. (2015) showed that this subduction happens primarily at the periphery of the filament due to the formation and degradation of frontal and high-strain regions at the filament edges.

The eddy zonal transport resembles, for both kinds of eddies, the structure and magnitude of the turbulent Reynolds flux. Eddies are mostly transitory structures that drift while rotating around their axis and are characterized, through the course of their life, by oscillations both of the dimension of their radius and of their rotational speed (Sangrà et al., 2005). This high variability explains why the lateral eddy transport mostly projects onto the turbulent transport. However, the CE and ACE zonal transports are also characterized by striations, which are not seen in the turbulent fluxes from the Reynolds decomposition. These striations suggest the existence of recurrent regions of formation and preferential pathways for these eddies. An excellent example is visible south of the Canary archipelago, where recurrent CE form near the coast (Barton et al., 2004) and drift offshore. This interpretation is supported by the fact that these striations show up in the time-mean Reynolds fluxes as well.

Further differences between the mesoscale and turbulent offshore flux components can be caused, for example, by the existence of other forms of turbulence that we are not accounting for with our eddy and filament masks. One of these contributions could be that of offshore fronts and filaments between eddies, which evolve in time in close connection with the detected mesoscale field.

## 5.2 Comparison with previous work

Our results highlight the importance of the mesoscale transport for the lateral export of $C_{org}$ from the north-western African coast. In particular, they point to the importance of filaments and eddies as a source of organic carbon and nutrients for the offshore oligotrophic waters.

The filament $C_{org}$ transport in the euphotic layer of the CanUS extends in our model up to 1000 km offshore, even though its intensity declines quickly offshore. This reach is larger than in previous observations, which indicated about 700 km of maximum offshore extension for the giant Cape Blanc filament, as seen from satellites (Ohde et al., 2015). However, our analysis shows that the filament offshore transport reaches the largest offshore extension at a depth of 20 m to 30 m. A few filament transects from in-situ measurement also show that the filament lateral transport of biogeochemical tracers deepens below the surface to depths of a few tens of meters as the filament is moving away from the continental shelf (Cravo et al., 2010; Rossi et al., 2013). This implies that the total offshore reach of a filament may not be visible from remotely sensed surface chlorophyll or SST, resulting in a underestimation of its range.

Our modeled mean filament offshore transport in the whole CanUS region amounts to about 2 Sv between 100 km and 200 km from the coast and integrated over the top 100 m depth. This corresponds to 0.7 $m^2$ $s^{-1}$ in transport per unit of length, which largely exceeds the estimated offshore Ekman transport per unit length in the CanUS, roughly between 0.4 $m^2$ $s^{-1}$ and 2 $m^2$ $s^{-1}$ during the upwelling (Arístegui et al., 2009; Barton, 2001). The filament transport increases to 3 Sv if the top 300 m are considered. A large fraction of this transport originates in the northern and central CanUS subregions. Our intensity is lower than previous estimates of 6 to 9 Sv for the same subregions (Barton et al., 2004), but it represents still a very large contribution to the offshore flux. One possible reason for our model's underestimation is the fact that it represents a climatological yearly average including the seasons of reduced filament activity. In contrast, observations are often done in periods of intense filament

transport, possibly biasing high the estimates upscaled from relatively few observations. Even though typical zonal velocities observed inside modelled filaments (100 m depth averages) are about 0.15 m s$^{-1}$, the flow often reaches speeds of about 0.3 m s$^{-1}$ and can peak at more than 0.5 m s$^{-1}$, in agreement with in situ observations (Pelegrí et al., 2005).

Consistent with previous studies (Arístegui et al., 2009; Pelegrí et al., 2005), most of the filaments in the CanUS are found in the northern subregion. However, our model simulates also a substantial amount of filament activity south of Cape Blanc in winter and in April and May, in agreement with Menna et al. (2016). Even though these seasonal filaments are shorter (rarely extending beyond 200 km) and less persistent than the ones observed in the north, their high occurrence along the coast in the season of intense activity is particularly striking. This strong seasonality makes it difficult to appreciate their possibly large, but intermittent impact on the local $C_{org}$ budget. Dedicated studies and in situ observations may help to better understand these structures, currently only partially characterized in the literature.

Our modeled eddies are in great part generated in the first 250 km from the coast, and very often interact with coastal filaments which feed them with tracers in a filament/eddy coupled system, as previously observed (Barton et al., 2004; Arístegui et al., 1997; Meunier et al., 2012). Our eddy translational velocities as well as the northward or southward deflection of the eddy tracks visible in the striations of the CE and ACE offshore transport agree with observations of global mesoscale eddy tracks (Chelton et al., 2011). The modeled eddy offshore flux extends far into the open ocean up to the edge of our analysis domain, propagating offshore for several months (Sangrà et al., 2007). The integrated eddy offshore transport in the first 100 m depth of the whole CanUS ranges from 1 Sv at 200 km from the coast to 0.7 Sv offshore, with 80 % of the transport taking place in the northern and central subregions. This is a remarkable lateral flux comparable to that of other major currents in the region, such as the Canary Current (1.5 Sv to 3 Sv), the Canary Upwelling Current (1 Sv to 1.5 $\pm$ 0.3 Sv) and the North Equatorial Current (0.5 Sv to 3 Sv) (Machín et al., 2006; Pelegrí and Peña-Izquierdo, 2015; Mason et al., 2011). In line with the results of Combes et al. (2013), we find that CE are responsible for a large part of the offshore tracer transport.

The negative correlation between SSH$'$ and $C_{org}'$ (Figure 9) agrees roughly with the satellite-derived cross correlation of satellite SSH and surface chlorophyll for our region (Gaube et al., 2014, Figure 1a). Differences in the $C_{org}$ concentration in the two kinds of eddies depend both on their $C_{org}$ availability at the moment of formation, and on the evolution of the tracers during their life. Animations of the $C_{org}$ field show that latitudinal differences in the $C_{org}'$ found in ACE and CE at their formation can be attributed to the opposite direction of the meandering coastal current from which they are shed, combined with the offshore gradient in $C_{org}$ (Gaube et al., 2014): This results in high $C_{org}$ eddy-core concentrations at formations in northern CE and southern ACE.

As highlighted in previous studies (Gruber et al., 2011; Lachkar and Gruber, 2011), the CanUS is characterized by a mesoscale activity that, even though important, is not as intense as in other upwelling systems. Therefore, we expect mesoscale activity to have an even more substantial impact in other EBUS regions. In spite of the presence of one giant filament in the CanUS, i.e., the Cape Blanc filament, filaments in the California Upwelling System (CalUS) were observed to extend, on average, twice as far into the open ocean (Marchesiello and Estrade, 2006). Moreover, for the CalUS, the STD(SSH) and the cross-shore eddy diffusivity exceeds by far that of the CanUS (Marchesiello and Estrade, 2006; Lachkar and Gruber, 2011), and the number of observed eddies from satellite data is substantially larger than that found in the CanUS at most latitudes

(Chaigneau et al., 2009). In the analysis presented by Nagai et al. (2015), the turbulent transport of $C_{org}$ in the CalUS, obtained with a Reynolds decomposition, represents 20-25 % of the total offshore fluxes at the surface and is dominant further below. Thus, even though the level of mesoscale activity is substantially larger in the CalUS, the fractional transport by mesoscale processes is only slightly larger than in the CanUS. This might also be caused by the eddy/filament induced subduction of organic matter, which is particularly strong in the CalUS, thus reducing the eddy-induced horizontal transport. A further quantification of the offshore transport by filaments and eddies based on the identification of the mesoscale structures in other upwelling region may help to better clarify the importance of nearshore and offshore mesoscale fluxes across the different systems.

## 5.3 Model limitations

As is the case for every model-based study, the impact of the inherent model limitations need to be clearly identified and put into relationship with the results. The first set of shortcomings regards model biases; the second set regards potential limitations of the eddy and filament identification algorithms; in the end we discuss our lack of representation of the DOC pool.

As identified and discussed by Lovecchio et al. (2017), our model underestimates the intensity of the flow in the southern CanUS subregion; concurrently, modeled POC shows a too-deep maximum in this CanUS sector compared to observations. The combination of these biases could result in an underestimation of the total offshore transport at these latitudes. For this reason, while we have good confidence in our results regarding the lateral $C_{org}$ transport in the northern and central CanUS, the results for the southern CanUS are more uncertain. The weaker circulation in the southern CanUS is reflected also in the underestimaton of the TKE at the same latitudes, especially along the coast and in the proximity of the Cape Verde archipelago (Figure 2). Even though the southern CanUS is known for having a modest filament activity and a reduced eddy activity compared to the northern CanUS (Arístegui et al., 2009; Chaigneau et al., 2009), our model enhances these zonal differences. We may argue that the weakening of the mean circulation implies the weakening of the turbulent flow with a similar factor, possibly not affecting the ratio between the two contributions, which is of interest in the present analysis.

In terms of eddies, our evaluation shows that the modeled large-eddy field (R $\geq$ 50 km) is only about two thirds of that seen in observations. However, differences in the performance of the eddy-finding algorithm, when applied to the ROMS and AVISO grids, must be taken into account. Given the different resolution, the SSH-based algorithm may, on average, be able to find smaller closed SSH contours around a SSH maxima/minima in the higher-resolution Atlantic telescopic in comparison to the AVISO grid. Given the abundance of small and medium eddies in the modeled field and the positive results of the evaluation of the integrated area covered by ACE and CE in the model, we expect our integrated eddy transport to be close to the real value.

In our analysis, we do not distinguish between regular ACE and anticyclonic mode-water eddies (ACMEs). As a result, the latter are included in our ACE budget and transport. These eddies have been observed in the southern CanUS subregion and may represent about 20 % of the total ACE population at these latitudes (Schütte et al., 2016a). Given the observed high productivity of ACMEs compared to regular ACE (Schütte et al., 2016b), including these eddies in the ACE budget likely increases the integrated ACE organic carbon availability and transport. Further analyses would be needed to correctly separate the ACMEs from the ACE contribution.

Our newly-developed SST-based filament detection algorithm was tested on our grid with satisfactory results, with the large majority of the filaments being detected and only a very limited over-detection in the southern CanUS. The algorithm performs particularly well in the zonal band located north of Cape Blanc, given the sharp offshore SST gradient. However, south of Cape Blanc, the detected filaments appear generally shorter than a visual inspection might suggest. Only during a few time steps we could see an over-detection of the extent of the filaments in the area surrounding the Cape Verde archipelago. Give this performance, we conclude that the our modeled filament transport is well represented by the filament-mask analysis. We refer to the supplement to the manuscript for more details and an accurate evaluation of the algorithm performance on our setup.

Mesoscale structures and especially filaments are known to export large quantities of DOC, which can represent up to 70 % of the total $C_{org}$ transported offshore (Santana-Falcón et al., 2016; García-Muñoz et al., 2005). However, only a small part of this exported DOC has a lability that contributes to a divergence, i.e., is being remineralized while it is being transported offshore (Hansell et al., 2009). Our model does not include a DOC pool. However, our small detritus pool behaves very similarly to a suspended POC pool given its very small sinking speed of 1 m day$^{-1}$. The possible repercussions of the lack of modeling of DOC in our model were already discussed by Lovecchio et al. (2017), where we also presented the results of a sensitivity study, in which we tested the implications of modeling a purely suspended POC pool for our lateral export and the fueling of the biological activity. The results of this sensitivity experiment demonstrated that, even though the total offshore transport can increase as a result of the shallower POC distribution, the divergence of the transport remains basically unchanged, with little repercussion on the offshore productivity.

## 6   Conclusions

Our study in the CanUS confirms that mesoscale processes contribute substantially to the long-range offshore transport of $C_{org}$ through a combination of mean and turbulent transport. In particular, filaments drive the total offshore flux of $C_{org}$ and its divergence in the nearshore, while far-reaching eddies, especially cyclones, extend this transport up to the middle of the gyre. The divergence of the mesoscale transport allows extra vertical export out of the productive layer and strongly contributes to the shaping of the pattern of nearshore net autotrophy and offshore net heterotrophy of the water column (Lovecchio et al., 2017). Even though the CanUS has moderate levels of mesoscale variability in comparison with other EBUS, the mesoscale contribution to the transport and to the fueling of the offshore biological activity strongly dominates in the nearshore 1000 km and amounts already to about 20% at larger offshore distances. This suggests that this mesoscale contribution may be even more crucial in other upwelling regions that have higher nearshore-generated mesoscale activity (Capet et al., 2013; Marchesiello and Estrade, 2006), such as the California Upwelling System (Nagai et al., 2015).

The key role of mesoscale processes in the lateral $C_{org}$ transport has several consequences. First, it implies that coarse global models may be unable to account for a great part of this flux out of the upwelling regions, possibly failing at reproducing the offshore rates of deep respiration and at fully capturing the three-dimensionality of the biological pump. Second, remote sensing approaches may underestimate this offshore transport. On the one hand, this may be due to the time-limited sampling owing to the frequent cloud cover preventing the detection of the chlorophyll and associated carbon content. On the other hand,

because of our modeled filament transport deepening offshore up to a few tens of meters below the surface, i.e., below the detection level of satellites, leading to a potential underestimation of the actual offshore reach of filaments. Third, the relevant share of $C_{org}$ found in the offshore region within mesoscale eddies also tells us that a fraction of the offshore biological activity is fueled discontinuously at the passage or death of such a structure. In particular, both the transit of an eddy, which is associated with enhanced vertical export (Subha Anand et al., 2017; Waite et al., 2016), and the death of an eddy provide a discontinuous, but substantial, input of carbon for the oligotrophic waters.

While our study shows relevant levels of eddy productivity offshore, further analyses are needed to disentangle the pathway of new production and recycling of the $C_{org}$ along long-lasting tracks of the northern and southern CanUS, and to understand the special role of mode water anticyclones in the budget and transport. Further studies can also help to better quantify the highly seasonal contribution of the many short filaments of the southern CanUS, of which little is known, and to investigate the role of the offshore transport of dissolved $C_{org}$, not included in our model.

## Appendix A:  Datasets used for the model evaluation

| Data source | Ref. time | Resolution | Variables | Reference |
|---|---|---|---|---|
| Aviso DUACS 2014 | 1993-2012 | 0.25°x 0.25°grid | daily sea surface height, daily geostrophic velocities | Maheu et al. (2014) |
| AVHRR | 1981-2014 | 0.25°x 0.25°grid | sea surface temperature | Reynolds et al. (2007) |
| CARS | 1955-2003 | 0.5°x 0.5°grid | sea surface salinity | Ridgway et al. (2002) |
| Aviso CMDT Rio05 | 1993-1999 | 0.5°x 0.5°grid | sea surface height | Rio and Hernandez (2004) |
| Argo DT-0.2 | 1941-2008 | 2°x 2°grid | mixed layer depth | Montégut et al. (2004); Argo (2000) |
| SeaWiFS | 1997-2010 | 9km grid | sea surface chlorophyll | NASA-OBPG (2010) |
| SeaWiFS VGPM | 1997-2010 | 9km grid | extrapolated net primary production (NPP) | Behrenfeld and Falkowski (1997) |
| SeaWiFS CbPM | 1997-2010 | 9km grid | extrapolated net primary production (NPP) | Westberry et al. (2008) |
| SeaWiFS POC | 1997-2010 | 9km grid | extrapolated surface particulate organic carbon (POC) | NASA-OB.DAAC (2010) |
| AMT | (2004-2014) | in-situ [0m,200m] depth | particulate organic carbon | BODC-NERC (2014) |
| Geotraces | (2010) | in-situ surface | particulate organic carbon | GEOTRACES (2010) |
| ANT | (2005) | in-situ [0m,200m] depth | particulate organic carbon | ANT (2005) |

**Table A1.** Description of the datasets used for the model evaluation. CMDT: Combined Mean Dynamic Topography; AVHRR: Advanced Very High Resolution Radiometer; CARS: CSIRO Atlas of Regional Seas; SeaWiFS: Sea-viewing Wide Field-of-view Sensor; WOD09: World Ocean Database 2009.

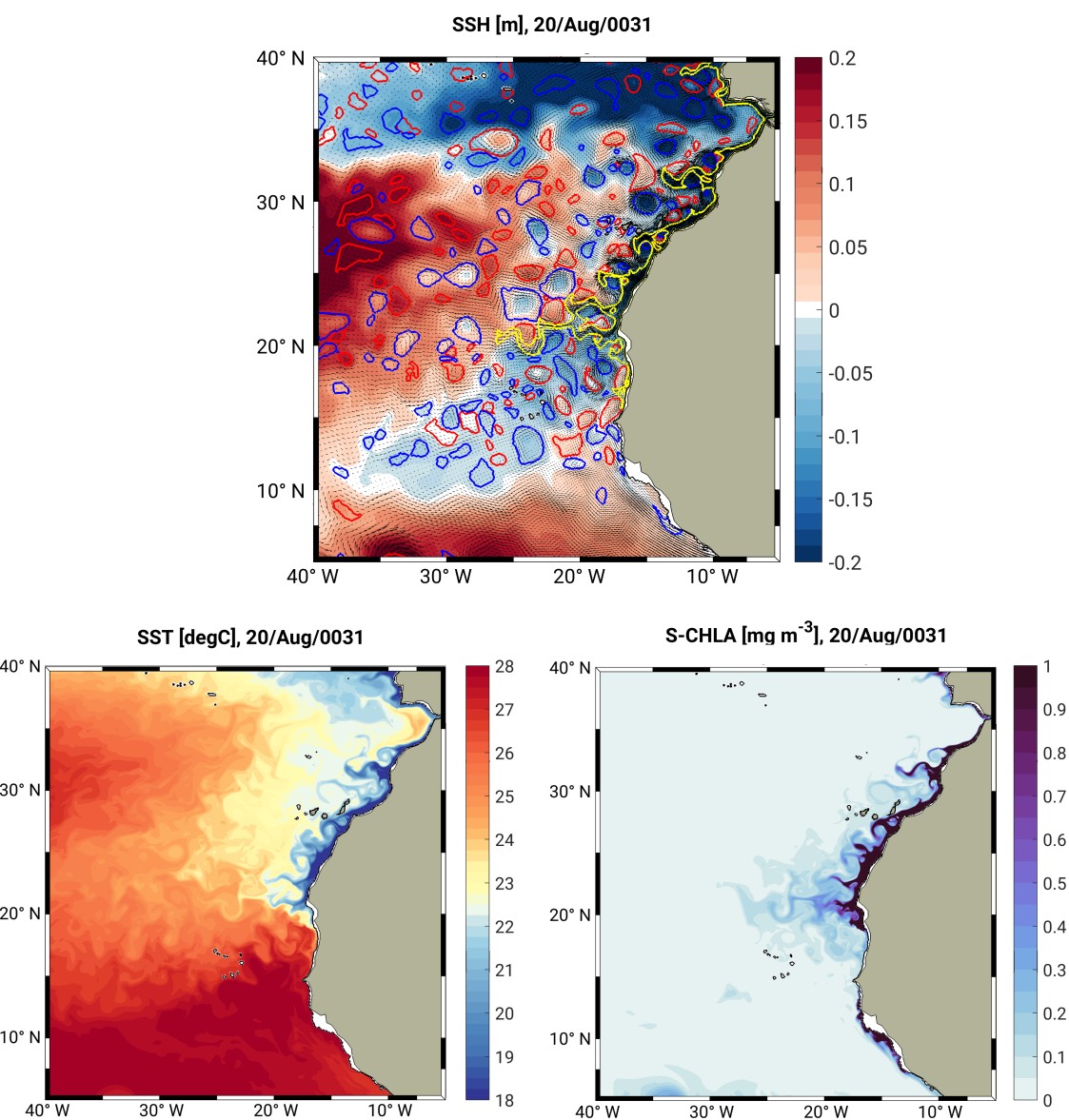

**Figure B1.** Example of the performance of the filament and eddy identification algorithms on 2-day mean fields. Subplot (a): red contours are warm-core anticyclonic eddies; blue contours are cold-core cyclonic eddies; yellow contour are coastal upwelling filaments. Subplot (b): sea surface temperature. Subplot (c): surface chlorophyll. The eddy identification algorithm is SSH-based, as in Faghmous et al. (2015). The filament identification algorithm is SST-based as described in the Methods section.

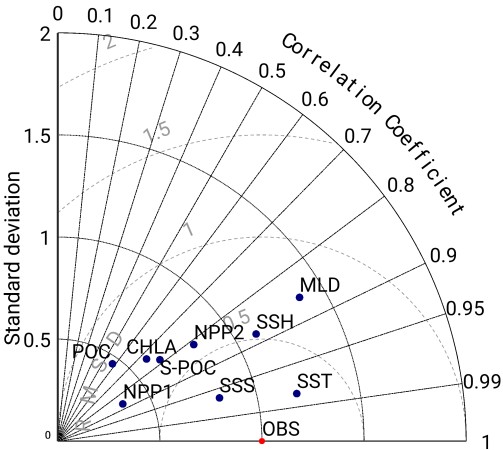

**Figure B2.** Taylor diagram for the Canary EBUS region of analysis as defined by the Budget boxes based on the 24 years climatoligigal annual mean fields used in the present study. Used datasets - Sea Surface Temperature (SST): AVHRR, Sea Surface Salinity (SSS): CARS, Sea Surface Height (SSH): Aviso CMDT Rio05, Mixed Layer Depth (MLD): Argo DT-0.2, Chlorophyll (CHLA): SeaWiFS, Net Primary Production dataset 1 (NPP1): SeaWiFS VGPM, Net Primary Production dataset 2 (NPP2): SeaWiFS CbPM, Surface Particulate Organic Carbon (S-POC): SeaWiFS POC, Particulate Organic Carbon (POC): cruise POC data (AMT, ANT, Geotraces). A detailed description of the data used for the evaluation is provided in Appendix A: Datasets, Table A1.

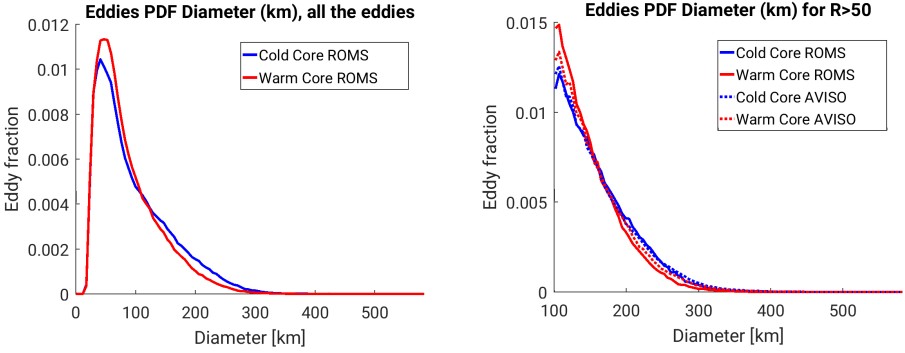

**Figure B3.** Distribution of the eddy diameters [km] in the CanUS region as defined by the budget analysis boxes: (a) Propability density function (PDF) of the diameter of all the eddies from ROMS; (b) comparison of the PDFs limited to large eddies (R > 50 km) for the eddies from ROMS and AVISO.

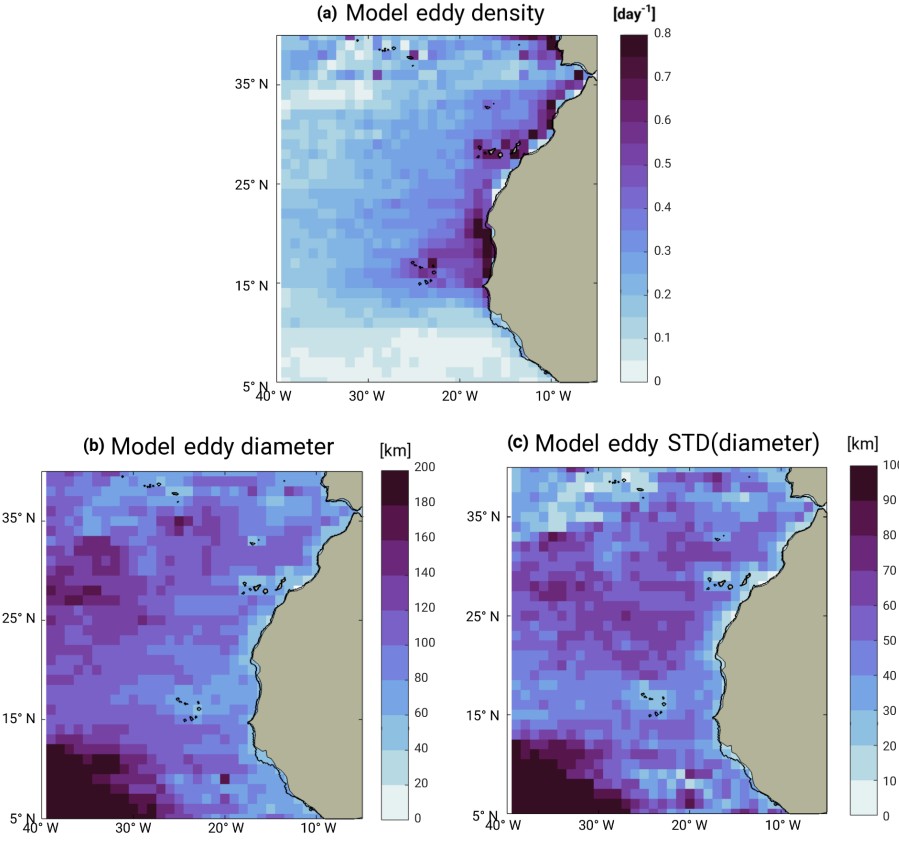

**Figure B4.** Statistics for all the modeled eddies with R≥15 km. (a) Modeled density of eddy centroids per day; (b) Modeled mean eddy diameter [km]; (c) Modeled Standard deviation of the eddy diameter [km]. All quantities have been averaged in 1deg x 1deg bins.

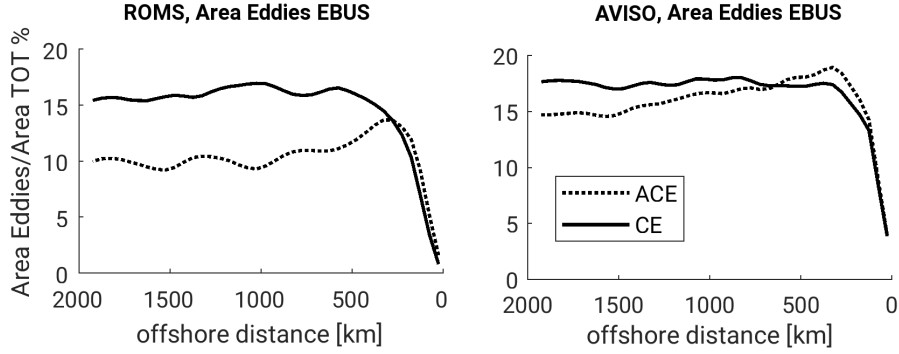

**Figure B5.** Portion of the CanUS (EBUS) occupied by the eddies, by eddy type. (a) Modeled ROMS eddy field; (b) AVISO eddy field.

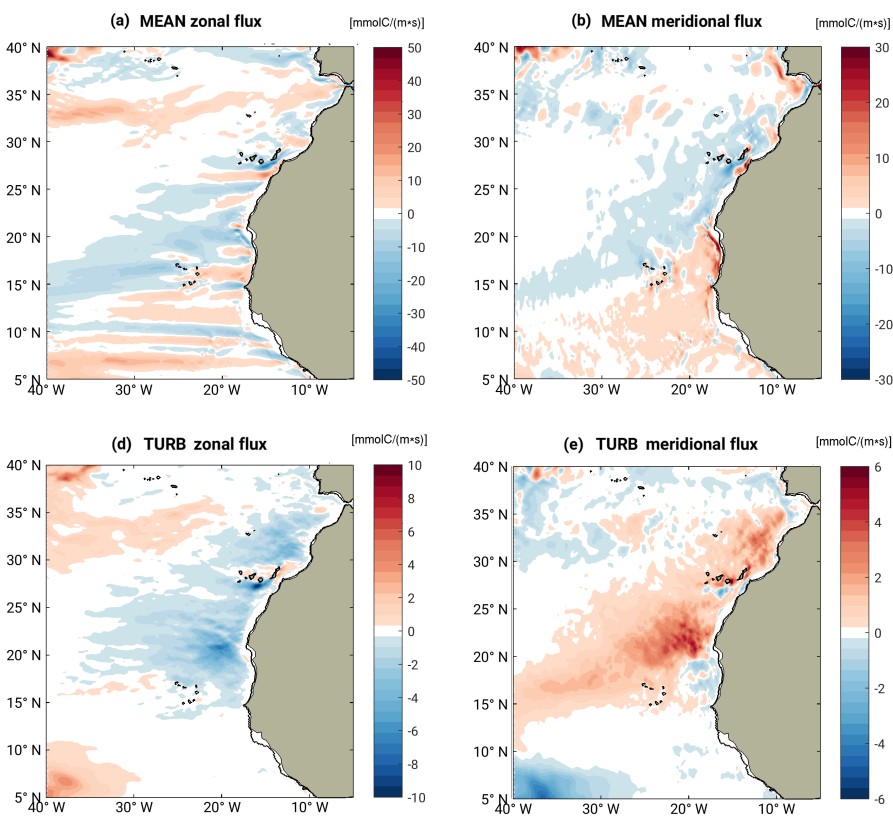

**Figure B6.** Reynolds decomposition of the lateral and vertical advective fluxes of $C_{org}$ below 100 m into their average mean $\langle \overline{\boldsymbol{u}}\,\overline{C}_{org} \rangle$ and average turbulent $\langle \boldsymbol{u}'C'_{org} \rangle$ components, as defined in the methods section. Fluxes integrated from 100 m depth to the bottom of the water column. Colorscale corresponding to those of Figure 5 for a better comparison.

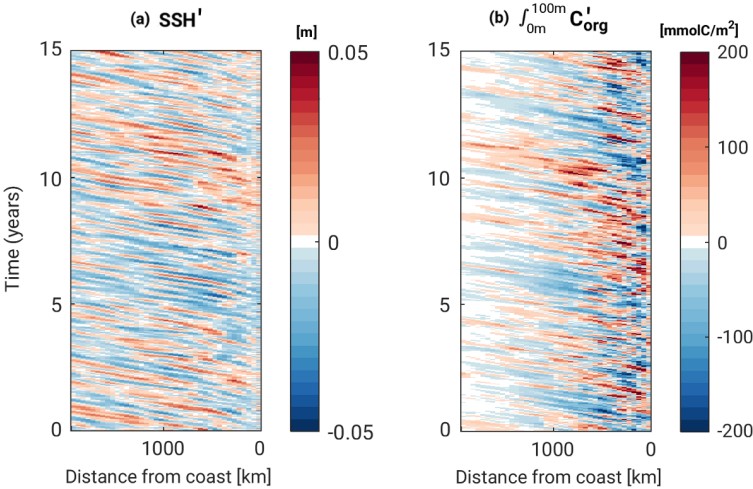

**Figure B7.** Hovmoeller diagram of the turbulent SSH anomalies (SSH′) and top 100 m organic carbon stock $C_{org}$ ($\int_0^{100m} C'_{org}$) for the central subregion of the CanUS. First 15 years of analysis data, bi-daily output. Their signature shows remarkable interannual variability despite the climatological forcing used for our simulation, highlighting the intrisic nature of turbulence and variability in the region.

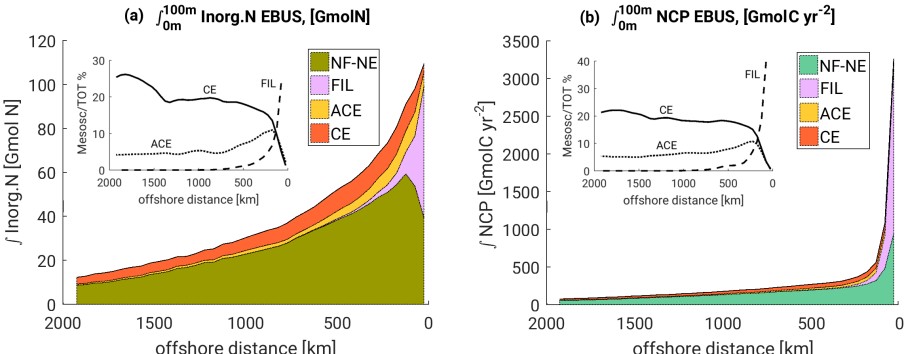

**Figure B8.** Inorganic nitrogen (Inorg.N) and Net Community Production (NCP) contained at different distances from the coast in filaments (FIL), cyclonic eddies (CE), anticyclonic eddies (ACE) and outside of the mesoscale structures (non-filament-non-eddy, NF-NE). Both quantities are integrated in the first 100 m depth and across the horizontal extension of the CanUS. Inserts: Percentage of the total Inorg.N and NCP found within ACE, CE and FIL, by offshore distance; the FIL share of Inorg.N and NCP in the first 50 km offshore is not plotted since their contribution in this range cannot be clearly identified.

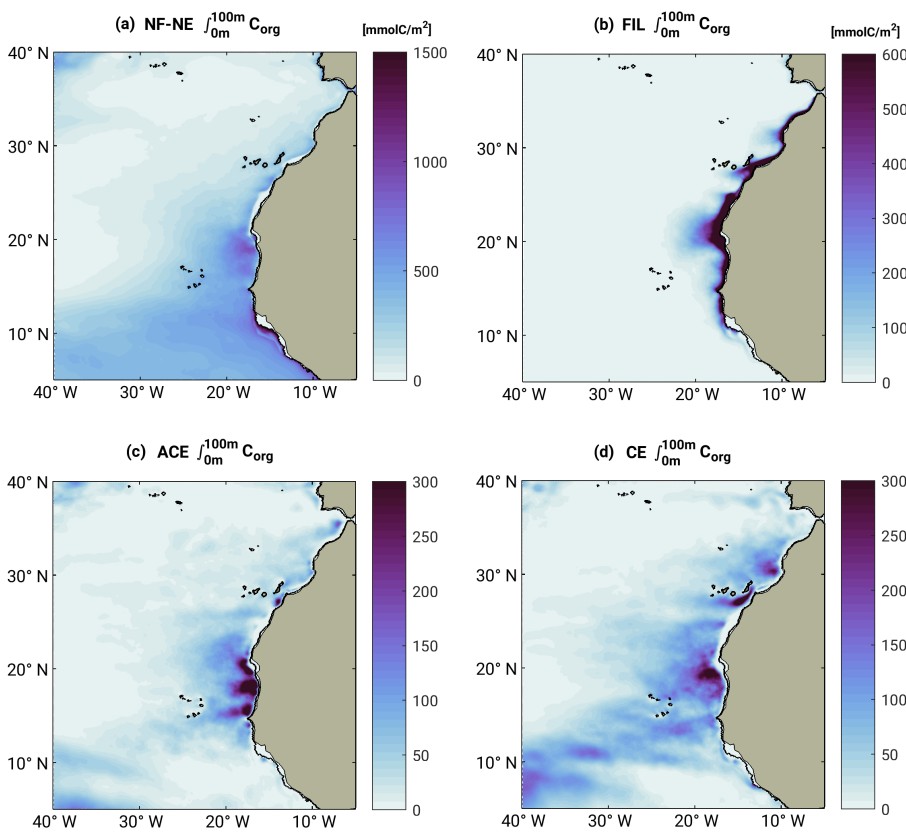

**Figure B9.** Spatial distribution of the 100m vertically integrated $C_{org}$ [mmolC m$^{-2}$] in: (a) non-filament-non-eddy (NF-NE) field, (b) filaments (FIL), (c) anticyclones (ACE), (d) cyclones (CE).

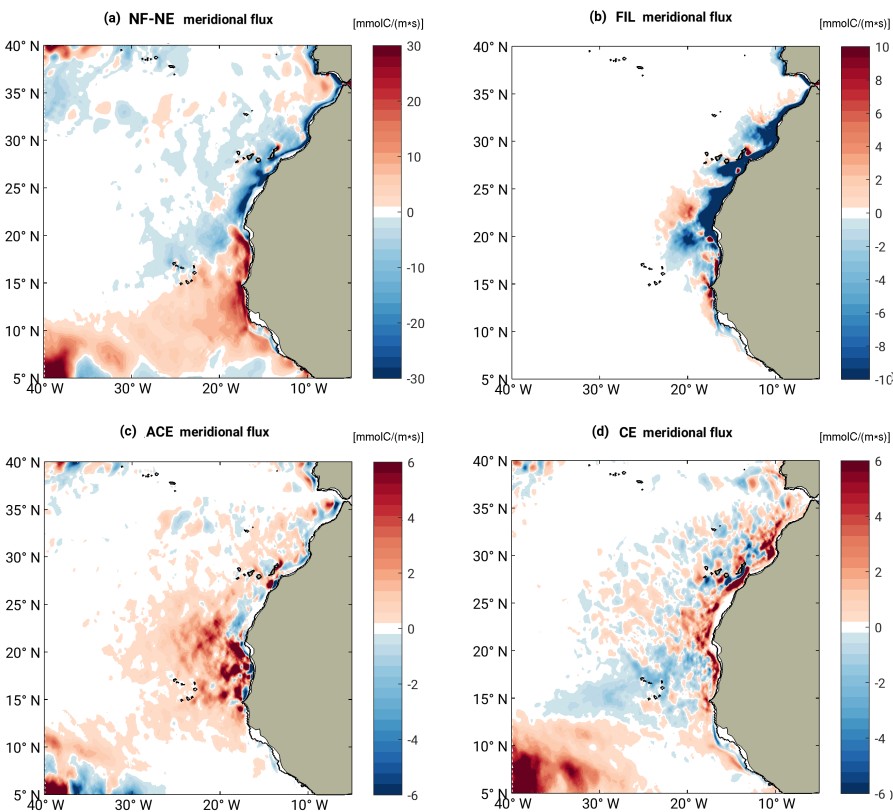

**Figure B10.** Decomposition of the meridional advective transport of $C_{org}$ into: (a) non-filament-non-eddy (NF-NE) flux, (b) filament (FIL) flux, (c) anticyclonic (ACE) flux and (d) cyclonic (CE) flux. Fluxes [mmolC m$^{-1}$ s$^{-1}$] are integrated in the first 100m depth.

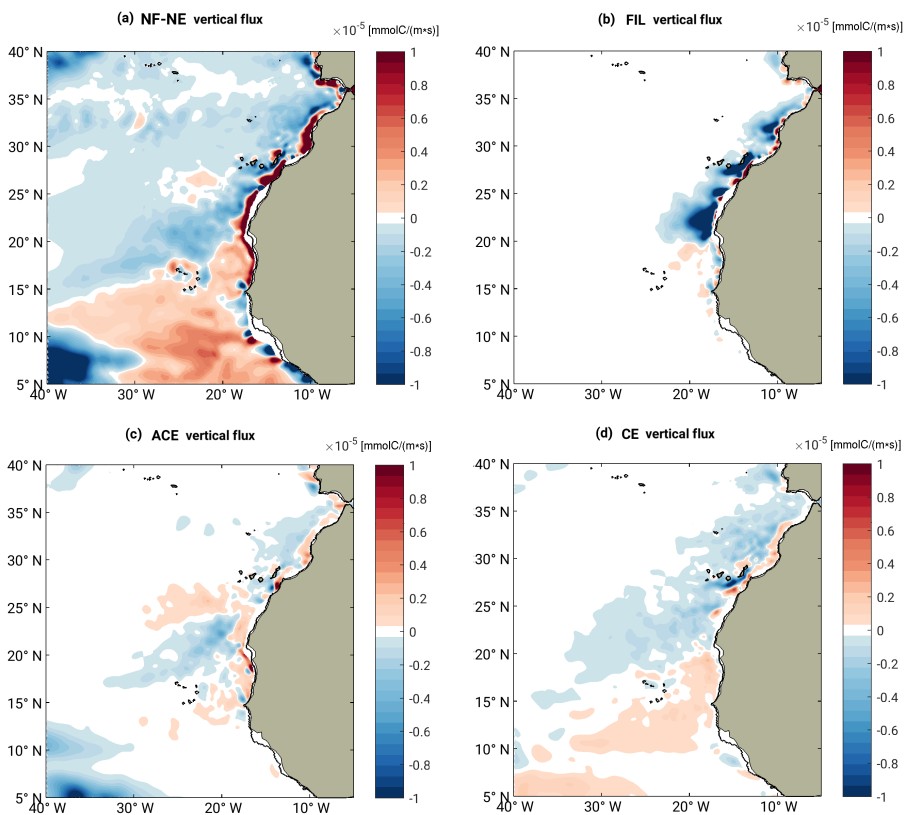

**Figure B11.** Decomposition of the vertical advective transport of $C_{org}$ into: (a) non-filament-non-eddy (NF-NE) flux, (b) filament (FIL) flux, (c) anticyclonic (ACE) flux and (d) cyclonic (CE) flux. Fluxes [mmolC m$^{-2}$ s$^{-1}$] are sliced at a depth of 100m.

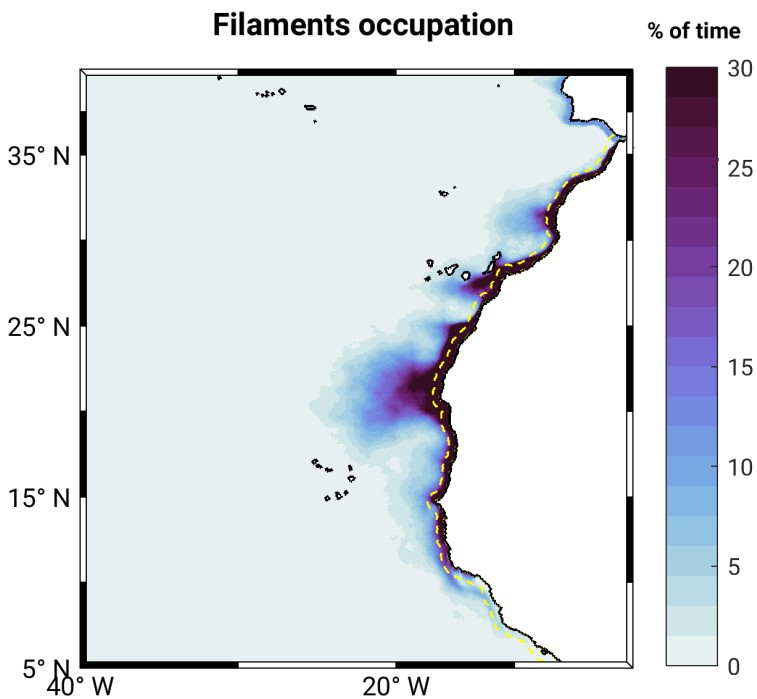

**Figure B12.** Percentage of the total time in which each grid node is occupied by a filament, according to our filament identification algorithm. The boundary corresponding to the first 50 km from the north-western African coast has been highlighted with a dashed yellow line. Regardless of the fact that this range of distances is often covered by the filament mask, alongshore currents are dominant in this region.

*Author contributions.* N.G. and E.L. conceived the study. E.L. and M.M. set up the experiment and improved the model. E.L. performed the analysis. All authors contributed to the interpretation of the results and to the writing of the manuscript. N.G. and M.M. supervised this study.

"The authors declare that they have no conflict of interest."

5 *Acknowledgements.* We would like to thank Martin Frischknecht for his valuable comments on the work and during the preparation of the manuscript, and Damian Loher for the technical support. We also thank Dr. Ivy Frenger for her fundamental help and suggestions that have substantially improved our analysis. This research was financially supported by the Swiss Federal Institute of Technology Zürich (ETH Zürich) and the Swiss National Science Foundation (Project CALNEX, grant No.149384). The simulations were performed at the HPC cluster of ETH Zürich, Euler, which is located in the Swiss Supercomputing Center (CSCS) in Lugano and operated by ETH

10 ITS Scientific IT Services in Zurich. Model output is available upon request. Please contact the corresponding author, Elisa Lovecchio (elisa.lovecchio@usys.ethz.ch), in that matter.

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
