# Peer review of "Mesoscale contribution to the long-range offshore transport of organic carbon from the Canary Upwelling System to the open North Atlantic"

_Biogeosciences, 2018_

## Referee Comment (RC1) · Anonymous Referee #1 · 10 Apr 2018

This study investigates the role of eddies and filaments in offshore carbon transport in the Canary Upwelling System, using Reynolds decomposition, and structure based eddy and filament decomposition to the results from a coupled biogeochemical model. The results indicate that the filaments provide the most intense and fast lateral transport within the regions 1000 km from the coast, with long-range isolated conduits to the farther offshore regions >2000 km caused by mesoscale eddies, with the dominant contribution from cyclonic eddies. These results are important findings for Canary Upwelling System, where there has been no such comprehensive study conducted so far. However, I have several concerns, which should be addressed by the authors before its publication. One of these issues is that the mean vertical carbon flux presented

in Figure 6c and 8c. There seems to be clear contrast between northern and southern regions separated at C. Blanc in plan view of the vertical carbon flux, which looks somewhat similar to the distribution of the mean meridional flux in Figure 6b. Also, in the vertical section, there is an upward flux at 60-70 m depths, right under and above the downward carbon fluxes. I'm puzzled by these mean vertical and meridional carbon flux distributions, and consider that this needs to be clarified. Also in some part, although the authors tried to explain the resultant interesting patterns obtained by the Reynolds decomposition and structure based extraction of eddy and filament contributions, they are often stated without sufficient evidence and would make readers feel that they are rather speculative. For example, the high covariance between anticyclonic eddies and carbon anomaly in northern offshore, is explained by the upwelling induced by the anticyclonic eddy spin down, without any analysis. To be concise, section 6 and 7 could be combined into one section. Other specific comments are as follows.

P2L13 "Coastal filaments are narrow (<50 km wide) structures that extend from the coast to several hundred kilometers into the open sea with rather large velocities (between 0.25 and 0.5 m s−1) " Is this velocity in "with rather large velocities (between 0.25 and 0.5 m s−1)", propagation speed or current velocity?

P2L15 "Offshore transport by filaments is typically accompanied by intense subduction, due to the high density of the cold upwelled waters" The subduction is intense in the cold filaments because frontogenesis at a cold filament generate two cell ageostrophic cross-frontal circulations, which work together to restratify the two fronts at the edge of the filament, and to narrow the width of the filament, producing the strong downwelling in the filament. This sentence seems not accurate, and I suggest that it should be modified with a reference of McWilliams et al. 2009, Cold filamentary intensification and oceanic surface convergence lines".

P2L26 "These non-linear structures propagate with velocities of a few centimeters per hour, about one order of magnitude slower than the filaments . . ." "a few centimeters per hour" is 10 $\mu$ms-1, and should be too slow for the mesoscale eddy propagation

speed.

P4L17 "CanUS also showed that eddies tend to reduce coastal production through the lateral export of the upwelled nutrients " I think that the reduction is caused by both lateral and vertical nutrient transport.

P5L8 "F =F +F′, were " Typo. "were" should be "where".

P6L4 "The last term r = uC′org + u′Corg term represents the sum of the residuals which we verified to be small, at least one order of magnitude or more smaller than the other terms" The original rule of the Reynolds decomposition is that the mean of the fluctuation is zero, and the mean of the mean is the same mean, but the authors' choice allows non-zero values for the former, as we as difference between the mean and the mean of the mean. Is there any difference between the results using the original method, in which monthly average has just one value at each grid point for each month, and that with the authors' method? If the difference is about 10%, which is the order of magnitude of r, then which method is better?

P6L20 "First, the above reference mean SST field. . ." This definition of the filament, does not include any shape criteria, and allows the region between straight upwelling front and the coast to be detected as a filament. Is it okay?

P7L21 "representation of turbulence" should need "geostrophic" after "of". The term "turbulence" throughout the manuscript sounds not adequate as it includes the meaning of three-dimensional microscale turbulence. The scale of the "turbulence" should be specified, or replace "turbulence" by "eddy"

P8L1 "The AVISO datasets. . .". The 1/4 degree is the resolution of the grid for mapping, the actual AVISO sea surface data's resolution should be even coarser.

P11L2 "small eddies are abundant found in the. . ." This should be "small eddies are abundantly found in the. . ." or "small eddies are abundant, and found in the. . ."

Figure 6c mean vertical flux Why is the mean vertical flux directed upward in a fairly

wide region in the southern part, and vice versa for the norther part? Is it vertical flux of omega flux, in which horizontal flux can contribute through the sloping grid bottom faces?

Figure 7e What is the green color?

P16L4 "Switching to the vertical fluxes, the differences between the mean and turbulent flux components are even more pronounced" and Figure 8c. I'm puzzled by the mean vertical carbon flux. The bands of down-up-downwelling flux indicates the increase of carbon at 20 m and the decrease at 80 m depth on average, unless there is compensating lateral flux divergence. I don't understand how the Ekman pumping results in this structure.

P16L30 "This signature is typical of that expected from an eddy-induced vertical displacement of the nutricline (McGillicuddy, 2016). " If this is the case, the eddy is linear Rossby waves, and not trapping to carry carbon when it propagates. I think eddy propagation speed c is a few centimeter per second, and velocity u > 0.2 m/s, so u/c>1, suggesting that eddy is nonlinear, which can hold a large volume of water inside to travel together westward, which is the dominant signal discussed in this manuscript, rather than wave induced isopycnal displacement.

P17L7 "In the offshore waters, ACE can also..." Is this mechanism same as the warm core eddy spin down, accompanied by weak upwelling at the center (e.g. Frictionally induced circulations and spin down of a warm‐core ring by Glenn R. Flierl and Richard P. Mied)? Do you have any evidence to support that this is the cause of positive C'org within ACE?

P22L3 "Between 100 km and 500 km from ..." and Figure 15d The structure based estimate mean flux divergence between 100-500km is negative. Does it mean that the mean flux is removing the carbon from the shore? If so this makes sense, because Ekman transport from the coast should act to remove carbon from the shore. But the distance is somewhat too far from the coast. Why does it start from 100 km not 0 km?

Also, in Figure 15, why does the Reynolds decomposition based mean flux divergence in 100-500 km show different results from the structure based results?

P22L12 "...offshore reach due to the their..." remove "the".

P22L19 "...mean deflection of their trajectories (Chelton et al., 2011)" The earlier study should be included "McWilliams,ÂăJ., andÂăG.ÂăFlierl,Âă1979:ÂăOn the evolution of isolated, nonlinear vortices.ÂăJ. Phys. Oceanogr.,Âă9,Âă1155–1182."

P22L28 "In particular, ACE are responsible for the northward ... due to their relatively fast decay that results in a slowing down of the clockwise rotation while they move offshore, these eddies induce a net northward transport of Corg" I don't understand this explanation for the net northward carbon transport by ACE. Considering the positive carbon gradient in zonal direction with clockwise stirring, the southward transport should be stronger. How is this result related to Reynolds decomposed eddy meridional flux in Figure 6e?

P22L31 "In the vertical direction..." I don't understand why NE-NF vertical flux is the stronger than eddy components. Also do these vertical carbon fluxes reflect the lateral structure of the vertical flow itself or carbon distribution? For confirmation, the total mean vertical flow at 100 m depth should be presented in appendix. The mean vertical flow should be largest at $\sim O(10\text{-}5\ ms\text{-}1)$ along the coast with the internal Rossby radius as the lateral width, which is very narrow compared to the model domain.

P23L4 The convergence at filament is revisited with the theory of frontogenesis by "McWilliams et al. 2009, Cold filamentary intensification and oceanic surface convergence lines"

P25L15 "Nagai et al. (2015) showed that this subduction happens primarily at the filament tip..." Remove "tip" and insert "periphery of" before filament

---

## Referee Comment (RC2) · Anonymous Referee #2 · 17 Apr 2018

**Review of manuscript #BG-2018-54 entitled "Mesoscale contribution to the long-range offshore transport of organic carbon from the Canary Upwelling System to the open North Atlantic" by E. Lovecchio et al.**

**General comments:**

The authors of the above manuscript used regional modelling to study the mesoscale contribution of the long-range cross shore transport from the eutrophic upwelling to the adjacent oligotrophic gyre, focussing on organic carbon in the Canary Upwelling System. They used two different, yet complementary, methodologies to do so: the classical "turbulence-based" method (Reynolds decomposition) and the more recently developed "structure-based" technics (e.g. eddy- and filament-tracking algorithms). Based on the Reynolds decomposition, they analysed both the mean and turbulent fluxes in several subsystems and in several offshore bands; then they compared these to the estimates derived from the "structure identification" methods. Interesting correspondences emerged and they concluded with the importance of the mesoscale transport in the lateral export of C-org from the coastal upwelling to the open ocean. Integrating the results from both methodologies, they further showed the prominence of filaments and eddies (acting over different regions and at different levels) as a source of carbon and nutrients for the offshore oligotrophic waters.

The topic is highly relevant and the paper is clear in its objectives and its presentation (well written and well referenced). It contains many interesting results that have been thoroughly discussed, and promising conclusions; overall, I liked the paper and I would recommend publication providing that my specific comments are addressed. The suggestions below should help improving the article and the figures, polishing the text and correcting remaining typos.

In addition, my only real concerns with this paper is its length. This contribution is significantly long (I acknowledge it contains many interesting results but some are a bit redundant) and the writing is slightly too didactic sometimes (it could often be more "to-the-point" and more condensed). For instance, it seems weird to have both "Summary and Synthesis" and "Conclusions" sections in the same article (although I enjoy the reading of both). I agree it has a lot of content (complex analyses & interesting well-supported results) but it may be "lost-at-sea" considering that, nowadays (and unfortunately), papers are more-and-more numerous and less-and-less read by the community. Their work could gain in visibility if they opt for shorter and/or separated contributions. I would suggest the authors to try reducing the length of the article (i) by focussing their analyses and the writing and (ii) maybe by keeping aside some content for another paper (especially the regional analyses over the three subsystems -keeping here only the whole CanUS- and the seasonality of the processes, cf. a comment below).

Overall, I recommend the publication of this manuscript in BG providing the authors revise their manuscript to address the points raised here.

**Specific comments:**

- Typing/English mistakes: p 1 line 19: "anticyclonic" has been perhaps forgotten; p 5 line11: delete "the"; p 6 line 19: filaments; p 9 line 4: delete "eddy"; p 11 line 2: delete "abundant" or "found"; p 11 line 25: delete "a"; p 16 line 11: could it be "going deeper than 500 m within 500 km from the shore (or over 0-500 km offshore)"; p 17 line 8: "which slow down the rotation…"; p 18 line 14: should it refer to Fig. B7 and/or 8 instead of Fig. B6 there?; p21 line 1: estimateS; p 22 line 33: responsible OF a; p 30 line 18: cyclonEs; etc..

- Page 2 lines 9-10: there exist plenty of earlier references to support this statement, including some that are already cited, such as Rossi et al. 2008 and Gruber et al. 2011.

- Page 2 lines 31-32 and page 30 lines 32-33 + page 31 lines 1-2: the "coherence" of eddies is still an open question to my mind; there is a bunch of papers around those questions, both from the observational and modelling point of views, and many situations (from very coherent to quite leaky) have been reported; so it seems no agreement has been reached yet. Please rewrite.

- Page 4 line 11: in addition of in-situ observations and modelling studies, there also exist purely satellite studies (e.g. Capet et al. 2014, not yet cited) as well as merged modelling & satellite study (e.g. Hernandez-Carrasco et al. 2014, already cited). Capet et al. 2014. Implications of Refined Altimetry on Estimates of Mesoscale Activity and Eddy-Driven Offshore Transport in the Eastern Boundary Upwelling Systems, Geophysical Research Letters, 41 (21), 7602–7610, doi:10.1002/2014GL061770. It could also be cited at page 6 line 27-28 (assumption of "prismatic" volume) and page 30, line 25.

- Page 5 lines 12-15: the turbulent field (e.g. mesoscale activity) exhibits temporal variability, at monthly and especially seasonal time-scales. Why did you include the monthly/seasonal variability of the fields into the mean fluxes? I presume that this choice could impact your results about the "turbulence-based" methods, as well as when comparing it against the "structure-based" approach (has it been done similarly?). These additional analyses could be kept for future work anyway.

- Page 6 lines 18-20: Why is the reference mean SST field coarsened onto a 2x2°grid? This coarsening procedure is not applied to the snapshots, right? What could be the impact of a different coarsening? How was the threshold of -0,3°C defined? What is the sensitivity if your results to these threshold?

- Figures 3 and 4: for ease of understanding, please report in the caption that the unit of some panels is "number of eddies per 1 degree bin per day" (if I understood correctly).

- According to their specific behaviours showed in Fig 7 inserts, it would be interesting to see the equivalent of Fig. 8 but for the different subsystems (to be kept for future work I suppose).

- Page 18 line 5-7: Were these velocities estimates taken from the Hovmöller plots of Fig. 10 or from a direct check of exemplary velocities modelled at the boundaries of those structures? If these propagation speeds (for each structure type) were derived from the Hovmöller of Fig. 10, are they in accord with those derived from Fig. B6?

- Fig. 12: inconsistency between the caption and the plots displayed. I am also wondering if the explanations described lines 1-14 p 20 were obtained after having analysed the simulated fields of a few typical structures (if so, please show some figures in Appendix)? If not, I presume those lines of text are rather "discussion", so that they should have more references and they could be transferred somewhere else.

- Page 21 line 10: I found that the expression "coastally confined" is a bit misleading to describe a process prominent over a region extending from the shores to more than 500 km offshore; maybe "shelf-confined"? (although the shelf might be even narrower than this…), else?

- Page 25 lines 7-11 and page 26 line 4: to further support those arguments, you could refer also to Rossi et al. 2013 (already cited) who documented (from in-situ observations in the Iberian Peninsula Upwelling) a filament of 50-60 m depth which experienced subduction when moving offshore (see their sect. 3.2.2 and 3.6).

- These "intermittent hotspots of downwelling" are puzzling; I wonder what mechanisms are involved and why are they so transitory?

- Page 26 line 10: compilations of offshore transport due to upwelling filaments were also reported in Sanchez et al. 2008.Physical description of an upwelling filament west of Cape St

Vincent in late October 2004. Journal of Geophysical Research—Oceans, vol.113, C07044, http://dx.doi.org/10.1029/2007JC004430.

- Page 29 lines 3-9: why maps of chlorophyll are mentioned here? I thought that your structure detection algorithms were indeed applied to modelled SST maps only, is this right? Or did you check also consistency with chloro maps? Please clarify.

---

## Referee Comment (RC3) · Anonymous Referee #3 · 17 Apr 2018

GENERAL STATEMENT

This is a numerical modelling study of the relative contribution of mesoscale structures such as upwelling filaments and eddies to the export of biogenic organic carbon from the NW Africa EBUE to the adjacent ocean. The work is very opportune in the current context of revisiting the biological pump to assess the importance of the horizontal fluxes of dissolved and suspended organic matter in ocean margins in general, and EBUEs in particular. The paper is solid, showing the limitations of the ROMS + NPZD coupled models honestly and applying appropriate methodologies (Reynolds flux decomposition and structure-identification algorithms) to separate the transport from fila-

ments and eddies. It is well written and understandable for scientist interested in this issue that are not modellers, as it is my case. The results are relevant, confirming the major role of filaments in the first few hundred kilometers off the coast and the rising importance of eddies further offshore. Therefore, I do not have any major concern about the manuscript.

However, I would also like to introduce a philosophical issue. The NPZD has difficulties to reproduce properly the C cycle in coastal upwelling zones for two reasons: ii) DOC is not properly included in the model; and ii) benthic nutrient mineralisation is not considered.

For the case of DOC, the authors have found a way to overcome the problem by introducing two POM pools (sinking and suspended) and considering DOC part of the suspended POC pool. It could work. In this regard, lateral Corg fluxes are reported without differentiating the pools considered (Phyto, Zoo, large POM, small POM + DOM). Are you reporting only small POM + DOM or also the other pools?

For the case of benthic mineralization, in EBUEs with wide continental shelves, as it the case of the NW Africa EBUE, 40-60% of the nutrients used for phytoplankton growth could come from the underlying sediments. Is this process included in the NPZD model in any way?

SPECIFIC COMMENTS & MINOR DETAILS

Page 1, line 4 – I would not say that a model will be able to "demonstrate quantitatively" that eddies and filaments are exporting organic matter from the coast to the open ocean at the NW Africa EBUE scale. A numerical model is a tool that tries to approach as much as possible to reality, but it is (rather) imperfect.

Page 5, caption of Figure 1 – explain what is "01/12/0030" and change "9.5°N and 32°N" by "32°N and 9.5°N".

Page 7, line 13 – salinity is unit less

Page 9, line 1 – It is a bit optimistic to state that the patterns of the model and the observations are "very similar". I would say just "similar" or even "roughly similar".

Page 11, line 2 – Please, erase "found"

Page 11, line 18 – Please, indicate what organic matter pools are included in Corg (Phyto?, Zoo?, large POM?, small POM + DOM?).

Page 12, Figure 5 – Why Corg was integrated through the entire water column instead of just in the upper 100 m. No scale for current vectors has been added in panel (a).

Page 20, Figure 12 – The footnote is not coherent with the figure.

─────────────────────────────

---

## Author Comment (AC1) · 4 Jun 2018

**Answer to Referee #1**

We thank Referee #1 for the time spent on reviewing our manuscript and for his/her thoughtful comments on the pattern of the carbon fluxes and on the eddy influence on the organic carbon anomalies and fluxes, which have helped us to improve our manuscript. In order to address these comments, we have carried out further analysis with the aim to clarify some questions. We include below our detailed answers to the three major concerns raised by the Referee and to all the specific comments and describe the proposed changes to the manuscript.

**Answers to Major comments**

**MC1) Pattern of the meridional and vertical organic carbon fluxes**

*"One of these issues is that the mean vertical carbon flux presented in Figure 6c and 8c. There seems to be clear contrast between northern and southern regions separated at C. Blanc in plan view of the vertical carbon flux, which looks somewhat similar to the distribution of the mean meridional flux in Figure 6b. Also, in the vertical section, there is an upward flux at 60-70 m depths, right under and above the downward carbon fluxes. I'm puzzled by these mean vertical and meridional carbon flux distributions, and consider that this needs to be clarified."*

**Answer to MC1**

As stated by Referee #1 the pattern of the mean meridional and vertical organic carbon fluxes (Figure 6 of the manuscript) differs strongly between the regions north and south of the Cape Verde front, according to the subregional differences in the circulation of the Canary Upwelling System, which include a large-scale anticyclonic circulation north of the front and a cyclonic circulation to the south (Arístegui et al. 2009; Pelegrí and Benazzouz, 2015; Pelegrí and Peña-Izquierdo 2015). The mean signature of the wind stress curl (negative north of the front, positive south of the front) also induces a change of sign in the mean vertical fluxes at the front, as in Figure 6c (see also Lovecchio et al. 2017, Figure 11, and the in-depth discussion of the subregional differences in the $C_{org}$ fluxes). In the present paper, we quickly refer to these differences in our introduction (page 3, lines 9-13). In order to clarify the drivers of the mean flux pattern, we will modify the first sentence of section 4.2 as follows:

"The Reynolds decomposition of the advective fluxes of $C_{org}$ in the top 100 m shows that the time-mean fluxes (Figure 6a-c) clearly agree with the known pattern of regional circulation and wind stress curl characterizing the CanUS (Arístegui et al. 2009, Pelegrí and Peña-Izquierdo 2015, Lovecchio et al. 2017). While these mean fluxes dominate the total fluxes (see also: Lovecchio et al. (2017), Figure 11, Total organic carbon fluxes), the turbulent fluxes contribute substantially to the total fluxes and to the lateral and vertical contributions to the flux divergence (Figure 6d-f)."

Also the alongshore average section of the vertical advective flux (Figure 8c of the manuscript) reflects the subregional differences described above, showing a composition of positive (southern CanUS) and negative (northern CanUS) mean vertical advective fluxes of organic carbon. We refer to this in page 16, line 5-6 of the manuscript, as: "The different signature of the wind stress curl and consequent Ekman pumping offshore in the northern and southern CanUS results, on average in the CanUS, in a mixed signature of the vertical transport in the open waters.". In order to better illustrate these differences we include in the present document the profile of the vertical advective flux calculated separately for the northern, central and southern CanUS (Figure 1). Furthermore we propose to modify the sentence mentioned above to:

"Offshore, the different sign of the wind stress curl (negative and positive, respectively) that characterizes the latitudes located north and south of the Cape Verde front results in an opposite sign of the time-mean vertical flux of organic carbon in the two zonal bands. These opposite fluxes, once averaged along the entire CanUS, give rise to an alternate pattern of positive and negative signatures in the profile of the mean vertical transport in the open waters."

[Figure]

*Figure 1:* Alongshore average sections of the vertical advective fluxes of organic carbon for the three CanUS subregions, as defined in the manuscript. The change of signature in the vertical flux offshore is in agreement with the change of sign of the wind stress curl in correspondence of the Cape Verde front.

**MC2) Analysis and discussion of the results of Reynolds decomposition and covariance**
*"In some part, although the authors tried to explain the resultant interesting patterns obtained by the Reynolds decomposition and structure based extraction of eddy and filament contributions, they are often stated without sufficient evidence and would make readers feel that they are rather speculative. For example, the high covariance between anticyclonic eddies and carbon anomaly in northern offshore, is explained by the upwelling induced by the anticyclonic eddy spin down, without any analysis."*

**Answer to MC2**
In order to address this comment we decided to make some changes in the Results section, keeping in mind also the suggestion of Referee #2 to shorten the manuscript where possible. We therefore propose the following changes to the manuscript:
- We agree with Referee #1 that there isn't enough evidence supporting the upwelling in ACE in the final stages of their life in the proximity of the Canary Archipelago (discussion of Figure 9). Animations of the model output suggest that this positive $C_{org}$ anomaly in ACE is rather associated with the extremely sharp offshore gradient in the average $C_{org}$ concentrations at these latitudes. This gradient, resulting in very low $C_{org}$ concentrations in the open waters, makes it possible for ACE formed near the coast to still contain on average more carbon than the open waters. Moreover, some of the ACE that cross these latitudes interact with filaments generated south of the Archipelago, which are particularly rich in $C_{org}$ due to the north-south positive gradient in the tracer concentrations. We therefore plan to revise the discussion of Figure 9 according to this analysis. We propose to change the paragraph on page 17, lines 1-9 as follows:
"Local deviations from the widespread negative correlation of SSH' and $C_{org}$' may also be explained in terms of eddy anomalies. The positive $C_{org}$' for ACE seen in the nearshore southern CanUS is likely a consequence of the northward flowing Mauritanian Current, which favors the trapping of carbon-rich coastal waters in ACE at the time of their formation. However, this signature is not far reaching. The signature of $C_{org}$' within eddies at the northern CanUS boundary is connected to the presence of incoming large eddies formed in the Azores current (Gaube et al., 2014, Figure 1), leading to ACE with positive $C_{org}$' and CE with negative $C_{org}$'. Also in the offshore waters of the northern CanUS, ACE have a positive $C_{org}$' signature. Animations of the model output suggest that this is possibly due to the sharp offshore gradient in $C_{org}$ at these latitudes, where coastally generated ACE still have a positive impact on the organic carbon budget of the carbon depleted open waters."
- To further reduce the speculative elements in the discussion of our results, we propose also to shorten section 4.4, limiting it to the discussion to the contribution of mesoscale activity to the organic carbon stock, while moving Figure 12 to the Appendix. This change to the manuscript is also listed in the answer to the general comment of Referee #2. The discussion of the nutrients

concentrations and production in the eddies (page 20, lines 1-14) will be limited to the following sentence:

"Moreover, the two types of eddies are characterized by a strong asymmetry in the nutrient concentrations and in the production rates (see also Appendix: and Figure B7), which likely are exacerbating the differences in $C_{org}$ between long living CE and ACE. The combination of relevant differences in both organic carbon availability and typical velocities of the three types of mesoscale structures determine the magnitude of their relative contribution to the total $C_{org}$ fluxes. "

**MC3) Manuscript structure: merging Summary and Synthesis with Conclusions**
*"To be concise, section 6 and 7 could be combined into one section."*

**Answer to MC3**
We plan to merge the two sections "Summary and synthesis" and "Conclusions" into a single "Conclusions" section (complying with the Biogeosciences "Manuscript preparation" guidelines). We will shorten the portion of text belonging to the current "Summary and synthesis" as follows, while leaving the final conclusions part largely unchanged:

"Our Reynolds flux decomposition shows that the turbulent component of the zonal flux of $C_{org}$ out of the coastal CanUS contributes, on average, from 5 % to above 30 % to the total zonal transport and 30 % to the total zonal flux divergence, extending out to 2000 km from the coast.  The turbulent zonal transport is mostly confined to the first 100 m of depth, but it shows a subsurface intensification, owing to a strong turbulent vertical downwelling. The contribution of the turbulent zonal flux to the total flux and to its divergence is particularly important in the northern and southern CanUS, while in the central CanUS its contribution pales in comparison to the already intense mean offshore transport. With the use of eddy and filaments masks we separate the contribution of mesoscale eddies and filaments to the availability and transport of $C_{org}$. Filaments are characterized by inner flow velocities that reach up to 0.5 m s$^{-1}$ and contain most of the total $C_{org}$ up to 200 km from the coast, but their $C_{org}$ share declines quickly offshore. Eddies, on the contrary, drift with speed of a few cm s$^{-1}$ but see a sharp increase of their $C_{org}$ content in the first 200 km from the coast and contain beyond this distance about 30 % of the total available $C_{org}$, two thirds of which is found in CE.  Thanks to their high advective speeds, large $C_{org}$ concentration and semi-permanent character, filaments dominate the nearshore zonal transport accounting for nearly 80 % of the total flux at 100 km offshore. The filament transport captures a large part of the mean Reynolds offshore transport near the coast, while vertically it is responsible for a strong turbulent vertical downwelling. The divergence of the filament offshore transport adds the majority of the extra $C_{org}$ to the first few hundreds km offshore, but the divergence drops off quickly, reaching zero at 1000 km. The eddy lateral transport, on the contrary, resembles in pattern and intensity the turbulent Reynolds offshore flux and accounts for about 20 % of the total zonal flux and total flux divergence. Eddies, which move slowly but contain about 30% of the $C_{org}$ available offshore, add $C_{org}$ to the offshore waters up to 2000 km from the coast; in particular CE are responsible for most of the offshore transport largely because of their elevated $C_{org}$ concentration and longer lifetimes."

**Answers to specific comments**

**P2L13** *"Coastal filaments are narrow (<50 km wide) structures that extend from the coast to several hundred kilometers into the open sea with rather large velocities (between 0.25 and 0.5 m s$^{-1}$) " Is this velocity in "with rather large velocities (between 0.25 and 0.5 m s$^{-1}$)", propagation speed or current velocity?*
**Answer:** This velocity refers to the current velocity found inside of the developed filaments. We clarified this modifying the sentence as follows: "Coastal filaments are narrow (<50 km wide)

structures that extend from the coast to several hundred kilometers into the open sea and are characterized, in their interior, by rather large flow velocities (between 0.25 and 0.5 m s$^{-1}$).”

**P2L15** *“Offshore transport by filaments is typically accompanied by intense subduction, due to the high density of the cold upwelled waters” The subduction is intense in the cold filaments because frontogenesis at a cold filament generate two cell ageostrophic cross-frontal circulations, which work together to restratify the two fronts at the edge of the filament, and to narrow the width of the filament, producing the strong downwelling in the filament. This sentence seems not accurate, and I suggest that it should be modified with a reference of McWilliams et al. 2009, Cold filamentary intensification and oceanic surface convergence lines”.*
**Answer:** We agree that this sentence is not clear. We will modify it to read as follows:
“Transport by filaments is typically accompanied by a deepening of the tracer fluxes in the offshore direction, due to the higher density of the upwelled water (Cravo et al., 2010). It is also associated with intense downwelling by the generation of ageostrophic cross-frontal circulation at the edges of the filament (McWilliams et al. 2009, Nagai et al. 2015).”

**P2L26** *“These non-linear structures propagate with velocities of a few centimeters per hour, about one order of magnitude slower than the filaments . . .” “a few centimeters per hour” is 10 μms-1, and should be too slow for the mesoscale eddy propagation speed.*
**Answer:** Thank you. We corrected this typo with “a few centimeters per second”

**P4L17** *“CanUS also showed that eddies tend to reduce coastal production through the lateral export of the upwelled nutrients ” I think that the reduction is caused by both lateral and vertical nutrient transport.*
**Answer:** Thank you, we will modify the sentence as follows:
“...CanUS also showed that eddies tend to reduce coastal production through the lateral export and subduction of the upwelled nutrients”

**P5L8** *“F = $\overline{F}$ + F', were ” Typo. “were” should be “where”.*
**Answer:** Thank you, we will correct this typo.

**P6L4** *“The last term r =  + <$\overline{u}$ $C_{\text{org}}$'> term represents the sum of the residuals which we verified to be small, at least one order of magnitude or more smaller than the other terms” The original rule of the Reynolds decomposition is that the mean of the fluctuation is zero, and the mean of the mean is the same mean, but the authors'choice allows non-zero values for the former, as we as difference between the mean and the mean of the mean.*
*Is there any difference between the results using the original method, in which monthly average has just one value at each grid point for each month, and that with the authors' method? If the difference is about 10%, which is the order of magnitude of r, then which method is better?*
**Answer:** As Referee #1 states, our choice of the reference mean (climatological monthly means interpolated to 2-day steps) results in non-zero residuals in the long-term average fluxes obtained from the Reynolds decomposition. The Referee asks what would happen with the use of simple non-interpolated monthly averages, therefore an average that has a constant value for each month. As far as there is a time varying component in the reference mean (interpolated in time or not), the residuals of the decomposition will not be zero, as the long-term averaging of the components and the reference mean are calculated differently. The suggested non-interpolated monthly mean would give zero residuals only if we calculated the flux components separately for each month, obtaining 12 couples of plots for mean and turbulent fluxes, each with zero residuals. However, this level of detail goes beyond the purpose of the present study. Instead, the use of non-interpolated monthly means would result in large jumps in our time evolution of the turbulent deviations, with unrealistic discontinuities in the moment of transition between one month and the following one. Furthermore,

the anomalies would have, on average, minimum values in the middle of the month and large values at the beginning and at the end of each month. We therefore consider our choice of reference mean a better choice than the use of non-interpolated monthly means. As stated in the manuscript (page 6, lines 5-6) the residuals of the Reynolds decomposition obtained with our choice of reference mean are at least one order of magnitude smaller (<10%) than the smaller component between mean and turbulent fluxes in each direction of flow. In order to better document our results, we include the plots of the residual fluxes below (Figure 2).

To clarify this point, we modify the above-mentioned sentence as follows:

"The last term r = $\langle u' \overline{C_{org}} \rangle + \langle \overline{u} \, C_{org}' \rangle$ term represents the sum of the residuals which arises from the use of a time-varying reference mean and which we verified to be small, at least one order of magnitude or more smaller than the other terms"

[Figure]

*Figure 2: Residual fluxes of the Reynolds decomposition obtained using the time variable reference mean adopted in the manuscript, plotted on the same scales as the turbulent fluxes in each direction of the flow.*

**P6L20** *"First, the above reference mean SST field. . ." This definition of the filament, does not include any shape criteria, and allows the region between straight upwelling front and the coast to be detected as a filament. Is it okay?*

**Answer:** With the present question Referee #1 expresses two concerns regarding the filament mask: (a) the fact that the mask covers the nearshore upwelling band at most time steps, and (b) the lack of an explicit shape criteria for the filament identification. We address them separately.

(a) As stated, the filament mask often covers the nearshore upwelling band roughly corresponding to the first 50 km offshore. However, this does not affect our main results and conclusions significantly. In fact, the filament flux divergence in the nearshore is calculated taking differences between the flux at 100 km offshore (well defined filaments) and the flux at the coast (zero). In the discussion of the filament contribution to the tracer concentrations we avoid to show the filament tracer stock in the first 50 km (inserts of Figures 11, 12). We also do not refer to the nearshore fluxes (0-100 km offshore range) in our summary figure (Figure 15). For the remaining figures, we state that this potential limitation must be kept in mind (see also the discussion at p 21 lines 5-9 of the present manuscript). Operatively, it is actually difficult (if not impossible) to unequivocally identify the base of a filament, distinguishing it from the nearshore flow that feeds into the structure. Possible alternatives would be either to remove the first 50 km of the filament mask and attribute the flux to the NF-NE transport (resulting in discontinuous pattern in the fluxes), or somehow split this portion of the flux between filaments and NF-NE transport. We consider these two methods more arbitrary than attributing the nearshore flux to the filament mask, while acknowledging the limitation in the manuscript.

(b) As regard to constraining the shape of the filaments, in the course of the analysis the filament mask was evaluated on the basis of both SST and surface Chlorophyll images (see Figure B1 for an

example) with very satisfactory results and an extremely limited number of cases of overestimation of the filaments either in time or in space. Of the few cases identified, they were all confined to the southern CanUS latitudes. The coarsening of the reference-mean field before calculating the temperature anomalies and the choice of an appropriate threshold in temperature anomaly (see Methods, page 6, lines 18-25) assure in fact the identification of narrow cold structures.

In order to provide a comprehensive description of the algorithm and of its sensitivity to the parameters, we propose to add a **supplement to the manuscript**, which includes a technical description of the algorithm and of its properties, and an evaluation of the algorithm performance.

**P7L21** *"representation of turbulence" should need "geostrophic" after "of". The term "turbulence" throughout the manuscript sounds not adequate as it includes the mean-ing of three-dimensional microscale turbulence. The scale of the "turbulence" should be specified, or replace "turbulence" by "eddy"*
**Answer:** We will add "geostrophic" before "turbulence".

**P8L1** *"The AVISO datasets. . .". The 1/4 degree is the resolution of the grid for mapping, the actual AVISO sea surface data's resolution should be even coarser.*
**Answer:** Thank you for this comment, we will rephrase as follows:
"The AVISO dataset grid, in fact, has a resolution of 1/4º ..."

**P11L2** *"small eddies are abundant found in the. . ." This should be "small eddies are abundantly found in the. . ." or "small eddies are abundant, and found in the. . ."*
**Answer:** Thank you, we will correct this typo.

**Figure 6c** *mean vertical flux Why is the mean vertical flux directed upward in a fairly wide region in the southern part, and vice versa for the northern part? Is it vertical flux of omega flux, in which horizontal flux can contribute through the sloping grid bottom faces?*
**Answer:** The mean vertical flux is directed according to the regional signature of the wind stress curl, which is negative north of the Cape Verde front and positive south of the front (see also answer to MC1 for a further description of the flux pattern in the region). This results in a mean downward and upward flux respectively north and south of the front. In the nearshore the effect of the mean upwelling becomes evident, resulting in an intense upward flux in the nearshore northern and central CanUS. In the southern CanUS, where the upwelling intensity is on average weaker and the slope is especially wide (Arístegui et al. 2009), downwelling cells are also visible in the nearshore. The flux is not calculated with the omega velocity, but with the purely vertical component of the velocity. In order to further clarify this point, we propose to add in the Methods section, following the description of the Reynolds decomposition:
"The vertical flux components were calculated using the purely vertical velocity from the model output"

**Figure 7e** *What is the green color?*
**Answer**
We propose to add the following to the figure caption:
"The green area shading results from the overlapping of opposing mean and turbulent flux contributions."

**P16L4** *"Switching to the vertical fluxes, the differences between the mean and turbulent flux components are even more pronounced" and Figure 8c. I'm puzzled by the mean vertical carbon flux. The bands of down-up-downwelling flux indicates the increase of carbon at 20 m and the decrease at 80 m depth on average, unless there is compen-sating lateral flux divergence. I don't understand how the Ekman pumping results in*

*this structure.*

**Answer:** Figure 8c shows the mean vertical component of the flux, averaged in the entire CanUS. As shown in Figure 6, the sign of the vertical component changes between northern and southern subregion. Therefore the mean plot shows a combination of downward organic carbon flux in the north and upward flux in the south. We refer to our answer to MC1 for a further explanation of the flux pattern, and for our proposed changes to the manuscript regarding this point.

**P16L30** *"This signature is typical of that expected from an eddy-induced vertical displacement of the nutricline (McGillicuddy, 2016). " If this is the case, the eddy is linear Rossby waves, and not trapping to carry carbon when it propagates. I think eddy propagation speed c is a few centimeter per second, and velocity u > 0.2 m/s, so u/c>1, suggesting that eddy is nonlinear, which can hold a large volume of water inside to travel together westward, which is the dominant signal discussed in this manuscript, rather than wave induced isopycnal displacement.*

**Answer:** We clarify this by modifying the sentence to read: "This signature is typical of that expected from an eddy-induced vertical displacement of the nutricline which occurs in the initial stage of eddy formation and intensification, possibly followed by the trapping of the tracer anomalies (McGillicuddy, 2016)."

**P17L7** *"In the offshore waters, ACE can also. . ." Is this mechanism same as the warm core eddy spin down, accompanied by weak upwelling at the center (e.g. Frictionally induced circulations and spin down of a warm core ring by Glenn R. Flierl and Richard P. Mied)? Do you have any evidence to support that this is the cause of positive C'org within ACE?*

**Answer:** We thank Referee #1 for highlighting this passage and we refer to the answer to MC2 for our proposed changes to the text.

**P22L3** *"Between 100 km and 500 km from . . ." and Figure 15d The structure based estimate mean flux divergence between 100-500km is negative. Does it mean that the mean flux is removing the carbon from the shore? If so this makes sense, because Ekman transport from the coast should act to remove carbon from the shore. But the distance is somewhat too far from the coast. Why does it start from 100 km not 0 km? Also, in Figure 15, why does the Reynolds decomposition based mean flux divergence in 100-500 km show different results from the structure based results?*

**Answer:** In Figure 15d the blue area does not represent the mean flux but the non-filament-non-eddy (NF-NE) flux, which is different from a mean transport (Reynolds decomposition, Figure 15b). The two fluxes must not be confused, as they are defined differently and represent different processes. The mean Reynolds component of the flux refers to a mathematical definition of mean flux, as defined in the Methods section, which can include the contribution of recurrent structures. The NF-NE component is defined as everything which is not identified by our mesoscale structure identification masks and therefore does not include (for example) recurrent filaments. But it may include some turbulent transport which is not identified as an eddy or a filament (e.g., variable small-scale fronts offshore). For this reason, the Reynolds-decomposition-based divergence of the mean flux and the structure-identification-based NF-NE flux divergence should not be the same, but the divergence of the components must nevertheless sum to 100% of the total divergence.
The plots in Figure 15 only refer to the offshore ranges of distances from 100km to 2000 km offshore, in which the divergence of the total offshore flux is positive and therefore receive organic carbon from the nearshore (see inserts of Figure 7a and Figure 14a, and the total offshore flux results previously presented in Lovecchio et al. 2017). Therefore, these plots have the aim of attributing the enhancement of the organic carbon availability offshore to the physical drivers. In the range of 0-100km offshore, instead, the divergence of the total offshore flux is negative,

removing organic carbon from the productive nearshore. We will clarify this point modifying the caption of Figure 15 as follows:

"Comparison between the results of the turbulence-based and structure-based methods for the entire EBUS in the range of 100km to 2000 km from the coast, where the divergence of the total offshore flux is positive, therefore resulting in an increase of the organic carbon availability (see also Lovecchio et al. 2017, Figure 10 for the divergence of the total offshore flux)."

The negative divergence of the NF-NE transport is determined by the fact that the non-mesoscale flux is weak at 100 km offshore, where most of the offshore transport is carried on by the filaments, while it intensifies moving towards 500 km from the coast.

We will clarify this point adding the following passage at the end of the first paragraph of section 4.5:

"The NF-NE flux continues to intensify also in the 100 km – 500 km range, therefore resulting in a negative contribution to the offshore flux divergence."

**P22L12** *". . .offshore reach due to the their. . ."* remove *"the"*.
**Answer:** Thank you for pointing out this typo, we will correct it.

**P22L19** *". . .mean deflection of their trajectories (Chelton et al., 2011)"* The earlier study should be included *"McWilliams and Flierl 1979: On the evolution of isolated, nonlinear vortices. Phys. Oceanogr., 1155–1182."*
**Answer:** Thank you for this suggestion, we will include this reference.

**P22L28** *"In particular, ACE are responsible for the northward . . . due to their relatively fast decay that results in a slowing down of the clockwise rotation while they move offshore, these eddies induce a net northward transport of Corg"* I don't understand this explanation for the net northward carbon transport by ACE. Considering the positive carbon gradient in zonal direction with clockwise stirring, the southward transport should be stronger. How is this result related to Reynolds decomposed eddy meridional flux in Figure 6e?
**Answer:** The regional gradient in the organic carbon concentration is not only negative in the offshore direction (zonal gradient) but also positive southward, with a reduced zonal gradient at lower latitudes (Aristeguí 2009, Demarq and Somoue 2015). ACE rotate clockwise and stir this gradient while moving offshore. In the nearshore, where the zonal gradient dominates, it is indeed possible to see a negative signature in the meridional ACE flux (Figure B9c). Away from the nearshore, where the meridional gradient and the ACE decay become relevant, the southern edge of the eddy stirs the organic carbon northward. The relatively fast decay of ACE also determines that in each point of the domain, on average, the western edge of a nearshore (therefore on average younger and more intense) ACE stirs the gradient northward, while the eastern edge of an offshore (older) ACE stirs it southward more slowly, resulting in a net northward meridional flux. In order to clarify this point, we will modify the sentence to read:

"In particular, ACE are responsible for the northward recirculation of the $C_{org}$ through the asymmetric stirring of the background regional gradient, which results from the combination of an offshore negative gradient (strongest in the nearshore) and a southward positive gradient in the $C_{org}$ concentration(Aristeguí 2009, Demarq and Somoue 2015). Due to their relatively fast decay and slowing down of the clockwise rotation while they move offshore, these eddies induce a net northward transport of the tracer. In fact, on average at any location, a younger and more energetic ACE stirs the $C_{org}$ northward, while an older and weaker ACE stirs it southward."

**P22L31** *"In the vertical direction. . ."* I don't understand why NE-NF vertical flux is the stronger than eddy components. Also do these vertical carbon fluxes reflect the lateral structure of the vertical flow itself or carbon distribution? For confirmation, the total mean vertical flow at 100 m depth should be presented in appendix. The mean vertical

*flow should be largest at ~O(10-5 ms-1) along the coast with the internal Rossby radius
as the lateral width, which is very narrow compared to the model domain.*

**Answer:** We thank Referee #1 for raising this interesting question, which points in the direction of our most recent analysis of the physical fluxes inside of the eddies, currently in development for a further dedicated publication. In order to address this question we therefore refer to our current understanding of the mesoscale eddy vertical fluxes according to our model results. Animations of the velocities inside of the eddies at each time step show dipolar or quadrupolar patterns in the vertical fluxes of well-formed stable eddies. These patterns average to a dipole in positive and negative vertical velocities in both composite cyclones and anticyclones at 100 m, in nice agreement with high-resolution observations of the mesoscale velocities in eddies (e.g., Barcelo-Llull et al., 2017, JPO). The pattern of positive and negative vertical velocities averages nearly to zero throughout the eddy surface, therefore constituting a minor net contribution to the advective fluxes in well-developed eddies. As a consequence, the net eddy contribution to the vertical advective flux is mostly driven by the large scale forcing, and in particular by the pattern of the wind stress curl (see answer to MC1 for a detailed explanation of the vertical flux pattern in the region). Submesoscale fluxes are not explicitly resolved by our model due to the limited resolution of our grid in the offshore waters. The total vertical advective flux at 100 m is published in our previous publication on Biogeosciences "On the long range offshore transport of organic carbon from the Canary Upwelling System to the open North Atlantic", and is visible in Figure 11c.
In order to clarify this point, we decided to modify lines page according to read:
"In the vertical direction, eddies have a minor role in the $C_{org}$ advection compared to the filaments, and their signature cannot be clearly distinguished from that of the NE-NF component. Inspection of the model output (not shown) reveals that mesoscale vertical velocities in well-developed eddies present a dipolar or quadrupolar pattern of upwelling and downwelling similar to that observed in in situ measurements (Barcelo-Llull et al., 2017), which results in a minor net contribution of these structures to the vertical flux."

**P23L4** *The convergence at filament is revisited with the theory of frontogenesis by
"McWilliams et al. 2009, Cold filamentary intensification and oceanic surface convergence lines"*
**Answer:** Thank you very much for this suggestion, we will add this reference in the manuscript.

**P25L15** *"Nagai et al. (2015) showed that this subduction happens primarily at the
filament tip. . ." Remove "tip" and insert "periphery of" before filament*
**Answer:** We will correct this sentence accordingly.

---

## Author Comment (AC2) · 4 Jun 2018

**Answer to Referee #2**

We thank Referee #2 for the time spent on reviewing our manuscript and for his/her thoughtful comments about the length of the manuscript, the timescales of the turbulent variability, the filament detection algorithm and for the many specific comments which have helped us to improve our manuscript. We include below our detailed answers to the general comment about the writing and length and to all the specific comments and describe the proposed changes to the manuscript.

**General comment**

**Length of the manuscript and writing**
*"...my only real concerns with this paper is its length. This contribution is significantly long (I acknowledge it contains many interesting results but some are a bit redundant) and the writing is slightly too didactic sometimes (it could often be more "to-the-point" and more condensed). For instance, it seems weird to have both "Summary and Synthesis" and "Conclusions" sections in the same article (although I enjoy the reading of both). I agree it has a lot of content (complex analyses & interesting well-supported results) but it may be "lost-at-sea" considering that, nowadays (and unfortunately), papers are more-and-more numerous and less-and-less read by the community. Their work could gain in visibility if they opt for shorter and/or separated contributions. I would suggest the authors to try reducing the length of the article (i) by focussing their analyses and the writing and (ii) maybe by keeping aside some content for another paper (especially the regional analyses over the three subsystems -keeping here only the whole CanUS- and the seasonality of the processes, cf. A comment below)."*

**Proposed solution**
We thank Referee #2 for his/her evaluation of the manuscript.  The length of the manuscript is indeed an issue and we will thrive to streamline and shorten it, to make it more accessible to the reader. In order to move in this direction, we have decided to combine the suggestions of Referee #1 (comments MC2, MC3, P17L7) and Referee #2 (this comment and specific comment SC6) and propose here a few changes to the manuscript. We list the changes below:

1 – We will shorten the Methods section by moving the (currently rather informal) description of the new filament detection algorithm to a dedicated **supplement to the manuscript**, in which we will also provide an evaluation of the algorithm sensitivity and of its performance.

2 – We will shorten the Evaluation section by moving Figure 4 to the Appendix, as it does not provide a comparison with the observations, and by removing a few repetitions in the discussion of the Figures.

3 – We will shorten the discussion of our results. In particular, we will merge subsection 4.3 and subsection 4.4 into a single subsection titled "From turbulent anomalies to the mesoscale contribution to the organic carbon stock" which will serve as a transition between the discussion of the Reynolds decomposition and the discussion of our structure-based analysis. This will allow us to present: velocities, $C_{org}$ anomalies and $C_{org}$ content of the mesoscale structures. With the purpose to ease the flow of the paper, we will invert the order of Figure 9 and Figure 10 and merging the first and last paragraph of the current subsection 4.3 into a single initial paragraph, followed by the discussion of the current Figure 9. We will cut part of the current subsection 4.4, limiting the discussion to the contribution of mesoscale activity to the organic carbon stock, in line with the focus of our paper. Figure 12 (mesoscale contribution to the nutrient stock and production) will be moved to the Appendix. This is also in line with the suggestion of Referee #1 to limit speculative discussion in the Results section.

4 – We will remove from our discussion the paragraph focusing on the processes regulating production and nutrient concentrations in the eddies according to previous literature (pg.27, ll.11-23), in order to maintain the focus on the organic carbon budget and transport.

5 - We will merge the two sections "Summary and synthesis" and "Conclusions" into a single "Conclusions" section (complying with the Biogeosciences "Manuscript preparation" guidelines), as also suggested by Referee #1. To this aim, we will shorten the portion belonging to the "Summary and synthesis" as follows, while leaving the final conclusions substantially unchanged: "Our Reynolds flux decomposition shows that the turbulent component of the zonal flux of $C_{org}$ out of the coastal CanUS contributes, on average, from 5 % to above 30 % to the total zonal transport and 30 % to the total zonal flux divergence, extending out to 2000 km from the coast. The turbulent zonal transport is mostly confined to the first 100 m of depth, but it shows a subsurface intensification, owing to a strong turbulent vertical downwelling. The contribution of the turbulent zonal flux to the total flux and to its divergence is particularly important in the northern and southern CanUS, while in the central CanUS its contribution pales in comparison to the already intense mean offshore transport. With the use of eddy and filaments masks we separate the contribution of mesoscale eddies and filaments to the availability and transport of $C_{org}$. Filaments are characterized by inner flow velocities that reach up to 0.5 m s$^{-1}$ and contain most of the total $C_{org}$ up to 200 km from the coast, but their $C_{org}$ share declines quickly offshore. Eddies, on the contrary, drift with speed of a few cm s$^{-1}$ but see a sharp increase of their $C_{org}$ content in the first 200 km from the coast and contain beyond this distance about 30 % of the total available $C_{org}$, two thirds of which is found in CE. Thanks to their high advective speeds, large $C_{org}$ concentration and semi-permanent character, filaments dominate the nearshore zonal transport accounting for nearly 80 % of the total flux at 100 km offshore. The filament transport captures a large part of the mean Reynolds offshore transport near the coast, while vertically it is responsible for a strong turbulent vertical downwelling. The divergence of the filament offshore transport adds the majority of the extra $C_{org}$ to the first few hundreds km offshore, but the divergence drops off quickly, reaching zero at 1000 km. The eddy lateral transport, on the contrary, resembles in pattern and intensity the turbulent Reynolds offshore flux and accounts for about 20 % of the total zonal flux and total flux divergence. Eddies, which move slowly but contain about 30% of the $C_{org}$ available offshore, add $C_{org}$ to the offshore waters up to 2000 km from the coast; in particular CE are responsible for most of the offshore transport largely because of their elevated $C_{org}$ concentration and longer lifetimes."
5 - We suggest to maintain the discussion of the subregional differences in the fluxes in the manuscript. These results provide essential information on the latitudinal differences in the flux contributions within the system, and can be of interest for studies that will focus on a specific portion of the CanUS. Moreover, they provide a term of comparison between our results and several previous studies which focused on a specific subregion.

Overall, the proposed changes will allow us to shorten the manuscript core (from Abstract to Conclusions) by roughly 10% of its length, i.e. from the initial 31 pages to 28 pages.

**Answers to specific comments**

**SC1)** *Typing/English mistakes:*
*1) p 1 line 19: "anticyclonic" has been perhaps forgotten;*
*2) p 5 line11: delete "the";*
*3) p 6 line 19: filaments;*
*4) p 9 line 4: delete "eddy";*
*5) p 11 line 2: delete "abundant" or "found";*
*6) p 11 line 25: delete "a";*
*7) p 16 line 11: could it be "going deeper than 500 m within 500 km from the shore (or over 0-500 km offshore)";*
*8) p 17 line 8: "which slow down the rotation...";*
*9) p 18 line 14: should it refer to Fig. B7 and/or 8 instead of Fig. B6 there?;*

*10) p21 line 1: estimateS;*
*11) p 22 line 33: responsible OF a;*
*12) p 30 line 18: cyclonEs; etc..*
**Answer:**
1) the sentence is actually complete as it refers to all types of eddies, and not anticyclonic eddies.
2) thank you, we will remove "the" as in: "and determined their turbulent values"
3) we will correct the sentence as in: "are upwelling filaments generated"
4) we will remove the typo and corrected as in: "the distribution of the modeled and observed large-eddy diameters"
5) we will correct this expression using: "abundantly found"
6) we will delete "a" as in: "Thin and short-lived filaments..."
7) we will use: " going deeper than 500 m within 500 km from the coast"
8) we decided to remove this sentence.
9) thank you for pointing this out, yes it should be the plot B8 of the current manuscript version, we will correct it accordingly.
10) we will correct this typo.
11) we will use: "are responsible for a strong"
12) thank you, we will correct the typo.

**SC2)** *Page 2 lines 9-10: there exist plenty of earlier references to support this statement, including some that are already cited, such as Rossi et al. 2008 and Gruber et al. 2011.*
**Answer:**
Thank you very much, we will add the references in this point.

**SC3)** *Page 2 lines 31-32 and page 30 lines 32-33 + page 31 lines 1-2: the "coherence" of eddies is still an open question to my mind; there is a bunch of papers around those questions, both from the observational and modelling point of views, and many situations (from very coherent to quite leaky) have been reported; so it seems no agreement has been reached yet. Please rewrite.*
**Answer:**
We agree with Referee #2 that significant progress still has to be made to understand the degree of isolation of mesoscale eddies from the surrounding environment.
We will rephrase the sentences as follows:
"Eddies trap water and tracers in their core during their formation. In stronger eddies, the degree of lateral isolation of the eddy core from the surrounding environment can be quite high, possibly resulting in the entrainment and long-range transport of trapped tracers at formation (Karstensen et al., 2015, 2017, Chelton et al.,2011)."
"Third, the relevant share of $C_{org}$ found in the offshore region within mesoscale eddies also tells us that a fraction of the offshore biological activity is fueled at the passage or death of such a structure."

**SC4)** *Page 4 line 11: in addition of in-situ observations and modelling studies, there also exist purely satellite studies (e.g. Capet et al. 2014, not yet cited) as well as merged modelling & satellite study (e.g. Hernandez-Carrasco et al. 2014, already cited). Capet et al. 2014. Implications of Refined Altimetry on Estimates of Mesoscale Activity and Eddy-Driven Offshore Transport in the Eastern Boundary Upwelling Systems, Geophysical Research Letters, 41 (21), 7602–7610, doi:10.1002/2014GL061770. It could also be cited at page 6 line 27-28 (assumption of "prismatic" volume) and page 30, line 25.*
**Answer:**
We thank Referee #2 for the suggestion and we will add these references in the text.

**SC5)** *Page 5 lines 12-15: the turbulent field (e.g. mesoscale activity) exhibits temporal variability, at monthly and especially seasonal time-scales. Why did you include the monthly/seasonal variability of the fields into the mean fluxes? I presume that this choice could impact your results about the "turbulence-based" methods, as well as when comparing it against the "structure-based" approach (has it been done similarly?). These additional analyses could be kept for future work anyway.*
**Answer:**
We agree with Referee #2 that the turbulent field exhibits a certain level of seasonal variability. However, seasonal changes in circulation and biological activity represent the dominant mode of temporal variability in the CanUS, both on the regional and on the subregional scale (Chavez and Messié, 2009, Mittelstaedt, 1991), i.e. on spatial scales that are larger than the mesoscale. For this reason, including the seasonal cycle into the turbulent field, for example using a climatological annual mean as a reference mean, would result in the turbulent deviations including strong signals that evolve on scales longer than the ones of interest for our study. In order to clarify this point, we will rephrase the sentence at page 5 lines 14-15 as follows:

"Throughout our analysis, the reference means are climatological monthly means of velocities and concentrations as calculated from the 24 years of the run used for the analysis, and interpolated them to bi-daily fields. This choice allows us to obtain smooth mean bi-daily fields while including both the dominant seasonal variability (Chavez and Messié, 2009) and the recurrent monthly oscillations of the large scale fields into the mean fluxes. In this sense, our turbulent field represent all those signals that vary locally on timescales shorter than the month."

**SC6)** *Page 6 lines 18-20: Why is the reference mean SST field coarsened onto a 2x2°grid? This coarsening procedure is not applied to the snapshots, right? What could be the impact of a different coarsening? How was the threshold of -0,3°C defined? What is the sensitivity if your results to these threshold?*
**Answer:**
The coarsening of the reference field onto a 2x2 degree grid allows the algorithm to better recognize the upwelling filaments as connected structures in the 2-day output SST field. This procedure is not applied to the bi-daily mean field. While developing the algorithm we tested the sensitivity of our results to the coarsening and to the SST threshold.
In order to provide a comprehensive description of the algorithm and of its sensitivity to the parameters, we propose to add a **supplement to the manuscript** which includes a technical description of the algorithm and its properties, as well as an evaluation of its performance.

**SC7)** *Figures 3 and 4: for ease of understanding, please report in the caption that the unit of some panels is "number of eddies per 1 degree bin per day" (if I understood correctly).*
**Answer:**
Thank you for your suggestion, we will add it in the caption.

**SC8)** *According to their specific behaviours showed in Fig 7 inserts, it would be interesting to see the equivalent of Fig. 8 but for the different subsystems (to be kept for future work I suppose).*
**Answer:**
Thank you for your suggestion, we include below the subregional plots of the zonal flux of organic carbon as a term of comparison with Figure 7 and Figure 8 of the manuscript.

[Figure]

**Figure re SC8:** *Reynolds mean and turbulent components of the zonal flux of $C_{org}$ averaged along lines of equal distance from the coast in the three CanUS subregions, as defined in the manuscript.*

**SC9)** *Page 18 line 5-7: Were these velocities estimates taken from the Hovmöller plots of Fig. 10 or from a direct check of exemplary velocities modelled at the boundaries of those structures? If these propagation speeds (for each structure type) were derived from the Hovmöller of Fig. 10, are they in accord with those derived from Fig. B6?*
**Answer:**
These velocities were calculated explicitly from the 2-day mean output. We plan to state this in the manuscript.

**SC10)** *Fig. 12: inconsistency between the caption and the plots displayed. I am also wondering if the explanations described lines 1-14 p 20 were obtained after having analysed the simulated fields of a few typical structures (if so, please show some figures in Appendix)? If not, I presume those lines of text are rather "discussion", so that they should have more references and they could be transferred somewhere else.*
**Answer:**
Thank you for pointing out this inconsistency, we will remove the reference to the subregions in this caption. Following this suggestion and the need to shorten the manuscript, we decided to shorten section 4.4, limiting the discussion to the contribution of mesoscale activity to the organic carbon stock, while moving Figure 12 to the Appendix. See our answer to the first comment (General comment) in this document for further details.

**SC11)** *Page 21 line 10: I found that the expression "coastally confined" is a bit misleading to describe a process prominent over a region extending from the shores to more than 500 km offshore; maybe "shelf-confined"? (although the shelf might be even narrower than this...), else?*
**Answer:**
Thank you for your suggestion, we will replace "shelf-confined" by "nearshore-confined".

**SC12)** *Page 25 lines 7-11 and page 26 line 4: to further support those arguments, you could refer also to Rossi et al. 2013 (already cited) who documented (from in-situ observations in the Iberian Peninsula Upwelling) a filament of 50-60 m depth which experienced subduction when moving offshore (see their sect. 3.2.2 and 3.6).*
**Answer:**

Thank you very much for this suggestion, we will add the reference in these points.

**SC13)** *These "intermittent hotspots of downwelling" are puzzling; I wonder what mechanisms are involved and why are they so transitory?*
**Answer:**
Animations of our model output show small scale and highly variable hotspots of downwelling in the filaments, mostly concentrated to the first 100 km of offshore range. These hotpots evolve quickly and often drift laterally following the filament flow. This corresponds to the previous findings of Nagai et al. 2015, and suggests us that these small scale fluxes are associated to the formation and degradation of frontal regions at the edges of the filaments, which are associated by high strain in the horizontal flow. The formation of such small scale downwelling is also favored by the relatively high resolution of our grid in the nearshore, ranging between 8 and 5 km of grid spacing. We will rephrase the mentioned passage as follows:
"Animations of the vertical velocities at 100 m depth in the filaments show transitory and irregular hotspots of strong downwelling, which evolve quickly and often drift laterally following the filament flow.The vertical transport inside filaments is therefore highly variable both due to horizontal movements of the downwelling cells and due to the intermittency of the process. With a numerical study, (Nagai et al. 2015) showed that this subduction happens primarily at the periphery of the filament due to the formation and degradation of frontal and high-strain regions at the filament edges, and it moves offshore during the filament formation, oscillating laterally with the structure."

**SC14)** *Page 26 line 10: compilations of offshore transport due to upwelling filaments were also reported in Sanchez et al. 2008.Physical description of an upwelling filament west of Cape StVincent in late October 2004. Journal of Geophysical Research—Oceans, vol.113, C07044, http://dx.doi.org/10.1029/2007JC004430.*
**Answer:**
Thank you very much for your suggestion, we will add this reference in the passage.

**SC15)** P*age 29 lines 3-9: why maps of chlorophyll are mentioned here? I thought that your structure*
*detection algorithms were indeed applied to modelled SST*
**Answer:**
The filament detection algorithm was indeed developed only on the basis of SST, which is the only field used by the algorithm. To evaluate the performance of the the algorithm, the masks were also compared to images of modeled surface chlorophyll. Surface chlorophyll constitutes a good tracer for the identification of the streams of upwelled water and the choice was made also in analogy to the traditional methods of single filament identification (by hand) from satellite images, often based on surface chlorophyll signatures. However, in order to avoid confusion we will remove this detail from the discussion section:
"Our newly-developed SST-based filament detection algorithm was tested on our grid with satisfactory results, with the large majority of the filaments being detected and very limited over-detection in the southern CanUS."
We will instead add the following sentence in the caption of Figure B1:
"The comparison of the detected mask with the S-CHL field, a good tracer for the identification of the streams of upwelled water, allows us to evaluate the filament-identification algorithm in analogy to the visual methods of single filament identification from satellite images."

Furthermore, we will include a dedicated **supplement to the manuscript** with the aim of providing a better description of the algorithm evaluation and of its performance.

---

## Author Comment (AC3) · 4 Jun 2018

**Answer to Referee #3**

We thank Referee #3 for the time spent on reviewing our manuscript and for his/her thoughtful comments on the representation of the organic carbon pool in our model. We include below our detailed answers to the Referee's major comment and to all the specific comments and describe the proposed changes to the manuscript.

**Major comment**

**Representation of the C cycle in NPZD**
*I would also like to introduce a philosophical issue. The NPZD has difficulties to reproduce properly the C cycle in coastal upwelling zones for two reasons: ii) DOC is not properly included in the model; and ii) benthic nutrient mineralisation is not considered.*
*For the case of DOC, the authors have found a way to overcome the problem by introducing two POM pools (sinking and suspended) and considering DOC part of the suspended POC pool. It could work. In this regard, lateral Corg fluxes are reported without differentiating the pools considered (Phyto, Zoo, large POM, small POM + DOM). Are you reporting only small POM + DOM or also the other pools?*
*For the case of benthic mineralization, in EBUEs with wide continental shelves, as it the case of the NW Africa EBUE, 40-60% of the nutrients used for phytoplankton growth could come from the underlying sediments. Is this process included in the NPZD model in any way?*

**Answer**
The organic carbon pool in the present NPZD model consists of four compartments, namely Phytoplankton, (PHY) Zooplankton (ZOO), Small Detritus (SDet) and Large Detritus (LDet).
The presented Corg fluxes refer to the total organic carbon pool, therefore to the sum of the four modeled pools. Of these organic carbon types, only ZOO does not sink (see Gruber et al, 2006 for a complete set of parameters). The NPZD model does not include any dissolved organic carbon pool. In spite of this limitation, the modeled SDet, with its small sinking speed and large abundance, shares strong similarities with a suspended POC pool and, to some extent, with semi-labile DOC. A full discussion of the consequences of this representation of the organic carbon pool in our model is provided in Lovecchio et al., 2017, and in the BGC Discussion page of the same paper, especially the answer to the Major Comment 1 of Referee #1.
With regard to benthic remineralization: sediments in the NPZD model are not a sink for POM but act as a temporal buffer, meaning that all the POM that sinks into the sediments is slowly remineralized there back into inorganic constituents, which are then released immediately back into the bottom water. No burial of POM is considered in our model. Thus sediments are indeed a source of nutrients for the water column in our model. A clear drawback is our lack of consideration of benthic denitrification and other special processes altering the biogeochemistry of the sediments and of the overlying water column.
In order to better clarify this point, we will add this statement to the Methods section:
"The modeled total $C_{org}$ pool consists of a non-sinking zooplankton class, a sinking phytoplankton class and two detritus pools, one slow and one fast sinking. Sediments in the NPZD model are not a sink for $C_{org}$ but act as a temporal buffer: all the material that sinks into the sediments is slowly remineralized then release back into inorganic constituents of the bottom layer. See Lovecchio et al., 2017 for a discussion of the strengths and limitations of the representation of the organic carbon fluxes in NPZD."

**Answers to specific comments**

**SC1)** *Page 1, line 4 – I would not say that a model will be able to "demonstrate quantitatively" that eddies and filaments are exporting organic matter from the coast to the open ocean at the NW Africa EBUE scale. A numerical model is a tool that tries to approach as much as possible to reality, but it is (rather) imperfect.*
**Answer:**
We will rephrase this sentence as follows.
"Yet a comprehensive analysis of this mesoscale flux and of its impact in the entire Canary Upwelling System (CanUS) has not been provided."

**SC2)** *Page 5, caption of Figure 1 – explain what is "01/12/0030" and change "9.5∘N and 32∘N" by "32∘N and 9.5∘N".*
**Answer:**
We thank Referee #3 for this comment, but we think that it's correct to list the latitudes in an increasing order. Day 01/12/0030 is the first day of the 12th month of the 30th year of simulation including spinup. We will rephrased this part of the caption as follows:
"from Dec. 1 of year 30 of the run (01/12/0030)."

**SC3)** *Page 7, line 13 – salinity is unit less*
**Answer:**
Thank you, we will correct this.

**SC4)** *Page 9, line 1 – It is a bit optimistic to state that the patterns of the model and the observations are "very similar". I would say just "similar" or even "roughly similar".*
**Answer:**
We will strike "very" from "very similar".

**SC5)** *Page 11, line 2 – Please, erase "found"*
**Answer:**
We will substitute the expression "abundant found" with "abundantly found"

**SC6)** *Page 11, line 18 – Please, indicate what organic matter pools are included in $C_{org}$ (Phyto?, Zoo?, large POM?, small POM + DOM?).*
**Answer:**
We will add a short description of the NPZD model following the first sentence of the methods, as proposed in the answer to the major comment:
"The modeled total $C_{org}$ pool consists in one non-sinking zooplankton class, one sinking phytoplankton class and two detritus pools, one slow and one fast sinking. Sediments in the NPZD model are not a sink for $C_{org}$ but act as a temporal buffer, meaning that all the material that sinks into the sediments is slowly remineralized there back into inorganic constituents. Please, refer to Lovecchio et al., 2017 for a discussion of the strengths and limitations of the representation of the organic carbon fluxes in NPZD."

**SC7)** *Page 12, Figure 5 – Why Corg was integrated through the entire water column instead of just in the upper 100 m. No scale for current vectors has been added in panel (a).*
**Answer:**
We will substitute the plots in Figure 5 with 100m-integrated organic carbon plots shown below in Figure 1.

[Figure]

***Figure 1:*** *Modeled 2-day mean variables for July 20 of year 33 of the simulation. The Corg components are integrated throughout the first 100 m depth.*

**SC8)** *Page 20, Figure 12 – The footnote is not coherent with the figure.*
**Answer:**
Thank you for pointing this out. We will correct the caption removing the reference to the subregions.

---

## Author Response (AR1)

**Author's response**

**"Mesoscale contribution to the long-range offshore transport of Organic Carbon from the Canary Upwelling System to the open North Atlantic"**

Elisa Lovecchio1, Nicolas Gruber1, Matthias Münnich1 1ETH-Zürich, Universitätstrasse 16, 8092 Zürich, Switzerland

Dear Prof. Gerhard Herndl,

thank you very much for taking our manuscript into further consideration for publication on Biogeosciences.

We have carefully revised our manuscript following your suggestions and the referees' comments. We include below our point by point answer to the referees including (in blue) the relevant changes to the manuscript for each point brought up by the referees, and a marked-up version of the manuscript. A considerable part of our efforts have gone in the direction of shortening the overall length of the manuscript and improve its readability, in compliance to the suggestions of the referees, while paying great attention not to cut on the scientific content and message.

We have also prepared a dedicated supplement in order to better document the structure and performance of our newly developed upwelling filament detection algorithm.

We look forward to your response. Sincerely, Elisa Lovecchio

**Answer to Referee #1**

We thank Referee #1 for the time spent on reviewing our manuscript and for his/her thoughtful comments on the pattern of the carbon fluxes and on the eddy influence on the organic carbon anomalies and fluxes, which have helped us to improve our manuscript. In order to address these comments, we have carried out further analysis with the aim to clarify some questions. We include below our detailed answers to the three major concerns raised by the Referee and to all the specific comments and describe the changes to the manuscript.

**Answers to Major comments**

**MC1) Pattern of the meridional and vertical organic carbon fluxes**

"One of these issues is that the mean vertical carbon flux presented in Figure 6c and 8c. There seems to be clear contrast between northern and southern regions separated at C. Blanc in plan view of the vertical carbon flux, which looks somewhat similar to the distribution of the mean meridional flux in Figure 6b. Also, in the vertical section, there is an upward flux at 60-70 m depths, right under and above the downward carbon fluxes. I'm puzzled by these mean vertical and meridional carbon flux distributions, and consider that this needs to be clarified."

**Answer to MC1**

As stated by Referee #1 the pattern of the mean meridional and vertical organic carbon fluxes (Figure 6 of the manuscript) differs strongly between the regions north and south of the Cape Verde front, according to the subregional differences in the circulation of the Canary Upwelling System, which include a large-scale anticyclonic circulation north of the front and a cyclonic circulation to the south (Arístegui et al. 2009; Pelegrí and Benazzouz, 2015; Pelegrí and Peña-Izquierdo 2015). The mean signature of the wind stress curl (negative north of the front, positive south of the front) also induces a change of sign in the mean vertical fluxes at the front, as in Figure 6c (see also Lovecchio et al. 2017, Figure 11, and the in-depth discussion of the subregional differences in the Corg fluxes). In the present paper, we quickly refer to these differences in our introduction (page 3, lines 9-13). In order to clarify the drivers of the mean flux pattern, we have modified the first sentence of section 4.2 as follows:

"The time-mean fluxes from the Reynolds decomposition (Figure 5a-c) clearly reflect the regional circulation and the windstress curl pattern that characterizes the CanUS (Arístegui et al., 2009; Pelegrí and Peña-Izquierdo, 2015; Lovecchio et al., 2017). While these mean fluxes dominate the total fluxes (see also: Lovecchio et al. (2017), their Figure 11), the turbulent fluxes contribute substantially to the total fluxes and to their divergence (Figure 5d-f)."

Also the alongshore average section of the vertical advective flux (Figure 8c of the manuscript) reflects the subregional differences described above, showing a composition of positive (southern CanUS) and negative (northern CanUS) mean vertical advective fluxes of organic carbon. We refer to this in page 16, line 5-6 of the manuscript, as: "The different signature of the wind stress curl and consequent Ekman pumping offshore in the northern and southern CanUS results, on average in the CanUS, in a mixed signature of the vertical transport in the open waters.". In order to better illustrate these differences we include in the present document the profile of the vertical advective flux calculated separately for the northern, central and southern CanUS (Figure 1). Furthermore we have modified the sentence mentioned above to:

"Offshore, the different sign of the wind stress curl (negative and positive, respectively) characterizing the latitudes located north and south of the Cape Verde front, results in an opposite sign of the time-mean vertical flux of organic carbon in the two zonal bands (see Figure 5c). These fluxes, averaged along the entire CanUS, give rise to an alternate pattern of upward and downward mean vertical transport."

*Figure 1:* Alongshore average sections of the vertical advective fluxes of organic carbon for the three CanUS subregions, as defined in the manuscript. The change of signature in the vertical flux offshore is in agreement with the change of sign of the wind stress curl in correspondence of the Cape Verde front.

**MC2) Analysis and discussion of the results of Reynolds decomposition and covariance**

"In some part, although the authors tried to explain the resultant interesting patterns obtained by the Reynolds decomposition and structure based extraction of eddy and filament contributions, they are often stated without sufficient evidence and would make readers feel that they are rather speculative. For example, the high covariance between anticyclonic eddies and carbon anomaly in northern offshore, is explained by the upwelling induced by the anticyclonic eddy spin down, without any analysis."

**Answer to MC2**

In order to address this comment we decided to make some changes in the Results section, keeping in mind also the suggestion of Referee #2 to shorten the manuscript where possible. We have therefore made the following changes:

- We agree with Referee #1 that there isn't enough evidence supporting the upwelling in ACE in the final stages of their life in the proximity of the Canary Archipelago (discussion of Figure 9). Animations of the model output suggest that this positive  $C_{org}$  anomaly in ACE is rather associated with the extremely sharp offshore gradient in the average  $C_{org}$  concentrations at these latitudes. This gradient, resulting in very low  $C_{org}$  concentrations in the open waters, makes it possible for ACE formed near the coast to still contain on average more carbon than the open waters. Moreover, some of the ACE that cross these latitudes interact with filaments generated south of the Archipelago, which are particularly rich in  $C_{org}$  due to the north-south positive gradient in the tracer concentrations. We have revised the discussion of Figure 9 according to this analysis and have changed the paragraph on page 17, lines 1-9 as follows:

"Deviations from the negative correlation of SSH' and Corg' can also be explained in terms of eddy anomalies. In the nearshore southern CanUS, the positive Corg' in ACE might stem by the shedding of eddies from the northward flowing Mauritanian Current, which favors the trapping of Corg -rich coastal waters in ACE at the time of their formation. At the northern CanUS boundary, the signature of the eddy Corg' is connected to the presence of incoming large eddies formed in the Azores current (Gaube et al., 2014, Figure 1), leading to ACE with positive Corg' and CE with negative Corg'. Also in the offshore waters of the northern CanUS, ACE have a positive Corg'. This is possibly due to the sharp offshore gradient in Corg at these latitudes, where coastally generated ACE can carry the elevated Corg to the offshore waters even though they are generally not very efficient in doing so." - To further reduce the speculative elements in the discussion of our results, we have shortened section 4.4, limiting it to the discussion to the contribution of mesoscale activity to the organic carbon stock, while moving Figure 12 to the Appendix. This change to the manuscript is also listed in the answer to the general comment of Referee #2. The discussion of the nutrients concentrations and production in the eddies (page 20, lines 1-14) is now limited to the following sentence: "Moreover, the two types of eddies are characterized by strong differences in their nutrient concentrations and in their production rates (see also Appendix: and Figure B8). This likely exacerbating the differences in Corg between CE and ACE, and especially for the long living ones."

**MC3) Manuscript structure: merging Summary and Synthesis with Conclusions**

"To be concise, section 6 and 7 could be combined into one section."

**Answer to MC3**

We have merged the Summary and conclusions into a single Conclusions section strongly limiting the summary, as most of the information is already highlighted in the abstract and discussion. Our new conclusions read as follows:

"Our study in the CanUS confirms that mesoscale processes contribute substantially to the longrange offshore transport of Corg through a combination of mean and turbulent transport. In particular, filaments drive the total offshore flux of Corg and its divergence in the nearshore, while far-reaching eddies, especially cyclones, extend this transport up to the middle of the gyre. The divergence of the mesoscale transport allows extra vertical export out of the productive layer and strongly contributes to the shaping of the pattern of nearshore net autotrophy and offshore net heterotrophy of the water column (Lovecchio et al., 2017). Even though the CanUS has moderate levels of mesoscale variability in comparison with other EBUS, the mesoscale contribution to the transport and to the fueling of the offshore biological activity strongly dominates in the nearshore 1000 km and amounts already to about 20% at larger offshore distances. This suggests that this mesoscale contribution may be even more crucial in other upwelling regions that have higher nearshore-generated mesoscale activity (Capet et al., 2013; Marchesiello and Estrade, 2006), such as the California Upwelling System (Nagai et al., 2015). The key role of mesoscale processes in the lateral Corg transport has several consequences. First, it implies that coarse global models may be unable to account for a great part of this flux out of the upwelling regions, possibly failing at reproducing the offshore rates of deep respiration and at fully capturing the three-dimensionality of the biological pump. Second, remote sensing approaches may underestimate this offshore transport. On the one hand, this may be due to the time-limited sampling owing to the frequent cloud cover preventing the detection of the chlorophyll and associated carbon content. On the other hand, because of our modeled filament transport deepening offshore up to a few tens of meters below the surface, i.e., below the detection level of satellites, leading to a potential underestimation of the actual offshore reach of filaments. Third, the relevant share of Corg found in the offshore region within mesoscale eddies also tells us that a fraction of the offshore biological activity is fueled discontinuously at the passage or death of such a structure. In particular, both the transit of an eddy, which is associated with enhanced vertical export (Subha Anand et al., 2017; Waite et al., 2016), and the death of an eddy provide a discontinuous, but substantial, input of carbon for the oligotrophic waters. While our study shows relevant levels of eddy productivity offshore, further analyses are needed to disentangle the pathway of new production and recycling of the Corg along long-lasting tracks of the northern and southern CanUS, and to understand the special role of mode water anticyclones in the budget and transport. Further studies can also help to better quantify the highly seasonal contribution of the many short filaments of the southern CanUS, of which little is known, and to investigate the role of the offshore transport of dissolved Corg, not included in our model."

**Answers to specific comments**

**P2L13** "Coastal filaments are narrow (

*Figure 2:* Residual fluxes of the Reynolds decomposition obtained using the time variable reference mean adopted in the manuscript, plotted on the same scales as the turbulent fluxes in each direction of the flow.

**P6L20** "First, the above reference mean SST field. . ." This definition of the filament, does not include any shape criteria, and allows the region between straight upwelling front and the coast to be detected as a filament. Is it okay?**

**Answer:** With the present question Referee #1 expresses two concerns regarding the filament mask: (a) the fact that the mask covers the nearshore upwelling band at most time steps, and (b) the lack of an explicit shape criteria for the filament identification. We address them separately.

(a) As stated, the filament mask often covers the nearshore upwelling band roughly corresponding to the first 50 km offshore. However, this does not affect our main results and conclusions significantly. In fact, the filament flux divergence in the nearshore is calculated taking differences between the flux at 100 km offshore (well defined filaments) and the flux at the coast (zero). In the discussion of the filament contribution to the tracer concentrations we avoid to show the filament tracer stock in the first 50 km (inserts of Figures 11, 12). We also do not refer to the nearshore fluxes (0-100 km offshore range) in our summary figure (Figure 15). For the remaining figures, we state that this potential limitation must be kept in mind (see also the discussion at p 21 lines 5-9 of the present manuscript). Operatively, it is actually difficult (if not impossible) to unequivocally identify the base of a filament, distinguishing it from the nearshore flow that feeds into the structure. Possible alternatives would be either to remove the first 50 km of the filament mask and attribute the flux to the NF-NE transport (resulting in discontinuous pattern in the fluxes), or somehow split this portion of the flux between filaments and NF-NE transport. We consider these

two methods more arbitrary than attributing the nearshore flux to the filament mask, while acknowledging the limitation in the manuscript.

(b) As regard to constraining the shape of the filaments, in the course of the analysis the filament mask was evaluated on the basis of both SST and surface Chlorophyll images (see Figure B1 for an example) with very satisfactory results and an extremely limited number of cases of overestimation of the filaments either in time or in space. Of the few cases identified, they were all confined to the southern CanUS latitudes. The coarsening of the reference-mean field before calculating the temperature anomalies and the choice of an appropriate threshold in temperature anomaly (see Methods, page 6, lines 18-25) assure in fact the identification of narrow cold structures. In order to provide a comprehensive description of the algorithm and of its sensitivity to the parameters, we have added a **supplement to the manuscript**, which includes a technical description of the algorithm and of its properties, and an evaluation of the algorithm performance.

**P7L21** "representation of turbulence" should need "geostrophic" after "of". The term "turbulence" throughout the manuscript sounds not adequate as it includes the meaning of three-dimensional microscale turbulence. The scale of the "turbulence" should be specified, or replace "turbulence" by "eddy" **Answer:** We have added "geostrophic" before "turbulence".

Answer. We have added geostrophic before turbulence.

**P8L1** "*The AVISO datasets*. . .". *The 1/4 degree is the resolution of the grid for mapping, the actual AVISO sea surface data's resolution should be even coarser.* **Answer:** Thank you for this comment, we now refer to the resolution of the AVISO grid

**P11L2** "small eddies are abundant found in the..." This should be "small eddies are abundantly found in the..." or "small eddies are abundant, and found in the..." **Answer:** Thank you, we have correted this typo.

**Figure 6c** *mean vertical flux Why is the mean vertical flux directed upward in a fairly wide region in the southern part, and vice versa for the northern part? Is it vertical flux of omega flux, in which horizontal flux can contribute through the sloping grid bottom faces?*

**Answer:** The mean vertical flux is directed according to the regional signature of the wind stress curl, which is negative north of the Cape Verde front and positive south of the front (see also answer to MC1 for a further description of the flux pattern in the region). This results in a mean downward and upward flux respectively north and south of the front. In the nearshore the effect of the mean upwelling becomes evident, resulting in an intense upward flux in the nearshore northern and central CanUS. In the southern CanUS, where the upwelling intensity is on average weaker and the slope is especially wide (Arístegui et al. 2009), downwelling cells are also visible in the nearshore. The flux is not calculated with the omega velocity, but with the purely vertical component of the velocity. In order to further clarify this point, we have added in the Methods section, following the description of the Reynolds decomposition:

"The vertical flux components were calculated using the purely vertical velocity from the model output"

Figure 7e What is the green color?

Answer

We have added the following to the figure caption:

"The green area shading results from the overlapping of opposing mean and turbulent flux contributions."

**P16L4** "Switching to the vertical fluxes, the differences between the mean and turbulent flux components are even more pronounced" and Figure 8c. I'm puzzled by the mean

vertical carbon flux. The bands of down-up-downwelling flux indicates the increase of carbon at 20 m and the decrease at 80 m depth on average, unless there is compensating lateral flux divergence. I don't understand how the Ekman pumping results in this structure.

**Answer:** Figure 8c shows the mean vertical component of the flux, averaged in the entire CanUS. As shown in Figure 6, the sign of the vertical component changes between northern and southern subregion. Therefore the mean plot shows a combination of downward organic carbon flux in the north and upward flux in the south. We refer to our answer to MC1 for a further explanation of the flux pattern, and for our proposed changes to the manuscript regarding this point.

**P16L30** "This signature is typical of that expected from an eddy-induced vertical displacement of the nutricline (McGillicuddy, 2016)." If this is the case, the eddy is linear Rossby waves, and not trapping to carry carbon when it propagates. I think eddy propagation speed c is a few centimeter per second, and velocity u > 0.2 m/s, so u/c>1, suggesting that eddy is nonlinear, which can hold a large volume of water inside to travel together westward, which is the dominant signal discussed in this manuscript, rather than wave induced isopycnal displacement.

**Answer:** Thank you for this comment. In the course of the manuscript revisions, this part has been shortened and the sentence is therefore not anymore in the final manuscript.

**P17L7** "In the offshore waters, ACE can also. . ." Is this mechanism same as the warm core eddy spin down, accompanied by weak upwelling at the center (e.g. Frictionally induced circulations and spin down of a warm core ring by Glenn R. Flierl and Richard P. Mied)? Do you have any evidence to support that this is the cause of positive C'org within ACE?

**Answer:** We thank Referee #1 for highlighting this passage and we refer to the answer to MC2 for our changes to the text.

**P22L3** "Between 100 km and 500 km from . . ." and Figure 15d The structure based estimate mean flux divergence between 100-500km is negative. Does it mean that the mean flux is removing the carbon from the shore? If so this makes sense, because Ekman transport from the coast should act to remove carbon from the shore. But the distance is somewhat too far from the coast. Why does it start from 100 km not 0 km? Also, in Figure 15, why does the Reynolds decomposition based mean flux divergence in 100-500 km show different results from the structure based results?

**Answer:** In Figure 15d the blue area does not represent the mean flux but the non-filament-noneddy (NF-NE) flux, which is different from a mean transport (Reynolds decomposition, Figure 15b). The two fluxes must not be confused, as they are defined differently and represent different processes. The mean Reynolds component of the flux refers to a mathematical definition of mean flux, as defined in the Methods section, which can include the contribution of recurrent structures. The NF-NE component is defined as everything which is not identified by our mesoscale structure identification masks and therefore does not include (for example) recurrent filaments. But it may include some turbulent transport which is not identified as an eddy or a filament (e.g., variable small-scale fronts offshore). For this reason, the Reynolds-decomposition-based divergence of the mean flux and the structure-identification-based NF-NE flux divergence should not be the same, but the divergence of the components must nevertheless sum to 100% of the total divergence. The plots in Figure 15 only refer to the offshore ranges of distances from 100km to 2000 km offshore, in which the divergence of the total offshore flux is positive and therefore receive organic carbon from the nearshore (see inserts of Figure 7a and Figure 14a, and the total offshore flux results previously presented in Lovecchio et al. 2017). Therefore, these plots have the aim of attributing the enhancement of the organic carbon availability offshore to the physical drivers. In the range of 0-100km offshore, instead, the divergence of the total offshore flux is negative,

removing organic carbon from the productive nearshore. We have clarified this point modifying the caption of Figure 15 as follows:

"Comparison between the results of the turbulence-based and structure-based methods for the entire EBUS in the range of 100km to 2000 km from the coast, where the divergence of the total offshore flux is positive, therefore resulting in an increase of the organic carbon availability (see also Lovecchio et al. 2017, Figure 10 for the divergence of the total offshore flux)."

The negative divergence of the NF-NE transport is determined by the fact that the non-mesoscale flux is weak at 100 km offshore, where most of the offshore transport is carried on by the filaments, while it intensifies moving towards 500 km from the coast.

We will clarify this point adding the following passage at the end of the first paragraph of section 4.5:

"An exception is the intensification of the NF-NE flux in the range between 100 km and 500 km, resulting in a negative offshore flux divergence (Figure 13)."

**P22L12** "...offshore reach due to the their..." remove "the". **Answer:** Thank you for pointing out this typo, we have corrected it.

**P22L19** ". . .mean deflection of their trajectories (Chelton et al., 2011)" The earlier study should be included "McWilliams and Flierl 1979: On the evolution of isolated, nonlinear vortices. Phys. Oceanogr., 1155–1182."

**Answer:** Thank you for this suggestion, we have included this reference.

**P22L28** "In particular, ACE are responsible for the northward . . . due to their relatively fast decay that results in a slowing down of the clockwise rotation while they move offshore, these eddies induce a net northward transport of Corg" I don't understand this explanation for the net northward carbon transport by ACE. Considering the positive carbon gradient in zonal direction with clockwise stirring, the southward transport should be stronger. How is this result related to Reynolds decomposed eddy meridional flux in Figure 6e?

**Answer:** The regional gradient in the organic carbon concentration is not only negative in the offshore direction (zonal gradient) but also positive southward, with a reduced zonal gradient at lower latitudes (Aristeguí 2009, Demarq and Somoue 2015). ACE rotate clockwise and stir this gradient while moving offshore. In the nearshore, where the zonal gradient dominates, it is indeed possible to see a negative signature in the meridional ACE flux (Figure B9c). Away from the nearshore, where the meridional gradient and the ACE decay become relevant, the southern edge of the eddy stirs the organic carbon northward. The relatively fast decay of ACE also determines that in each point of the domain, on average, the western edge of a nearshore (therefore on average younger and more intense) ACE stirs the gradient northward, while the eastern edge of an offshore (older) ACE stirs it southward more slowly, resulting in a net northward meridional flux. In order to clarify this point, we have modified the sentence to read:

"In particular, ACE are responsible for the northward recirculation of the  $C_{org}$  through the asymmetric stirring of the background regional gradient. This is a consequence of the unique combination of an offshore negative gradient (strongest in the nearshore) and a southward positive gradient in  $C_{org}$  (Arístegui et al., 2009). Due to their relatively fast decay and the slowing down of the clockwise rotation while they move offshore, the ACE induce a net northward transport of  $C_{org}$ . This is because a younger and more energetic ACE stirs the  $C_{org}$  northward in the nearshore areas, while an older and weaker ACE stirs it southward in the offshore areas. CE, being more stable along their tracks, have a weaker effect on the net meridional transport."

**P22L31** "In the vertical direction. . ." I don't understand why NE-NF vertical flux is the stronger than eddy components. Also do these vertical carbon fluxes reflect the lateral structure of the vertical flow itself or carbon distribution? For confirmation, the total

mean vertical flow at 100 m depth should be presented in appendix. The mean vertical flow should be largest at  $\sim O(10-5 \text{ ms}-1)$  along the coast with the internal Rossby radius as the lateral width, which is very narrow compared to the model domain.

**Answer:** We thank Referee #1 for raising this interesting question, which points in the direction of our most recent analysis of the physical fluxes inside of the eddies, currently in development for a further dedicated publication. In order to address this question we therefore refer to our current understanding of the mesoscale eddy vertical fluxes according to our model results. Animations of the velocities inside of the eddies at each time step show dipolar or quadrupolar patterns in the vertical fluxes of well-formed stable eddies. These patterns average to a dipole in positive and negative vertical velocities in both composite cyclones and anticyclones at 100 m, in nice agreement with high-resolution observations of the mesoscale velocities in eddies (e.g., Barcelo-Llull et al., 2017, JPO). The pattern of positive and negative vertical velocities averages nearly to zero throughout the eddy surface, therefore constituting a minor net contribution to the advective fluxes in well-developed eddies. As a consequence, the net eddy contribution to the vertical advective flux is mostly driven by the large scale forcing, and in particular by the pattern of the wind stress curl (see answer to MC1 for a detailed explanation of the vertical flux pattern in the region). Submesoscale fluxes are not explicitly resolved by our model due to the limited resolution of our grid in the offshore waters. The total vertical advective flux at 100 m is published in our previous publication on Biogeosciences "On the long range offshore transport of organic carbon from the Canary Upwelling System to the open North Atlantic", and is visible in Figure 11c. In order to clarify this point, we modified these lines according to read:

"In the vertical, eddies have a minor role in the transport of C org compared to the filaments, and their signature cannot be clearly distinguished from that of the NE-NF transport. Inspections of the model output reveal (not shown) that mesoscale vertical velocities in well-developed eddies present a dipolar or quadrupolar pattern of upwelling and downwelling similar to the one observed by insitu measurements (Barceló-Llull et al., 2017). This compensating nature of the vertical velocities results in a minor net contribution of these mesoscale structures to the vertical flux."

**P23L4** The convergence at filament is revisited with the theory of frontogenesis by "McWilliams et al. 2009, Cold filamentary intensification and oceanic surface convergence lines"

Answer: Thank you very much for this suggestion, we have added this reference to the manuscript.

**P25L15** "*Nagai et al. (2015) showed that this subduction happens primarily at the filament tip. . . " Remove "tip" and insert "periphery of" before filament* **Answer:** We have corrected the sentence accordingly.

**Answer to Referee #2**

We thank Referee #2 for the time spent on reviewing our manuscript and for his/her thoughtful comments about the length of the manuscript, the timescales of the turbulent variability, the filament detection algorithm and for the many specific comments which have helped us to improve our manuscript. We include below our detailed answers to the general comment about the writing and length and to all the specific comments and describe the proposed changes to the manuscript.

**General comment**

**Length of the manuscript and writing**

"...my only real concerns with this paper is its length. This contribution is significantly long (I acknowledge it contains many interesting results but some are a bit redundant) and the writing is slightly too didactic sometimes (it could often be more "to-the-point" and more condensed). For instance, it seems weird to have both "Summary and Synthesis" and "Conclusions" sections in the same article (although I enjoy the reading of both). I agree it has a lot of content (complex analyses & interesting well-supported results) but it may be "lost-at-sea" considering that, nowadays (and unfortunately), papers are more-and-more numerous and less-and-less read by the community. Their work could gain in visibility if they opt for shorter and/or separated contributions. I would suggest the authors to try reducing the length of the article (i) by focussing their analyses and the writing and (ii) maybe by keeping aside some content for another paper (especially the regional analyses over the three subsystems -keeping here only the whole CanUS- and the seasonality of the processes, cf. A comment below)."

**Proposed solution**

We thank Referee #2 for his/her evaluation of the manuscript. The length of the manuscript is indeed an issue and we will thrive to streamline and shorten it, to make it more accessible to the reader. In order to move in this direction, we have decided to combine the suggestions of Referee #1 (comments MC2, MC3, P17L7) and Referee #2 (this comment and specific comment SC6) and have made substantial changes to the manuscript in order to make it more concise. We list the changes below:

1 – We have shortened the Methods section by moving the (rather informal) description of the new filament detection algorithm to a dedicated **supplement to the manuscript**, where we provide also an evaluation of the algorithm sensitivity and of its performance.

2 - We have moved the Evaluation section by moving Figure 4 to the Appendix, as it does not provide a comparison with the observations, and rephrased the section in order to remove a few repetitions in the discussion of the Figures.

3 – We have shortened the discussion of our results, while paying attention not to cut out on the scientific content. In particular, have merged subsection 4.3 and subsection 4.4 into a single subsection titled "From turbulent anomalies to the mesoscale contribution to the organic carbon stock" which serves as a transition between the discussion of the Reynolds decomposition and the discussion of our structure-based analysis. This allows us to present: velocities, Corg anomalies and Corg content of the mesoscale structures. With the purpose to ease the flow of the paper, have inverted the order of Figure 9 and Figure 10 and merged the first and last paragraph of the current subsection 4.3 into a single initial paragraph, followed by the discussion of the current Figure 9. We have cut part of the current subsection 4.4, limiting the discussion to the contribution of mesoscale activity to the organic carbon stock, in line with the focus of our paper. Figure 12 (mesoscale contribution to the nutrient stock and production) has been moved to the Appendix. This is also in line with the suggestion of Referee #1 to limit speculative discussion in the Results section. 4 – We have removed from our discussion the paragraph focusing on the processes regulating production and nutrient concentrations in the eddies according to previous literature (pg.27, ll.11-23), in order to maintain the focus on the organic carbon budget and transport.

5 - We have merged the Summary and conclusions into a single Conclusions section strongly limiting the summary, as most of the information is already highlighted in the abstract and discussion. Our new conclusions read as follows:

"Our study in the CanUS confirms that mesoscale processes contribute substantially to the longrange offshore transport of Corg through a combination of mean and turbulent transport. In particular, filaments drive the total offshore flux of Corg and its divergence in the nearshore, while far-reaching eddies, especially cyclones, extend this transport up to the middle of the gyre. The divergence of the mesoscale transport allows extra vertical export out of the productive layer and strongly contributes to the shaping of the pattern of nearshore net autotrophy and offshore net heterotrophy of the water column (Lovecchio et al., 2017). Even though the CanUS has moderate levels of mesoscale variability in comparison with other EBUS, the mesoscale contribution to the transport and to the fueling of the offshore biological activity strongly dominates in the nearshore 1000 km and amounts already to about 20% at larger offshore distances. This suggests that this mesoscale contribution may be even more crucial in other upwelling regions that have higher nearshore-generated mesoscale activity (Capet et al., 2013; Marchesiello and Estrade, 2006), such as the California Upwelling System (Nagai et al., 2015). The key role of mesoscale processes in the lateral Corg transport has several consequences. First, it implies that coarse global models may be unable to account for a great part of this flux out of the upwelling regions, possibly failing at reproducing the offshore rates of deep respiration and at fully capturing the three-dimensionality of the biological pump. Second, remote sensing approaches may underestimate this offshore transport. On the one hand, this may be due to the time-limited sampling owing to the frequent cloud cover preventing the detection of the chlorophyll and associated carbon content. On the other hand, because of our modeled filament transport deepening offshore up to a few tens of meters below the surface, i.e., below the detection level of satellites, leading to a potential underestimation of the actual offshore reach of filaments. Third, the relevant share of  $C_{org}$  found in the offshore region within mesoscale eddies also tells us that a fraction of the offshore biological activity is fueled discontinuously at the passage or death of such a structure. In particular, both the transit of an eddy, which is associated with enhanced vertical export (Subha Anand et al., 2017; Waite et al., 2016), and the death of an eddy provide a discontinuous, but substantial, input of carbon for the oligotrophic waters. While our study shows relevant levels of eddy productivity offshore, further analyses are needed to disentangle the pathway of new production and recycling of the Corg along long-lasting tracks of the northern and southern CanUS, and to understand the special role of mode water anticyclones in the budget and transport. Further studies can also help to better quantify the highly seasonal contribution of the many short filaments of the southern CanUS, of which little is known, and to investigate the role of the offshore transport of dissolved Corg, not included in our model."

6 - We have maintained the discussion of the subregional differences in the fluxes in the manuscript. These results provide essential information on the latitudinal differences in the flux contributions within the system, and can be of interest for studies that will focus on a specific portion of the CanUS. Moreover, they provide a term of comparison between our results and several previous studies which focused on a specific subregion.

Overall, the proposed changes have allowed us to shorten the manuscript core (from Abstract to Conclusions) by more than 10% of its length, i.e. from the initial 31 pages to 27 pages.

**Answers to specific comments**

2) p 5 line11: delete "the";

**SC1)** *Typing/English mistakes: 1) p 1 line 19: "anticyclonic" has been perhaps forgotten;*

3) p 6 line 19: filaments;

4) p 9 line 4: delete "eddy";

5) p 11 line 2: delete "abundant" or "found";

6) p 11 line 25: delete "a";

7) p 16 line 11: could it be "going deeper than 500 m within 500 km from the shore (or over 0-500 km offshore)";

8) p 17 line 8: "which slow down the rotation...";

9) p 18 line 14: should it refer to Fig. B7 and/or 8 instead of Fig. B6 there?;

10) p21 line 1: estimateS;

11) p 22 line 33: responsible OF a;

12) p 30 line 18: cyclonEs; etc..

**Answer:**

1) the sentence is actually complete as it refers to all types of eddies, and not anticyclonic eddies.

2) the paragraph was rephrased while revising the manuscript.

3) we have moved this paragraph to the supplement.

4) the entire paragraph was rephrased while revising the manuscript.

5) the entire paragraph was rephrased while revising the manuscript.

6) we have deleted "a" as in: "Thin and short-lived filaments...".

7) the entire paragraph was rephrased while revising the manuscript.

8) we decided to remove this sentence.

9) thank you for pointing this out, yes it should be the plot B8 of the current manuscript version, we have corrected it accordingly.

10) we have corrected this typo.

11) the entire paragraph was rephrased while revising the manuscript.

12) thank you, we will correct the typo.

**SC2)** Page 2 lines 9-10: there exist plenty of earlier references to support this statement, including some that are already cited, such as Rossi et al. 2008 and Gruber et al. 2011. **Answer:**

Thank you very much, we have added the references in this point.

**SC3)** Page 2 lines 31-32 and page 30 lines 32-33 + page 31 lines 1-2: the "coherence" of eddies is still an open question to my mind; there is a bunch of papers around those questions, both from the observational and modelling point of views, and many situations (from very coherent to quite leaky) have been reported; so it seems no agreement has been reached yet. Please rewrite.

**Answer:**

We agree with Referee #2 that significant progress still has to be made to understand the degree of isolation of mesoscale eddies from the surrounding environment.

We have rephrased the sentences as follows:

"Eddies trap water and tracers in their core during their formation. In stronger eddies, the degree of lateral isolation of the eddy core from the surrounding environment can be quite high, possibly resulting in the entrainment and long-range transport of trapped tracers at formation (Karstensen et al., 2015, 2017, Chelton et al., 2011)."

"Third, the relevant share of C org found in the offshore region within mesoscale eddies also tells us that a fraction of the offshore biological activity is fueled discontinuously at the passage or death of such a structure."

**SC4)** Page 4 line 11: in additi